# The potential of epigenetic therapy to target the 3D epigenome in endocrine-resistant breast cancer

Joanna Achinger-Kawecka [1,2,8] ✉, Clare Stirzaker [1,2,8], Neil Portman[1,2], Elyssa Campbell [1], Kee-Ming Chia[1], Qian Du [1,2], Geraldine Laven-Law[3], Shalima S. Nair[1], Aliza Yong[1], Ashleigh Wilkinson[1], Samuel Clifton[1], Heloisa H. Milioli [1,2], Sarah Alexandrou [1,2], C. Elizabeth Caldon [1,2], Jenny Song[1], Amanda Khoury[1,2], Braydon Meyer[1], Wenhan Chen[1], Ruth Pidsley[1,2], Wenjia Qu[1], Julia M. W. Gee [4], Anthony Schmitt[5], Emily S. Wong [6,7], Theresa E. Hickey[3], Elgene Lim[1,2] & Susan J. Clark [1,2] ✉

Three-dimensional (3D) epigenome remodeling is an important mechanism of gene deregulation in cancer. However, its potential as a target to counteract therapy resistance remains largely unaddressed. Here, we show that epigenetic therapy with decitabine (5-Aza-mC) suppresses tumor growth in xenograft models of pre-clinical metastatic estrogen receptor positive (ER+) breast tumor. Decitabine-induced genome-wide DNA hypomethylation results in large-scale 3D epigenome deregulation, including de-compaction of higher-order chromatin structure and loss of boundary insulation of topologically associated domains. Significant DNA hypomethylation associates with ectopic activation of ER-enhancers, gain in ER binding, creation of new 3D enhancer–promoter interactions and concordant up-regulation of ER-mediated transcription pathways. Importantly, long-term withdrawal of epigenetic therapy partially restores methylation at ER-enhancer elements, resulting in a loss of ectopic 3D enhancer–promoter interactions and associated gene repression. Our study illustrates the potential of epigenetic therapy to target ER+ endocrine-resistant breast cancer by DNA methylation-dependent rewiring of 3D chromatin interactions, which are associated with the suppression of tumor growth.

Approximately 70% of breast cancers are driven by estrogen receptor-alpha (ERα). ERα is a critical ligand-activated transcription factor that controls breast cancer cell proliferation and tumor growth upon exposure to estrogenic hormones[1]. Drugs that target ERα pathways are highly effective in the treatment of ER+ breast cancer; however, de novo or acquired resistance to these agents (endocrine resistance) affects a large proportion (>30%) of patients and is the major cause of breast cancer mortality. Endocrine resistance has previously been shown to be associated with epigenetic alterations, including DNA methylation, chromatin accessibility, histone modifications and binding of different

[1]Garvan Institute of Medical Research, Sydney, New South Wales, Australia. [2]School of Clinical Medicine, Faculty of Medicine and Health, UNSW Sydney, Sydney, New South Wales, Australia. [3]Dame Roma Mitchell Cancer Research Laboratories, Adelaide Medical School, University of Adelaide, Adelaide, South Australia, Australia. [4]Breast Cancer Molecular Pharmacology Group, School of Pharmacy and Pharmaceutical Sciences, Cardiff University, Cardiff, Wales, UK. [5]Arima Genomics, Inc., Carlsbad, CA, USA. [6]Victor Chang Cardiac Institute, Sydney, New South Wales, Australia. [7]School of Biotechnology and Biomolecular Sciences, UNSW Sydney, Sydney, New South Wales, Australia. [8]These authors contributed equally: Joanna Achinger-Kawecka, Clare Stirzaker. ✉e-mail: j.achinger@garvan.org.au; s.clark@garvan.org.au

transcription factors[2]. In particular, differential ER transcription factor binding leads to altered expression of estrogen-responsive genes in endocrine-resistant breast cancer[1] and is associated with clinical response to endocrine therapies[3,4].

Epigenetic alterations also influence the 3D genome architecture, from the local level of chromatin interactions to the higher level organization of topologically associated domains (TADs) and chromosome compartments[5]. Although cancer cells maintain the general pattern of 3D genome folding, distinctive structural changes occur in cancer genomes at all levels of 3D organization. The 3D genome structure is also disrupted in endocrine-resistant ER+ breast cancer cells[6,7], notably through long-range chromatin changes at ER-enhancer binding sites that are DNA hypermethylated in resistant cells[7].

DNA demethylating agents such as decitabine (5-aza-2′-deoxycytidine) have emerged as promising therapeutic strategies for treating various cancers[8]. Decitabine is approved by many international regulatory agencies, including the US Food and Drug Administration and the European Commission, for treating hematological cancers[8]. In solid cancers (including colorectal and ovarian cancer), decitabine has been shown to demethylate regulatory regions that result in the re-activation of tumor suppressor genes[9,10]. Additionally, treatment with DNA demethylating agents can stimulate immune response pathways in cancer cells through increased transcription of DNA repeat elements, which induces a viral mimicry response[11,12]. However, the direct effect of epigenetic drugs on the tumor cells, including epigenome and 3D genome structure, remains largely unexplored, especially in clinically relevant patient-derived model systems or clinical samples.

To elucidate the mechanism of epigenetic therapy with decitabine, we assessed the molecular consequences of treatment on DNA methylation, 3D genome architecture and transcriptional programs in endocrine-resistant ER+ patient-derived xenograft (PDX) models. Our data revealed that decitabine treatment inhibited tumor growth and resulted in DNA hypomethylation that was associated with 3D epigenome remodeling, gain in ER binding and activation of ER-mediated transcription, highlighting the potential of epigenetic therapy for the treatment of ER+ endocrine-resistant breast cancer.

## Results

### Decitabine inhibits tumor growth and decreases cell proliferation

To study the efficacy of epigenetic therapy in the context of endocrine-resistant ER+ breast cancer and to establish its impact on the 3D genome and epigenome, we used two different PDX models (Gar15-13 and HCI-005) (Fig. 1a) (see Methods). Gar15-13 and HCI-005 PDXs were derived from the metastases of patients who were ER+ and had disease progression following one or more lines of endocrine therapy. These models have been used for several pre-clinical studies[13–15].

Using a low, well-tolerated and non-cytotoxic dose of decitabine (0.5 mg kg$^{-1}$; Extended Data Fig. 1a), we first interrogated the anti-cancer effect of epigenetic therapy on tumor growth. Following tumor implantation and an initial period of growth (to a volume of 150–200 mm$^3$), mice were randomized to twice-weekly injections of PBS (vehicle) or 0.5 mg kg$^{-1}$ decitabine. Treatment continued with twice-weekly measurements of tumor volume for 35 days or until tumor volume exceeded 1,000 mm$^3$. At the endpoint, mice were culled and tumor material was collected for analysis. In both Gar15-13 and HCI-005 PDX models, decitabine treatment elicited a strong growth-inhibitory response (Fig. 1b,c) and a significant reduction in proliferative index at endpoint (Fig. 1d). No significant change was found in the proportion of ER+ cells; however, a small but significant reduction in nuclear ER staining with decitabine treatment was observed (Extended Data Fig. 1b). Importantly, our genetic and epigenetic analyses showed that a high degree of intra-tumor clonal heterogeneity was retained following decitabine treatment in both PDX models (see Supplementary Note and Supplementary Tables 1 and 2).

### Decitabine induces hypomethylation and enhancer activation

To determine whether decitabine treatment induced alterations in the DNA methylome of the PDX tumors, we used Infinium EPIC Methylation arrays on four biological replicates of vehicle-treated and decitabine-treated PDX tumors at endpoint. All decitabine-treated tumors exhibited genome-wide DNA methylation loss (Extended Data Fig. 1c,d), with Gar15-13 tumors showing more hypomethylation than HCI-005 tumors (Fig. 1e and Extended Data Fig. 1e) (average methylation difference of 14.55% and 8.74%, respectively). To characterize the extent and location of genome-wide DNA methylation loss, we identified differentially methylated regions (DMRs) between vehicle-treated and decitabine-treated Gar15-13 and HCI-005 tumors (Supplementary Table 3). We found that the hypomethylated DMRs in both PDX models were mainly located at non-coding genomic regions (introns and intergenic) (Extended Data Fig. 1f,g) and were significantly enriched at putative enhancer regions (Fig. 1f and Extended Data Fig. 1h) (permutation test, $P < 0.001$). In agreement, there was extensive DNA hypomethylation at putative enhancers in Gar15-13 tumors (approximately 18.38% change in median DNA methylation; Extended Data Fig. 2a) and HCI-005 tumors (approximately 9.24% change in median DNA methylation; Extended Data Fig. 2b), whereas promoters were less demethylated (approximately 10.12% in Gar15-13 and 2.16% in HCI-005; Extended Data Fig. 2c,d).

Finally, to establish whether decitabine-induced DNA hypomethylation results in the activation of enhancers, we profiled active enhancer histone mark H3K27ac in three vehicle-treated and three decitabine-treated Gar15-13 tumors using CUT&RUN[16] (Supplementary Table 4). We identified 17,909 gained and 1,706 lost H3K27ac peaks in decitabine-treated tumors (Fig. 1g and Extended Data Fig. 2e). Notably, gained H3K27ac peaks were located mainly at distal regulatory elements (Extended Data Fig. 2f) and were enriched at DNA hypomethylated DMRs (Fig. 1h), suggesting that demethylation was associated with ectopic enhancer activation.

---

**Fig. 1 | Decitabine inhibits tumor growth and induces widespread DNA hypomethylation. a**, Schematic of study design. Created with Biorender.com. WGBS: whole-genome bisulfite sequencing; TF, transcription factor **b**, Gar15-13 PDX growth curves for vehicle-treated (100 nM PBS, $n = 7$ mice) and decitabine-treated (0.5 mg kg$^{-1}$, $n = 7$ mice) tumors. Data are represented as mean ± s.e.m. and analyzed using a two-tailed, unpaired Student's $t$-test at the ethical or experimental endpoint. *$P < 0.001$. Endpoint test details are $t = 5.678$, df = 8, $P = 0.0009$. **c**, HCI-005 PDX growth curves for vehicle-treated (100 nM PBS, $n = 8$ mice), and decitabine-treated (0.5 mg kg$^{-1}$, $n = 7$ mice) tumors. Data are represented as mean ± s.e. and analyzed using a two-tailed, unpaired Student's $t$-test at the ethical or experimental endpoint. *$P < 0.001$. Endpoint test details are $t = 5.231$, df = 9, $P = 0.0001$. **d**, Ki-67 positivity at endpoint in Gar15-13 and HCI-005 PDXs. Data were analyzed using a two-tailed, unpaired Student's $t$-test. *$P < 0.001$. Endpoint test details are $t = 4.748$, df = 11, $P = 0.0006$ and $t = 4.698$, df = 12, $P = 0.0005$ for Gar15-13 and HCI-005, respectively. **e**, Distribution of DNA methylation for vehicle-treated and decitabine-treated Gar15-13 PDXs ($n = 4$ biological replicates each). Box plots show median, interquartile range and maximum–minimum. Data were analyzed using the two-sided $Z$-test. **f**, O/E fold change enrichment of DMRs in Gar15-13 decitabine compared to vehicle across TAMR ChromHMM regulatory regions. *$P < 0.001$ (permutation test). Numbers located within each specific region are presented in the respective column. **g**, Overlap of consensus H3K27ac peaks between vehicle-treated and decitabine-treated Gar15-13 PDXs ($n = 3$ biological replicates each). Average signal intensity of H3K27ac at gained and lost H3K27ac binding sites in Gar15-13 PDXs. **h**, O/E fold change enrichment of hypomethylated DMRs in Gar15-13 decitabine compared to vehicle across gained and lost H3K27ac peaks. *$P < 0.001$ (permutation test). The numbers located within each specific region are presented in the respective column.

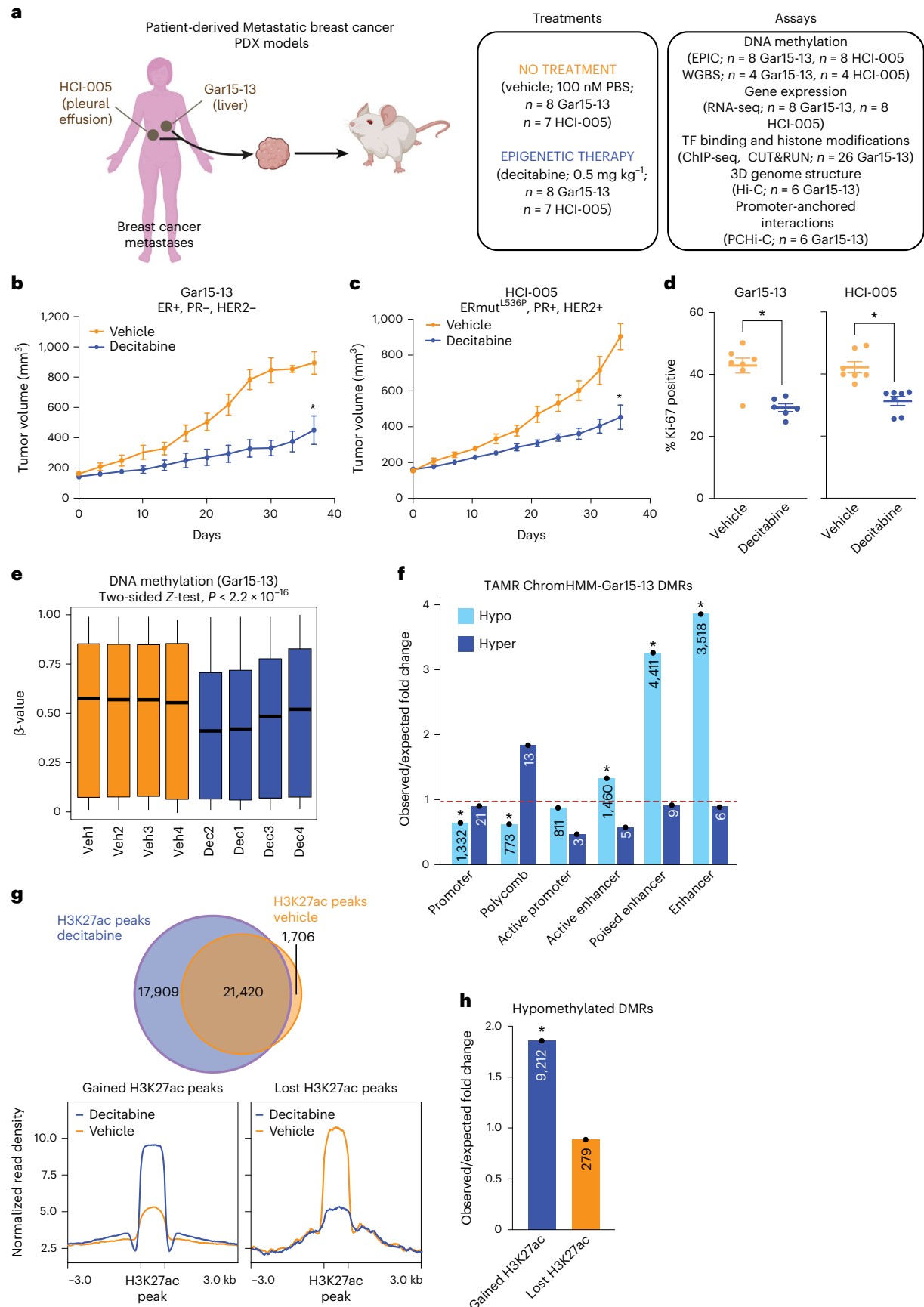

## Decitabine induces activation of transposable elements

We next evaluated genome-wide DNA methylation levels at different classes of transposable elements[17]. We observed genome-wide loss of DNA methylation at all transposable element sub-groups (Extended Data Fig. 2g and Extended Data Fig. 2h) with ~12% loss of DNA methylation in decitabine-treated tumors. Additionally, we observed transposable element expression alterations and activation of anti-viral signaling (Supplementary Note), previously reported in other cancers[11,12]. Notably, the extent of DNA hypomethylation measured at transposable elements was less than genome-wide and significantly less than at enhancer regions (Extended Data Fig. 2a,b).

## Loss of DNA methylation results in 3D genome de-compaction

To determine whether decitabine-induced DNA hypomethylation also leads to global changes in 3D genome architecture, we analyzed in situ Hi-C performed on vehicle-treated and decitabine-treated tumors in Gar15-13 PDX in triplicate (see Supplementary Note and Supplementary Table 5). First, to detect open (active) and closed (silent) genomic compartments (A and B, respectively) we performed PCA analysis of the Hi-C data[18]. We compared the eigenvalues between vehicle-treated and decitabine-treated tumors and observed that although most bins retained the same compartment status between samples (either A to A or B to B), a large number of bins in the decitabine-treated tumors became more A-type compared to the vehicle-treated tumors (that is, B-to-A switch) (Fig. 2a and Extended Data Fig. 3a). We quantified compartment switching and identified 643 compartments that switched assignment (Fig. 2b), with 64% of changes involving compartment activation (B-type to A-type) (Fig. 2b). Notably, we observed significant DNA hypomethylation at B-to-A switches, while A-to-B switched regions maintained similar DNA methylation levels (Fig. 2c). Using RNA-seq data (Supplementary Table 6 and 7), we detected an overall increase in expression of genes located at regions that switched their assignment from B to A in decitabine-treated tumors, whereas genes located at A to B switching compartments did not significantly change expression (Extended Data Fig. 3b). The newly activated compartments hosted 87 genes with increased expression and 21 genes displaying decreased expression (Extended Data Fig. 3c,d and Supplementary Table 8). The upregulated genes were significantly enriched at B-to-A switching compartments (2.2-fold observed over expected (O/E), $P < 0.001$). Furthermore, we found significantly decreased interaction strength between closed compartments (B–B interactions; two-tailed Student's $t$-test, $P = 0.025$), no change in contacts between active compartments (A–A interactions; two-tailed Student's $t$-test, $P = 0.26$) and increased contacts between A and B compartments (two-tailed Student's $t$-test, $P = 0.011$) (Fig. 2d,e and Extended Data Fig. 3e,f). Gained A-compartment interactions were also significantly enriched for stable A compartments (O/E = 1.7, $P < 0.001$), suggesting increased interactivity between new A compartments and stable A compartments.

Secondly, we investigated the impact of decitabine treatment on the organization of TADs. We observed a significant decrease in average TAD insulation score in decitabine-treated compared to vehicle-treated tumors (~36.53 in decitabine and ~46.74 in vehicle) (Fig. 2f and Extended Data Fig. 4a). Consistent with loss of TAD boundaries and potential merging of TADs, the total number of TADs was decreased in decitabine-treated samples (Fig. 2g) and their corresponding average domain size increased (two-sided Student's $t$-test, $P = 0.0289$; Extended Data Fig. 4b). Analysis of differential TAD boundaries revealed that a large percentage (43.2%) of vehicle-specific boundaries were lost in decitabine-treated tumors (Fig. 2h), characterized by a decreased average insulation score (Extended Data Fig. 4c). However, we found no significant association between differential TAD boundaries, differential DNA methylation and differential gene expression (see Supplementary Note). We next evaluated whether the change in TAD boundary insulation in response to decitabine treatment is caused by a change in CTCF binding occupancy. We performed CTCF CUT&RUN in vehicle-treated and decitabine-treated Gar15-13 tumors (three replicates each) (Supplementary Note and Supplementary Table 4). First, we used Diffbind to identify differential CTCF binding sites after decitabine treatment and found 872 gained and 34 lost CTCF peaks with decitabine treatment (false discovery rate, FDR < 5%) (Extended Data Fig. 4d,e). We found that common CTCF sites were significantly enriched at unaltered TAD boundaries (Extended Data Fig. 4f). However, altered (decitabine-specific or vehicle-specific) TAD boundaries were not enriched for gained or lost CTCF binding sites (Extended Data Fig. 4f,g). We exemplify one such region in which a TAD was lost in decitabine-treated tumors concomitant with a loss of boundary insulation and no change in CTCF binding occupancy at the altered TAD boundary (Fig. 2i; further examples in Extended Data Fig. 4h,i).

Together, these results indicate that DNA hypomethylation induced by decitabine treatment in vivo leads to significant de-compaction of 3D chromatin architecture, with reduced B-type compartments, increased interactions within A-type compartments and concomitant increase in regional gene expression. Although most TADs maintained their structure after decitabine treatment, their boundaries became less insulated, with no significant change in CTCF occupancy at the altered TAD boundaries, suggesting increased intra-tumor heterogeneity in TAD structure and loss of some TAD boundaries at the regions of chromosomal compartment de-compaction.

## Loss of DNA methylation alters 3D enhancer–promoter wiring

To gain insights into chromatin interactions at the level of individual promoters and enhancers, we investigated genome-wide promoter-anchored contacts in three decitabine-treated and three vehicle-treated tumors using Promoter Capture Hi-C (PCHi-C) (Supplementary Note and Supplementary Table 9), which allows for a significant increase in the sequencing coverage of promoter-anchored interactions compared to Hi-C (Fig. 3a). We show that promoter (bait) regions were significantly enriched for active and poised promoters as well as active ChromHMM enhancer states in both vehicle-treated and decitabine-treated tumors (Fig. 3b). Notably, putative enhancer

---

**Fig. 2 | Loss of DNA methylation results in de-compaction of chromatin.**
**a**, Correlation between average eigenvalues per bin in vehicle-treated and decitabine-treated Gar15-13 PDX tumors. **b**, Top panel: distribution of stable (A to A; B to B) and switching (A to B; B to A) compartments in decitabine-treated Gar15-13 tumors compared to vehicle-treated tumors. Bottom panel: distribution of different types of switching compartments (A to B; B to A) in decitabine-treated tumors compared to vehicle-treated tumors. **c**, DNA methylation levels at compartment regions that switched their assignment from B to A and from A to B in decitabine-treated ($n = 4$ biological replicates) and vehicle-treated ($n = 4$ biological replicates) PDX tumors. Black line indicates median ± s.d. Box plots show median, interquartile range and maximum–minimum DNA methylation. Data were analyzed using the two-sided $Z$-test. **d**, Average contact enrichment (saddle plots) between pairs of 50 kb loci arranged by their PC1 eigenvector in vehicle-treated and decitabine-treated tumors. Average data from $n = 3$

biological replicates shown. The numbers at the center of the heatmaps indicate compartment strength calculated as the $\log_2$ transformed ratio of (A–A + B–B) / (A–B + B–A) using the mean values. **e**, Saddle plots calculated using the averaged PC1 obtained from vehicle-treated ($n = 3$ biological replicates) and decitabine-treated ($n = 3$ biological replicates) tumors. **f**, Density plot of insulation scores calculated in vehicle-treated and decitabine-treated tumors. **g**, Number of TADs identified in vehicle-treated and decitabine-treated ($n = 3$ biological replicates each) PDX tumors. **h**, Overlap between TAD boundaries identified in vehicle-treated and decitabine-treated tumors. **i**, Snapshot of region on chromosome 1, showing vehicle-treated and decitabine-treated tumor Hi-C matrixes. Loss of a TAD in decitabine-treated samples is indicated with an arrow, concomitant with decreased insulation at that region. Merged Hi-C data from replicates ($n = 3$) at 10 kb resolution. Merged CTCF CUT&RUN signal shown below.

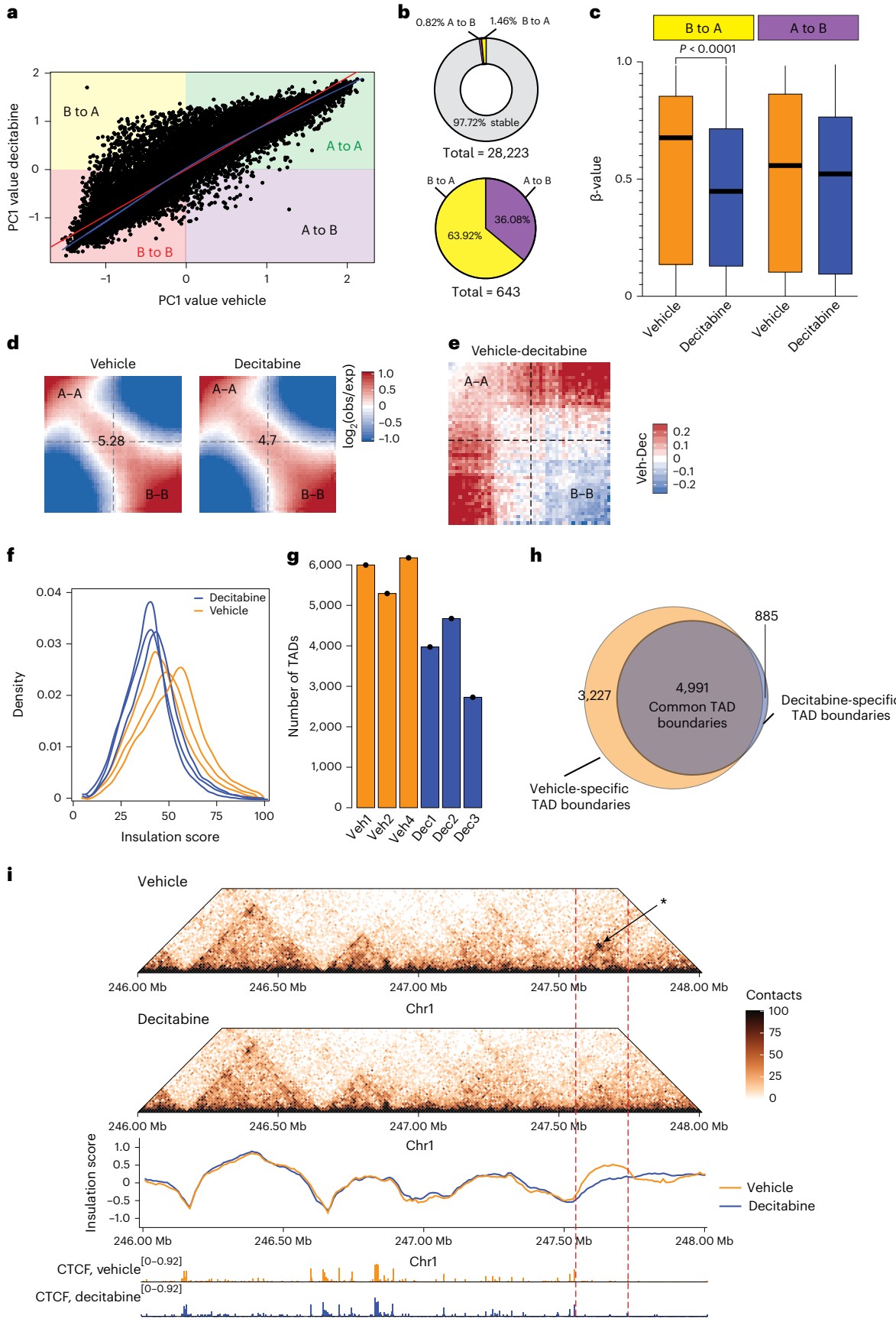

other-end (OE) interacting regions (that is enhancer OEs; exemplified in Fig. 3a) showed significant differential enrichment, whereby active promoters were enriched in vehicle-treated tumors and enhancers were enriched in decitabine-treated tumors (Fig. 3b).

To directly identify differential promoter-anchored interactions, we used the Chicdiff[19] pipeline (see Methods). In total, we found 13,088 stable and 4,111 dynamic (gained or lost) contacts for promoters and 55,186 stable and 26,912 dynamic contacts for enhancer OEs (Fig. 3c). The majority of promoter regions were common between the decitabine-treated and vehicle-treated tumors; however, decitabine treatment resulted in a large gain in the number of dynamic enhancer OEs, while only a small number of enhancer OEs were lost (24,694 gained and 2,218 lost with decitabine treatment) (Fig. 3c). Additionally, gained enhancer OEs were significantly enriched for gained H3K27ac binding sites (Extended Data Fig. 5a). Notably, interactions at gained enhancer OEs with decitabine treatment were associated with longer interaction distances compared to those that were maintained or lost (Fig. 3d), consistent with an increased number of long-range interacting enhancers connecting to these promoters. We then compared the total number of unique promoter and enhancer OEs involved in interactions between vehicle-treated and decitabine-treated tumors and found a significant increase in the total number of identified enhancer OEs in decitabine tumors while the interacting promoters remained the same (Extended Data Fig. 5b,c). On average, we detected 3.73 unique enhancer OEs per promoter in vehicle samples and 7.06 unique enhancer OEs per promoter in decitabine samples (Fig. 3e). We then calculated the number of interacting enhancer OEs for each individual promoter in vehicle-treated and decitabine-treated tumors (Fig. 3f). We found that the majority of interacting promoters in vehicle-treated tumors showed a large gain of enhancer OEs in decitabine-treated tumors, suggesting reprogramming of one-to-many enhancer–promoter interactions. Furthermore, we identified gained multi-way interactions that had, on average, significantly higher CHiCAGO scores in decitabine-treated tumors compared to vehicle (Wilcoxon $P < 0.001$) (Extended Data Fig. 5d), consistent with an overall increase in the total number of interactions with decitabine treatment. We found that the gain of interactions was associated with a shift from B-compartment assignment toward compartment A in decitabine-treated tumors (~76%) (Fig. 3g and Extended Data Fig. 5e), while a loss of interactions was associated with the switch from A-type to B-type assignment (~80%). This was particularly pronounced at lost interactions involving promoter bait regions (>90% switched from A to B) (Fig. 3g). Together, these results support that 3D chromatin interactions are rewired following decitabine-induced DNA methylation loss, leading to increased promoter-anchored interactions involving multiple enhancers connecting to gene promoters.

## Altered transcription and gain in ER binding at enhancers

To examine the transcriptional consequences of decitabine-induced rewiring of 3D chromatin interactions, we next analyzed RNA-seq data corresponding to four replicates of decitabine-treated and vehicle-treated Gar15-13 tumors. Gene set enrichment analysis (GSEA)[20] of all expressed genes revealed that decitabine treatment negatively correlated with gene signatures of cell proliferation and cell cycle (E2F targets, G2M checkpoint and Myc targets) (Fig. 4a) as well as genes involved in viral mimicry response (see Supplementary Note). Surprisingly, decitabine treatment also enriched for multiple hallmarks related to hormone signaling (estrogen response early and estrogen response late) (Fig. 4a) and up-regulation of a significant proportion of genes belonging to the 'estrogen response' hallmark (Extended Data Fig. 6a).

To directly address whether rewired enhancer–promoter interactions are involved in altered transcription, we identified genes connected to newly gained enhancer OEs and compared their average expression between vehicle-treated and decitabine-treated tumors. We identified a total of 4,025 genes at new enhancer–promoter interactions (Supplementary Table 10), of which 417 were upregulated after decitabine treatment ($P < 0.05$; log(fold change) > 1) (Fig. 4b). Upregulated genes were significantly enriched at gained OE interactions as compared to all genes (Fisher's exact test, $P < 2.2 \times 10^{-16}$). Our data suggest that the dynamic increase in the number of enhancer OEs connected to a promoter results in an overall increase in the expression of genes, in agreement with the current models of transcriptional control through enhancer–promoter interactions[21].

To further explore the specific role of rewired interactions in the altered transcriptional program, we evaluated which transcription factors are associated with these gained interactions. Notably, key transcription factors involved in ER+ breast cancer were highly enriched, including methylation-sensitive estrogen response elements (EREs) and ELF5 (ETS transcription factor family members), as well as architectural proteins CTCF and ZNF165 (Fig. 4c). Additionally, we compared transcription factor motifs enriched at gained interactions (Fig. 4c) to those enriched at DNA hypomethylated DMRs (Extended Data Fig. 6b) and found a number of overlapping motifs (CTCF, ERE, PBX and NRF1), with an addition of known methylation-sensitive transcription factors (AP1, Jun, NRF1[22]) and pioneer factors for ER binding FOXA1, FOXP1 and Fosl2[1,23]. Together, these data suggest a potential role of DNA hypomethylation in facilitating these new interactions.

Given the known role of ER transcription factor in inducing 3D chromatin interactions in ER+ breast cancer cells[7,24–26] and methylation-sensitive binding[27], we profiled ER binding site (ERBS) patterns genome-wide in vehicle-treated ($n = 4$) and decitabine-treated ($n = 4$) tumors using ER ChIP-seq (Supplementary Note and Supplementary Table 4) to determine whether ER binding was specifically altered by DNA hypomethylation. Differential binding analyses (Diffbind[3]) revealed reprogramming of ER binding characterized by 1,095 gained ERBS and 279 lost ERBS following decitabine treatment compared to vehicle treatment (FDR < 5%) (Fig. 4d) and a stronger average signal at gained ERBS in decitabine samples than in vehicle samples, while lost sites showed a moderate decrease in binding intensity genome-wide (Fig. 4e).

Remarkably, over 75% of all gained ERBS were located at distal regulatory regions associated with active and poised enhancers (Fig. 4f and Extended Data Fig. 6c) enriched for gained H3K27ac binding sites (Extended Data Fig. 6d), and these sites were enriched for the ERE DNA

**Fig. 3 | Loss of DNA methylation rewires 3D enhancer–promoter interactions.**
**a**, Browser snapshot of interaction landscape at the *PRRSL* gene demonstrating increased coverage of promoter-anchored interactions in PCHi-C at 1.5 kb resolution compared to Hi-C at 10 kb resolution. Bait and other end (OE) regions are marked for illustrative purposes. **b**, ChromHMM (TAMR) annotation of CHiCAGO significant interaction bait (promoter) and OE regions (putative enhancers) in decitabine-treated and vehicle samples (*$P < 0.001$, permutation test). **c**, Overlap between promoter bait and OE enhancer regions for CHiCAGO significant interactions in vehicle-treated and decitabine-treated tumors. Merged data across $n = 3$ biological replicates shown. **d**, Violin plots showing the $\log_{10}$ genomic distance of promoter interactions whose enhancer OEs are gained, maintained or lost following decitabine treatment. *$P < 0.0001$, two-sided Wilcoxon rank sum test. Merged data across $n = 3$ biological replicates shown. **e**, Average number of enhancer OE interactions per promoter bait. Error bars indicate the interquartile range. $P$ value from two-sided Wilcoxon rank sum test. **f**, Number of enhancer OE interactions per promoter bait for each CHiCAGO significant promoter-anchored interaction in vehicle-treated and decitabine-treated tumors. Merged data across $n = 3$ biological replicates shown. Data analyzed with two-tailed Pearson's correlation test. **g**, Overlap of promoter baits and enhancer OEs that are either gained or lost in decitabine with compartments that switch with decitabine (A to B or B to A).

motif, followed by FOXA1 (Fig. 4g). Lost ERBS were most frequently positioned close to transcription start sites (TSS) (>40% less than 1 kb from TSS) (Extended Data Fig. 6c), associated with active promoters (Extended Data Fig. 6e) and enriched for Sp1 and NFY promoter

DNA motifs (Extended Data Fig. 6f). Both lost and gained ERBS were highly enriched for the FOXA1 motif (Fig. 4g and Extended Data Fig. 6f). Additionally, we found a significant loss (~44.4%) of DNA methylation at gained ERBS (Fig. 4h and Extended Data Fig. 6g), as illustrated

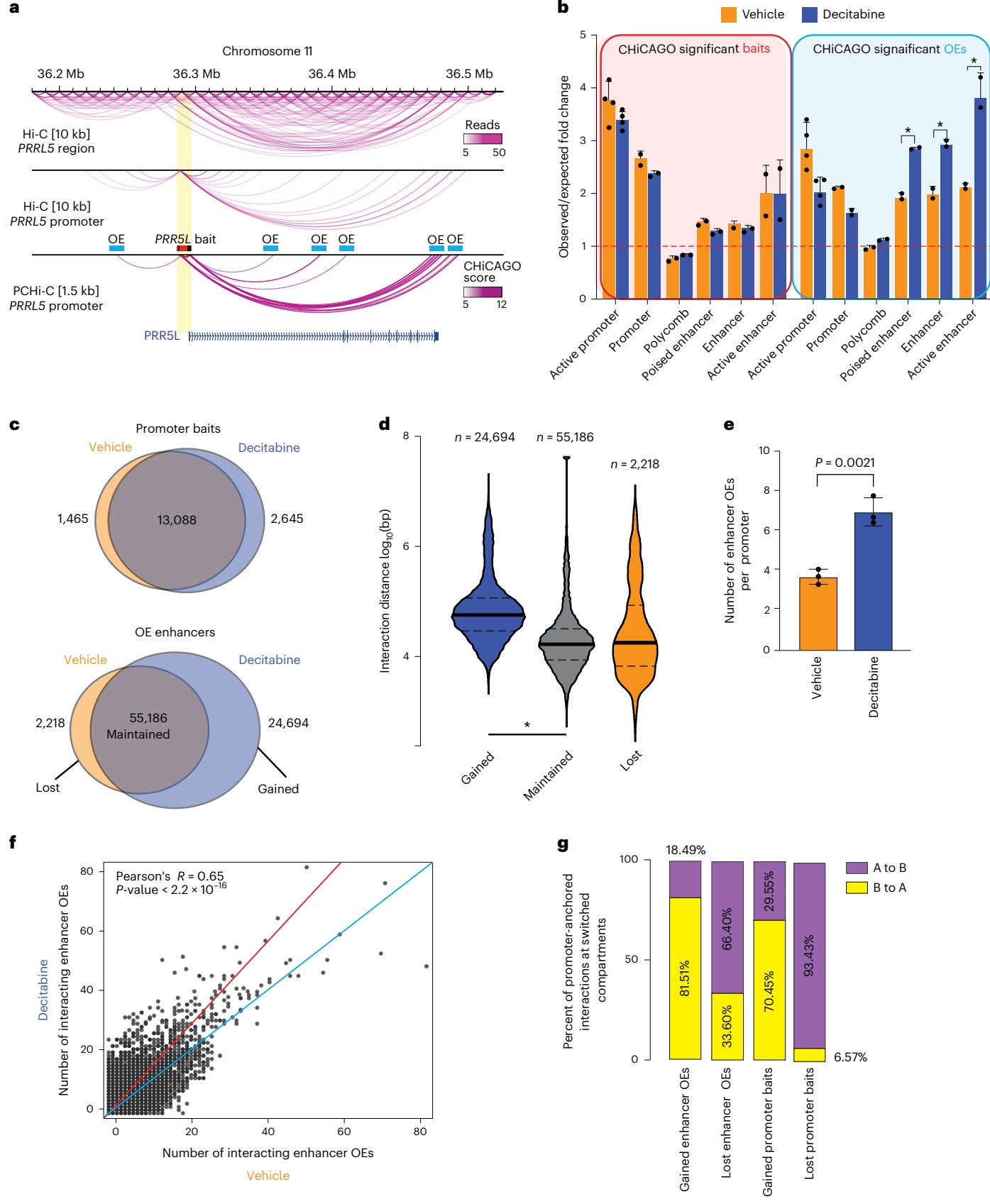

in Fig. 4i at the *ANKRD2* gene locus (Fig. 4i and Extended Data Fig. 6h). By contrast, the small proportion of ERBS that were lost remained unmethylated in both vehicle-treated and decitabine-treated treated samples (~6.92% DNA methylation change; Extended Data Fig. 6i), suggesting that this subset of ERBS were altered independently of a direct change in DNA methylation.

## Rewired ER-bound chromatin interactions at ER target genes

To determine whether this gain in ER-enhancer binding was associated with rewired 3D chromatin interactions, we integrated the gained ERBS with ectopic enhancer–promoter interactions and associated transcriptional programs. Consistent with ERE motifs enriched at gained enhancer OEs (Fig. 4c), we found significant enrichment for gained ERBS (Fig. 5a) and a genome-wide increase in ER binding density at ectopic enhancer OEs induced by decitabine treatment (Fig. 5b). We propose that these ER-associated enhancer–promoter interactions are mediated by a change in ER binding at enhancer OEs ('ER-bound interactions').

We next focused on the gained ER-bound enhancer–promoter interactions by identifying connected genes and comparing their expression between vehicle-treated and decitabine-treated tumors. The majority (~74%) of these genes showed an overall increase in expression following decitabine treatment (Fig. 5c) and included established ER target genes (for example, *B4GALT1, MYO3B, SEMA3G*) as well as genes associated with good clinical outcome in ER+ breast cancer (for example, *SPATA18, SCUBE2, GALNT5, IGFBP4*). At the *SPATA18* locus, multiple 3D enhancer–promoter interactions are gained with decitabine treatment, concomitant with gain in ER binding and gain in H3K27ac at a putative enhancer, loss of DNA methylation and 1.5-fold up-regulation of the ER target gene (Fig. 5d and Extended Data Fig. 7a). Moreover, high expression of the *SPATA18* gene is associated with good prognosis in ER+ breast cancer (Extended Data Fig. 7a). *SCUBE2* (Fig. 5e and Extended Data Fig. 7b), *B4GALT1* (Fig. 5f and Extended Data Fig. 7c) and *MYO3B* (Extended Data Fig. 7d) genes also exemplify the relationship between decitabine-induced gain of multiple ER-bound enhancer–promoter interactions and activation of their ER target genes that are associated with good prognosis in ER+ breast cancer. Together, these results reveal a link between decitabine-induced DNA hypomethylation, rewiring of ER-bound enhancer–promoter interactions and an alteration in the ER transcriptional program.

## DNA methylation dynamics and 3D chromatin interactions

Finally, to determine the dynamics between DNA methylation alterations and 3D enhancer–promoter rewiring and expression changes, we performed a time-course of decitabine followed by a period of long-term recovery in an established cell line model of endocrine-resistance TAMR[7,28,29] (tamoxifen-resistant) cells. TAMR cells were treated with a low dose of decitabine daily for 7 days to induce hypomethylation, followed by no treatment for 28 days to allow for re-methylation of CpG sites (Fig. 6a). We confirmed loss and recovery of DNMT1 protein expression by western blot (see Supplementary

Note). We assessed changes in DNA methylation (Supplementary Table 3), mRNA expression (Supplementary Table 6) and 3D enhancer–promoter interactions (PCHi-C; Supplementary Table 9) on cells at day 7 of decitabine treatment ('day-7 decitabine') and day 28 post decitabine treatment ('decitabine recovery') as well as passage-matched control cells ('control early' and 'control late') in duplicate. As expected, day-7 decitabine treatment resulted in widespread DNA hypomethylation in the TAMR cells (~41.84% change in median DNA methylation, two-tailed Mann–Whitney test, $P < 0.0001$; Fig. 6b and Extended Data Fig. 8a). Substantial genome-wide recovery of DNA methylation was observed following 28 days of recovery compared to matched vehicle-treated control (~25.46% change in median DNA methylation, two-tailed Mann–Whitney test, $P < 0.0001$; Fig. 6b and Extended Data Fig. 8a). Similar to the PDX decitabine-treated samples, we found that DNA hypomethylation changes in TAMR cells after 7 days of decitabine treatment were enriched for ChromHMM[7] enhancers (Fig. 6c) and ERBS[3] (Fig. 6d).

To study the dynamics of DNA re-methylation on 3D chromatin interactions, we first identified DNA regions that were substantially re-methylated in decitabine recovery samples compared to day-7 decitabine samples (>30% gain in DNA methylation; Supplementary Table 3). We found that regions that were re-methylated were enriched in poised enhancers but depleted in promoter regions (Fig. 6e). In fact, 10,195 probes located at ChromHMM enhancers that were hypomethylated after day-7 decitabine treatment gained methylation after 28 days of recovery (Extended Data Fig. 8b) and were also enriched for ER binding[3] (Fig. 6d). To determine whether 3D enhancer–promoter interactions were indeed altered in the decitabine time-course, we performed PCHi-C in duplicate (Supplementary Table 9). We found that chromatin interactions separate the control and decitabine-treated samples on the x axis, with decitabine recovery samples clustering together with control samples and away from the day-7 decitabine samples on the y axis, suggesting substantial recovery of chromatin interactions 28 days post decitabine treatment (Extended Data Fig. 8c). Similar to the PDX results (Fig. 3f), we found that day-7 decitabine-treated TAMR samples resulted in a large gain in the number of new enhancer OEs connected to bait promoters (Fig. 6f). Moreover, these additional ectopic interactions were mostly lost after decitabine recovery (Fig. 6g). Based on these results, we defined two classes of gained interactions following decitabine treatment: 'gained and maintained' interactions and 'gained and lost' interactions (Fig. 6h). Importantly, ~73.4% of day-7 decitabine gained OE enhancer (64,044) interactions were lost in decitabine recovery samples (47,007 OE enhancers gained and lost) (Fig. 6i), whereas gained and maintained interactions showed decreasing ChICAGO significance scores, suggesting some reduction in interaction strength after 28 days of DNA methylation recovery (Extended Data Fig. 8d).

We further found that gained enhancer–promoter interactions at day 7 of decitabine treatment were significantly associated with an overall increase in gene expression (195 upregulated genes; $P < 0.05$; log(fold change) > 1.5; Fisher's exact test $P < 2.2 \times 10^{-16}$) (Fig. 7a). No significant increase in expression for genes involved in gained interaction

---

**Fig. 4 | Rewired 3D chromatin interactions align with altered transcription.**
**a**, Normalized enrichment scores (NES) for signature gene sets representing differentially expressed genes in RNA-seq data from Gar15-13 PDX tumors treated with decitabine compared to vehicle ($n = 4$ biological replicates; FDR < 0.05). **b**, Decitabine versus vehicle differential expression of genes that are located at enhancer–promoter interactions gained with decitabine treatment. Data analyzed with two-sided Fisher's exact test. **c**, Transcription factor motifs significantly enriched at promoter-interacting enhancers (enhancer OEs) gained with decitabine treatment. Only motifs with binomial $P < 0.05$ are shown. **d**, Overlap of consensus ER peaks in vehicle-treated and decitabine-treated Gar15-13 PDX tumors ($n = 4$ biological replicates each). Heatmaps indicate ER ChIP-seq signal intensity at ERBS gained and lost in decitabine-treated compared to vehicle-treated tumors. **e**, Average signal intensity of ER ChIP-seq binding

(Gar15-13 vehicle-treated and decitabine-treated tumors) at gained and lost ERBS with decitabine treatment. **f**, ChromHMM (TAMR) annotation (*$P < 0.001$, permutation test) of ERBS gained with decitabine treatment compared to matched random regions across the genome. Size of the overlap is presented in the respective column. **g**, Transcription factor motifs enriched at ERBS gained with decitabine treatment compared to matched random regions generated from ERE binding motifs across the genome. **h**, DNA methylation levels (β-values) at gained ERBS in decitabine-treated and vehicle-treated PDX tumors ($n = 4$ biological replicates each). **i**, Browser snapshot of ER ChIP-seq together with EPIC DNA methylation (vehicle and decitabine treatments, $n = 4$ biological replicates each) showing gain of ER binding and loss of DNA methylation at an enhancer region of ER target gene *ANKRD2*.

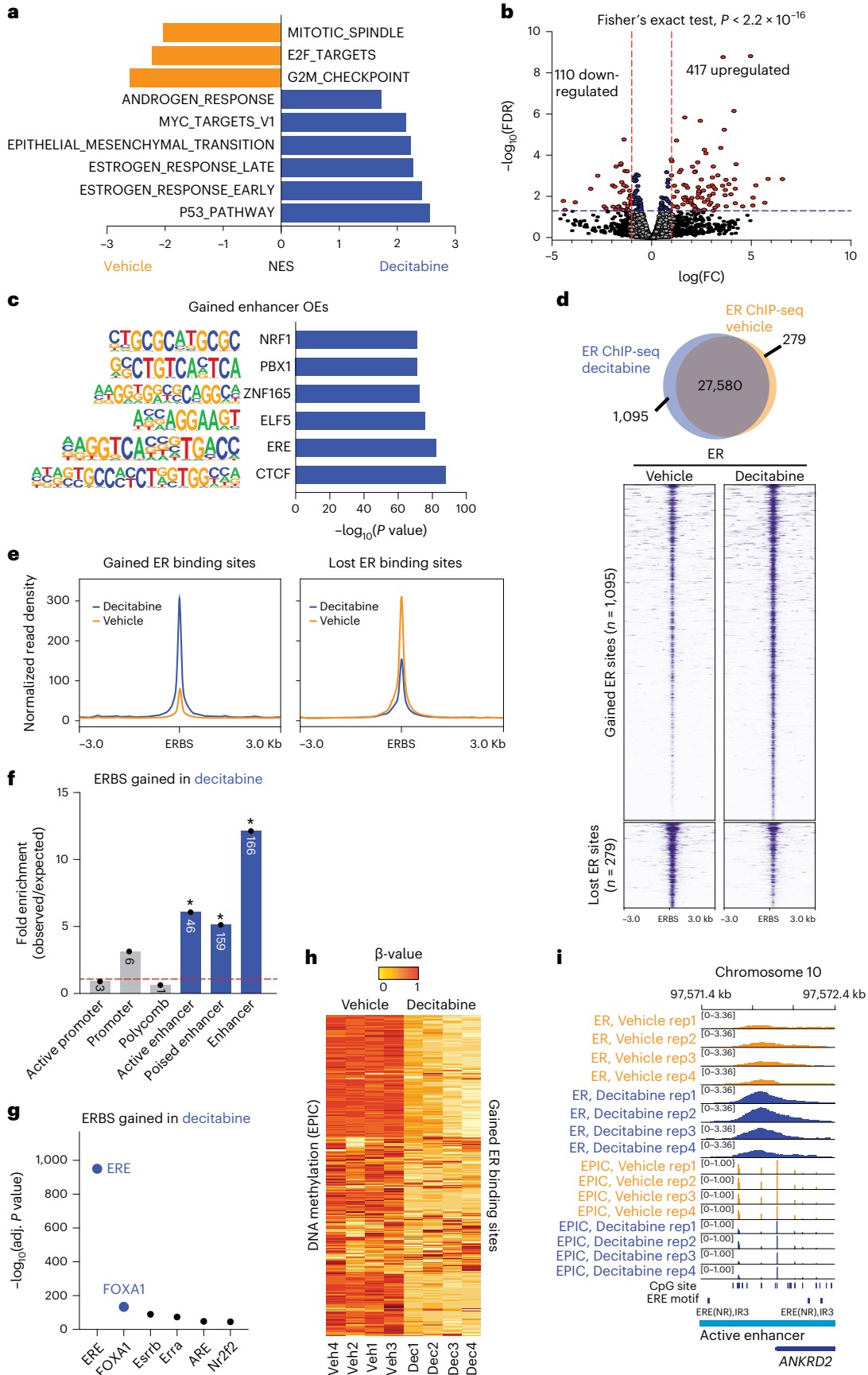

was observed in decitabine recovery versus control-late samples (49 upregulated genes; Fisher's exact test, $P = 0.3167$) (Fig. 7b). After 28 days of recovery, we observed that loss of gene expression was concordant with a reversal of ectopic enhancer–promoter interactions, including at key ER target genes (Fig. 7b). Further evidence of a direct relationship

between DNA hypomethylation and ectopic 3D enhancer–promoter interactions is exemplified at ER target genes also identified in the PDX data: *SPATA18* (Fig. 7c), *B4GALT* (Fig. 7d), *EVL* and *MYO3B* (Extended Data Fig. 9a,b). Notably, we also found a subset of genes that remained upregulated after 28 days of DNA methylation recovery, in which the

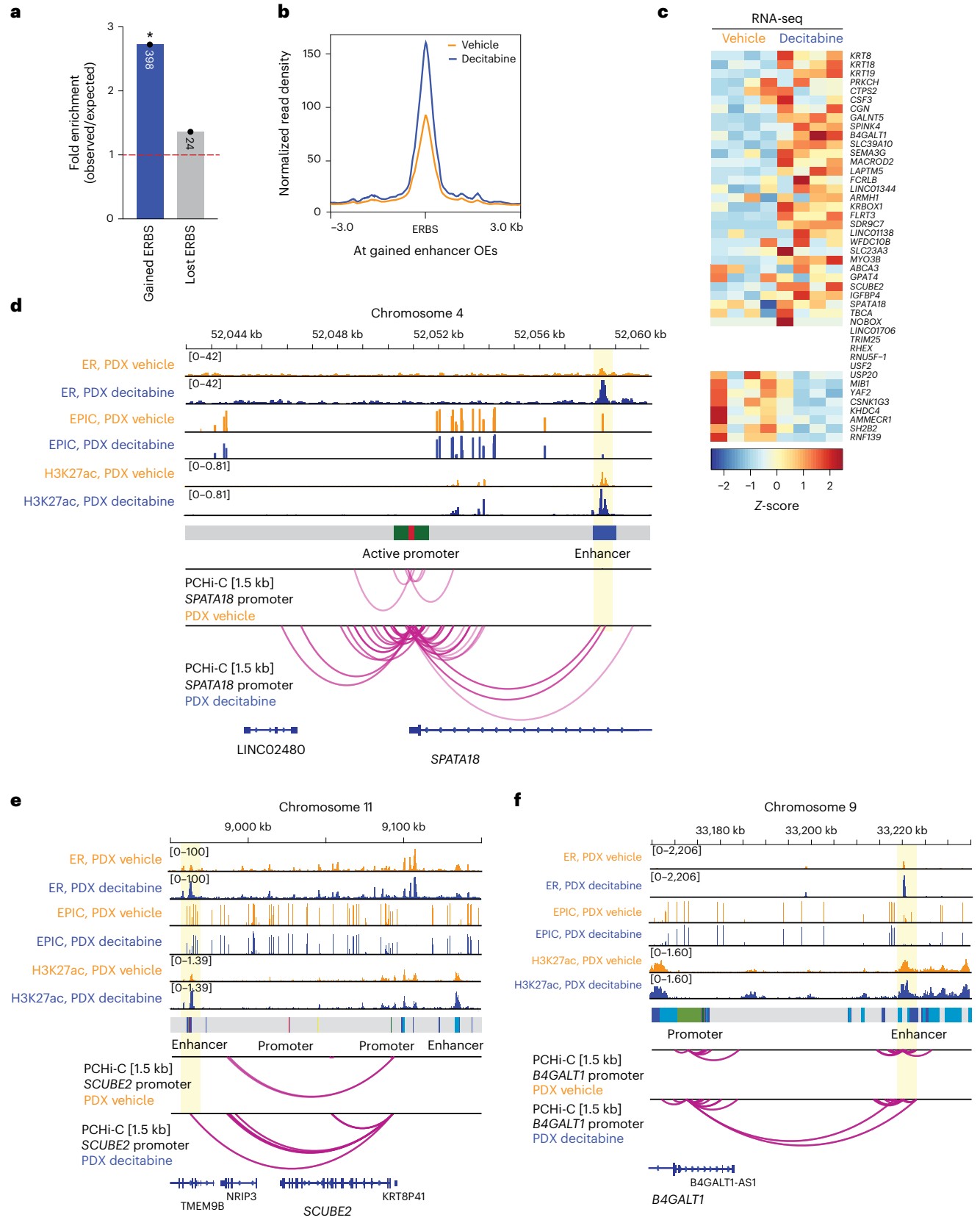

**Fig. 5 | Rewired ER-bound interactions are associated with activation of ER target genes. a**, O/E fold change enrichment of gained enhancer OEs for ER binding gained and lost following decitabine treatment (*$P$ < 0.0001, permutation test). **b**, Average ER ChIP-seq signal intensity (Gar15-13 vehicle-treated and decitabine-treated tumors) at ERBS located at DNA hypomethylation-induced enhancer OEs. **c**, Expression of genes connected to gained ER-mediated enhancer OEs in vehicle-treated and decitabine-treated tumors. **d**, Browser snapshots showing the promoter-anchored interactions at the *SPATA18* gene, together with the average ER ChIP-seq signal, EPIC DNA methylation, H3K27ac CUT&RUN signal, ChromHMM track and PCHi-C interaction track. Merged replicate data are shown ($n$ = 4 biological replicates each; $n$ = 3 biological replicates each for CUT&RUN and PCHi-C). In decitabine-treated tumors, the *SPATA18* promoter displays an increased number of interactions with an upstream enhancer region, which gains ER and H3K27ac binding with decitabine treatment, concomitant with loss of DNA methylation. Expression of the *SPATA18* gene was upregulated in decitabine-treated tumors (shown in Extended Data Fig. 7a). **e**, Browser snapshots showing promoter-

anchored interactions at the *SCUBE2* ER target gene, together with ER ChIP-seq, EPIC DNA methylation, H3K27ac CUT&RUN signal, ChromHMM track and PCHi-C interaction track. Merged replicate data are shown ($n$ = 4 biological replicates each; $n$ = 3 biological replicates each for CUT&RUN and PCHi-C). In decitabine-treated tumors, the *SCUBE2* promoter displays additional interactions with a distal enhancer, which gains ER and H3K27ac binding with decitabine treatment. Expression of the *SCUBE2* gene was significantly upregulated in decitabine-treated tumors (shown in Extended Data Fig. 7b). **f**, Browser snapshots showing promoter-anchored interactions at the *B4GALT1* ER target gene, together with ER ChIP-seq, EPIC DNA methylation, H3K27ac CUT&RUN signal, ChromHMM track and PCHi-C interaction track. Merged replicate data are shown ($n$ = 4 biological replicates each, $n$ = 3 biological replicates for CUT&RUN and PCHi-C). In decitabine-treated tumors, the *B4GALT1* promoter displays additional long-range interactions with a distal enhancer, which gains ER and H3K27ac binding with decitabine treatment. Expression of the *B4GALT1* gene was significantly upregulated in decitabine-treated tumors (shown in Extended Data Fig. 7c).

chromatin contacts were still partially or fully maintained (gained and maintained interactions) (Extended Data Fig. 9c).

## Discussion

Three-dimensional epigenome remodeling, including widespread changes to DNA methylation and 3D chromatin structure, is an emerging mechanism of gene deregulation in cancer. Our previous work demonstrated that DNA hypermethylation and concomitant loss of ER binding at enhancers was associated with alterations in 3D chromatin interactions in ER+ endocrine-resistant breast cancer. Therefore, we were motivated to determine whether these 3D chromatin alterations could be resolved with epigenetic therapies that induce DNA hypomethylation.

Here, using PDX models of ER+ endocrine-resistant breast cancer, we show that treatment with decitabine induced DNA hypomethylation and had potent anti-tumor activity associated with suppression of tumor growth and cell proliferation gene pathways. Given that long-term drug treatment can result in the selection of intrinsically resistant colonies[30,31], we first inferred genetic (copy number variations and single nucleotide variants) and epigenetic heterogeneity to assess the impact of low-dose decitabine on genetic and epigenetic clonal evolution. We found that PDX tumors retained their high degree of clonal heterogeneity following decitabine treatment in both PDX models. To further assess the broader functional impact of DNA hypomethylation, we analyzed multiple layers of 3D genome organization, including chromosomal compartments, TADs and 3D chromatin interactions, and integrated the 3D data with DNA methylation, transcriptome, and ER, CTCF transcription factor and H3K27ac histone modification binding profiles in decitabine-treated and vehicle-treated PDX tumors. Collectively, our data support a model whereby low-dose decitabine treatment results in DNA hypomethylation, leading to ectopic enhancer activation, reprogramming of ER chromatin binding and rewiring of enhancer–promoter

interactions that, together, results in activation of ER target genes. Importantly, we identified rewired ER-bound chromatin interactions that connect ER-enhancers to specific target genes, which included estrogen response hallmark genes involved in cell cycle inhibition and tumor suppression, consistent with reduced tumor growth observed in the PDX models. Finally, we confirm a mechanistic link between decitabine-induced DNA hypomethylation, rewiring of 3D chromatin interactions and gene activation using 'recovery' DNA methylation experiments in a cell line model of endocrine-resistant breast cancer.

Decitabine has been previously demonstrated to have some therapeutic efficacy in multiple subtypes of breast cancer and in overcoming drug resistance[32]. Transient low-dose treatment with decitabine resulted in a decrease in promoter DNA methylation and gene re-expression, and had an anti-tumor effect on in vivo in breast cancer cells[33]. Low-dose decitabine has also been shown to prevent cancer recurrence by disrupting the pre-metastatic environment in breast and other cancers[34]. In triple-negative breast cancer PDX organoids, decitabine sensitivity was positively correlated with protein levels of DNMTs[35]. A recent study of decitabine in a panel of breast cancer cell lines found that decitabine also induced genes within apoptosis, cell cycle, stress and immune pathways[36]. However, knockdown of key effectors of the immune pathway did not affect decitabine sensitivity, suggesting that breast cancer growth suppression by decitabine is independent of viral mimicry[36].

We found that the low-dose decitabine treatment resulted in minimal DNA hypomethylation at repetitive elements. Despite this finding, we observed a relatively large number of transposable elements becoming activated with treatment, consistent with previous studies[37]. Loss of DNA methylation at repetitive elements and expression of transposable elements has been shown to drive viral mimicry response in tumors treated with epigenetic therapies[11,12]. In agreement, our results indicate that treatment with decitabine results in up-regulation of multiple

**Fig. 6 | Dynamics between DNA methylation and 3D chromatin interactions. a**, Experimental design for the TAMR cell line study. Created with Biorender.com. **b**, Distribution of DNA methylation for control early, control late, day-7 decitabine and decitabine recovery ($n$ = 2 technical replicates each) TAMRs for all EPIC probes. Black line indicates median ± s.d. Box plots show median, interquartile range and maximum–minimum DNA methylation. **c**, O/E fold change enrichment of DMRs in day-7 decitabine TAMRs compared to control early across TAMR ChromHMM regulatory regions (*$P$ < 0.001, permutation test). The numbers located within each specific region are presented in the respective column. **d**, O/E fold change enrichment of day-7 decitabine hypomethylated DMRs (compared to control early) and decitabine recovery re-methylated DMRs (compared to day-7 decitabine) for ER binding in TAMRs[3] (*$P$ < 0.0001, permutation test). **e**, O/E fold change enrichment of EPIC DMRs that become re-methylated in decitabine recovery TAMRs compared to day-7 decitabine cells across TAMR ChromHMM regulatory regions (*$P$ < 0.001, permutation test).

The numbers located within each specific region are presented in the respective column. **f**, Number of enhancer OEs per promoter bait for each promoter-anchored interaction in day-7 decitabine and control early TAMRs. Merged data across replicates shown. Data were analyzed by two-tailed Pearson's correlation test. **g**, Number of enhancer OEs per promoter bait for each promoter-anchored interaction in decitabine recovery and control late TAMRs. Merged data across replicates shown. Data were analyzed by two-tailed Pearson's correlation test. **h**, Schematic representation of two identified classes (gained and lost; gained and maintained) of gained chromatin interactions in TAMRs. **i**, Overlap of enhancer OEs between day-7 decitabine and control early (left panel) and decitabine recovery and control late TAMRs (right panel). Bottom diagram shows overlap between gained interactions in day-7 decitabine versus control early and in decitabine recovery versus control late, demonstrating the number of gained and lost versus gained and maintained interactions. Merged data across replicates shown.

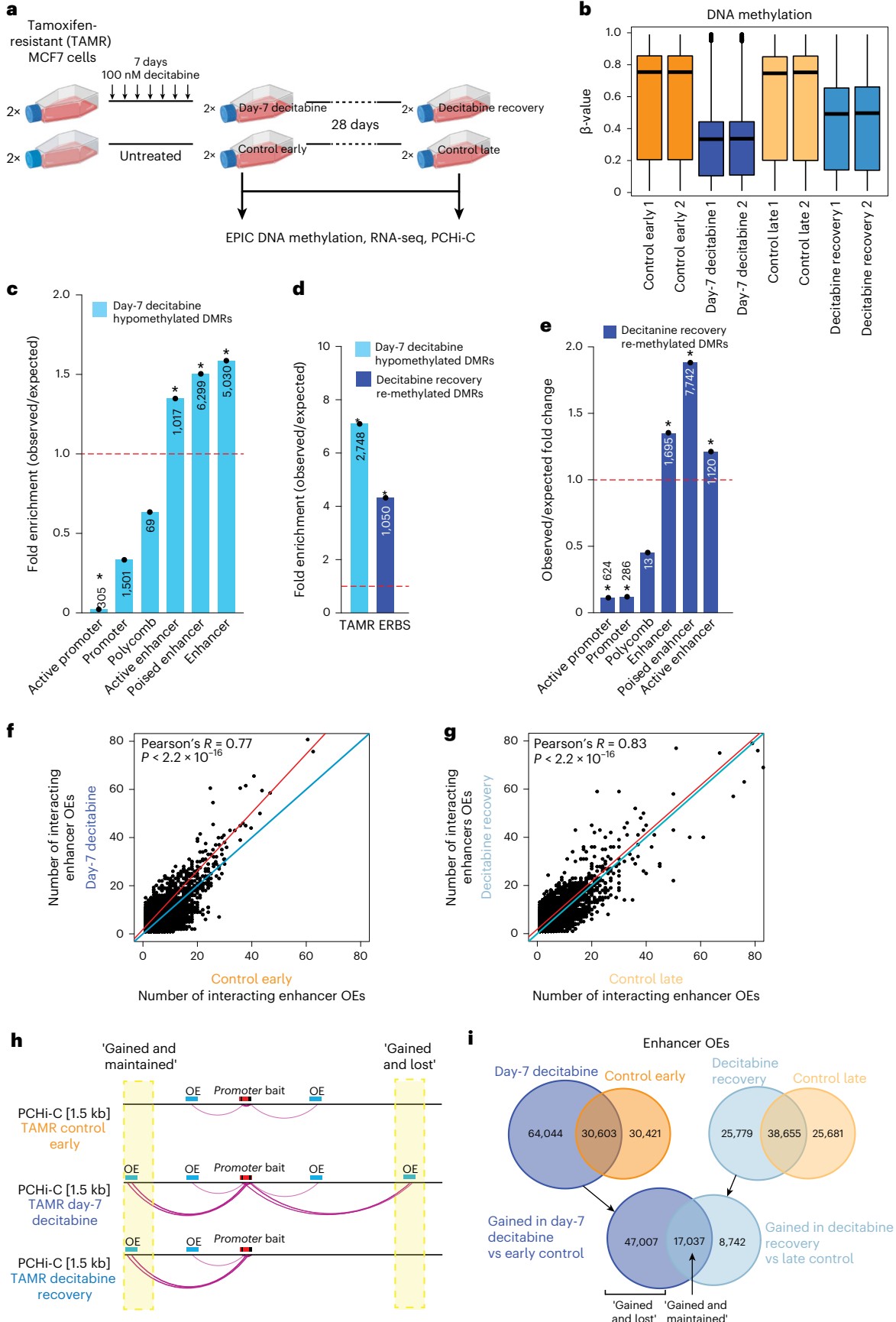

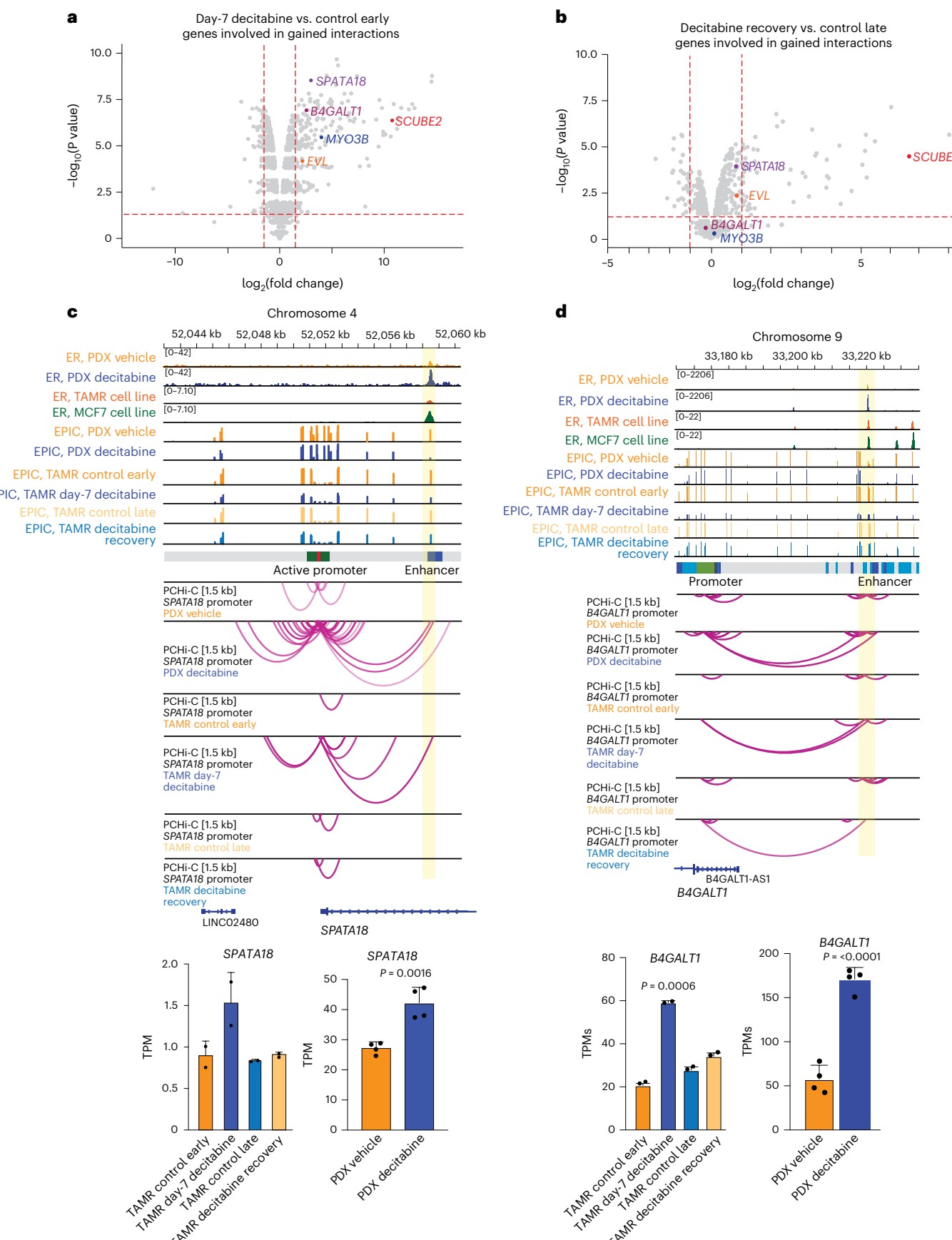

**Fig. 7 | Dynamics of altered ER-bound 3D chromatin interactions on gene transcription. a**, Differential expression of genes involved in gained interactions in day-7 decitabine-treated TAMRs. Genes included in representative examples are labeled. **b**, Differential expression of genes involved in gained interactions in decitabine recovery TAMRs. Genes included in representative examples are labeled. **c**, Browser snapshots showing promoter-anchored interactions at the *SPATA18* ER target gene. Gar15-13 PDX vehicle-treated and decitabine-treated data tracks are overlayed with ER ChIP-seq for TAMR and MCF7 cell lines[3], EPIC methylation for TAMRs, ChromHMM track and PCHi-C for TAMR cell line data. Merged replicate data are shown (*n* = 4 biological replicates each for Gar15-13 and *n* = 2 technical replicates for TAMRs). In decitabine-treated PDXs and TAMRs (day-7 decitabine), the *SPATA18* promoter displays an increased number of interactions with an upstream enhancer region, which gains ER binding with decitabine-treatment in PDXs, concomitant with loss of DNA methylation in both PDXs and TAMRs. These ectopic chromatin interactions are lost after 28 days of decitabine recovery with partial recovery of DNA methylation at that locus. *SPATA18* gene expression was significantly upregulated in decitabine-treated

versus vehicle-treated PDXs (bottom right, RNA-seq transcripts per million, TPM) and in day-7 decitabine-treated TAMRs and suppressed in decitabine recovery TAMRs (bottom left, RNA-seq TPM). **d**, Browser snapshots showing promoter-anchored interactions at the *B4GALT1* ER target gene. Gar15-13 PDX vehicle-treated and decitabine-treated data tracks are overlayed with ER ChIP-seq for TAMR and MCF7 cell lines[3], EPIC methylation for TAMRs, ChromHMM track and PCHi-C for TAMR cell line data. Merged replicate data are shown (*n* = 4 biological replicates each for Gar15-13 and *n* = 2 technical replicates for TAMRs). In decitabine-treated PDXs and TAMRs (day-7 decitabine), the *B4GALT1* promoter displays an increased number of long-range interactions with a distal enhancer region, which gains ER binding with decitabine treatment in PDXs, concomitant with loss of DNA methylation in both PDXs and TAMRs. These ectopic chromatin interactions are partially reversed after 28 days of recovery with recovery of DNA methylation at that enhancer locus. *B4GALT1* expression increased decitabine-treated versus vehicle-treated PDXs (bottom right, RNA-seq TPM) and in day-7 decitabine-treated TAMRs compared to control early samples and was restored in decitabine recovery TAMRs (bottom left, RNA-seq TPM).

---

immune pathways, which could promote anti-tumor immunity. However, the immunodeficient NOD-*scid IL2ry*^*null* mice required for the PDX experiments in our study largely lack mature immune cells, and therefore the potential immune response could not solely account for the tumor inhibitory effects of decitabine treatment observed in our study. This highlights the need to study both the immune-based and tumor-based mechanisms that underpin response to epigenetic therapies.

There have been limited studies to date on the role of DNA methylation in shaping the 3D genome organization[38,39]. Simultaneous profiling of DNA methylation and 3D genome in single cells revealed pervasive interactions between these two epigenetic layers in regulating gene expression[40]. Furthermore, previous studies have linked DNA hypomethylation with de-compaction of chromatin and loss of compartmental organization[41–43]. Our data also showed that DNA hypomethylation specifically results in closed (B-type) to active (A-type) compartment shifting and reduced interactions between B-type compartments. Additionally, the binding of CCCTC-binding factor (CTCF), an insulator protein involved in creating chromatin loops and domain boundaries, has been shown to be methylation-sensitive at a small number of sites[44–47]. Notably, in our study, we show that the TAD structure is disrupted after decitabine treatment, without a significant change in CTCF binding at altered TAD boundaries. Decitabine-induced DNA hypomethylation resulted in only a low number of altered CTCF binding sites. Our findings are complementary to previous studies showing that the vast majority of unoccupied, methylated CTCF motifs remain unbound upon loss of DNA methylation[46,48].

Although DNA methylation may play a role in altering ER binding at regulatory elements[29,49], no studies have examined the potential effect on ER-bound 3D chromatin interactions. We previously suggested that DNA methylation differences at enhancers underpin differential ER binding events associated with endocrine resistance[29]. Furthermore, ER-bound 3D chromatin interactions have been reported to be altered in endocrine-resistant cells[6,7]. We now show that decitabine-induced DNA hypomethylation also results in a gain of H3K27ac at ER-bound enhancers, suggesting an important functional role in promoting ectopic 3D chromatin interactions. Our high-resolution promoter interaction data also revealed an increase in the number of interacting enhancers connecting to gene promoters induced by decitabine. We speculate that the overall increase in enhancer connectivity results in the creation of active transcription hubs[21] or frequently interacting regions[50] at activated genes. This is in agreement with recent reports of transcriptional activation occurring in non-membrane-bound nuclear compartments that harbor multi-way enhancer–promoter interacting hubs[51] as well as additive effects of multi-way enhancer interactions on gene expression[52].

In summary, our work highlights a novel molecular mechanism of epigenetic therapy in endocrine-resistant ER+ breast cancer. We provide mechanistic insights into how decitabine-induced DNA

hypomethylation promotes 3D epigenome remodeling, including rewiring of ER-mediated 3D enhancer–promoter chromatin interactions. Epigenetic therapy, therefore, has the potential to overcome cancer therapy resistance by targeting the 3D epigenome architecture to resolve gene deregulation and reduce cancer growth.

## Online content

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

## Methods

All animal experiments presented in this study were conducted according to regulatory standards approved by the Garvan Institute of Medical Research, St. Vincent's Hospital Animal Ethics Committee.

### PDX models of ER+ breast cancer

All in vivo experiments, procedures and endpoints were approved by the Garvan Institute of Medical Research Animal Ethics Committee (HREC nos. 14/35 and 15/25, ARA no. 21/11) and were performed at the Garvan Institute of Medical Research using standard techniques as described previously[53] in accordance with relevant national and international guidelines. The Gar15-13 model was generated in-house at St Vincent's Hospital under the Human Research Ethics protocol (HREC/16/SVH/29) and the HCI-005 model was developed by the Welm laboratory at the Huntsman Cancer Institute (University of Utah)[15]. Gar15-13 was derived from a resected breast cancer liver metastasis of ER+, progesterone negative (PR−), HER2-negative (HER2−) metastatic breast cancer[13]. HCI-005 was derived from a pleural effusion of ER+ (ERmut$^{L536P}$), PR+, HER2+ metastatic breast cancer. Growth of HCI-005 was supported by estrogen supplementation in the form of a 60-day 17-β-estradiol pellet implanted simultaneously with the tumor chunks. Mice implanted with Gar15-13 did not receive estrogen supplementation, as this model does not require additional estrogen for growth[13].

During surgery, 4 mm$^3$ sections of tumor tissue were implanted into the fourth inguinal mammary gland of 6–8-week-old female NOD-*scid* IL2Ry$^{null}$ mice, obtained from Australian BioResources (Sydney, Australia). Mice were socially housed at the Garvan Institute of Medical Research specific pathogen-free animal facility, in temperature-controlled and light-cycle-controlled rooms, and were given ad libitum access to food, water and nesting materials. For HCI-005, tumor growth was supported by the implantation of an E2 pellet inserted subcutaneously through the incision site before it was sealed with an Autoclip wound clip. When tumors became palpable, tumor growth was assessed twice weekly by caliper measurement (using the formula: width$^2$ × length / 2). Once tumors reached 200 mm$^3$, mice were randomized to treatment arms using an online randomization tool (https://www.graphpad.com/quick-calcs/randomize1.cfm) (n = 6–8 mice per group for therapeutic studies; exact numbers specified in figure legends).

### Pharmacological treatments in PDX models

The DNA methyltransferase inhibitor decitabine (5-Aza-2′-deoxycytidine; Sigma, cat. no. 3656) was reconstituted in PBS and stored at −80°C. Decitabine was administrated intraperitoneally (0.5 mg kg$^{-1}$ per mouse in 100 µl PBS), two times weekly. Vehicle mice were treated with 100 µl PBS intraperitoneally. Mice were treated for 60 days or until tumor volume reached 1,000 mm$^3$. Upon reaching the ethical or pre-defined experimental endpoint, mice were killed and the primary tumor was collected. After weighing, the tumor was cut into pieces that were allocated to be snap-frozen, fixed overnight at 4 °C in 10% neutral-buffered formalin or embedded in cryo-protective optimal cutting temperature compound before being snap-frozen. Frozen samples were kept at −80°C. Formalin-fixed samples were sent to the Garvan Institute Histology Core Facility for paraffin embedding. Tumor growth curves were analyzed in GraphPad Prism (GraphPad Software) by two-tailed, unpaired *t*-test. Tumor mass at endpoint was analyzed by two-tailed Mann–Whitney *t*-test as per figure legends unless otherwise specified.

### Cell culture

MCF7 breast cancer cells and the corresponding endocrine-resistant sub-cell lines were kindly given to our laboratory by J. Gee (Cardiff University, UK). Tamoxifen-resistant MCF7 cells (TAMR[28]) were previously generated by the long-term culture of MCF7 cells in phenol red-free RPMI medium containing 10% charcoal-stripped FBS (Gibco) and 4-OH-tamoxifen ($1 × 10^{-7}$ M; TAM). All cell lines were authenticated by short-tandem repeat profiling (Cell Bank, Australia) and cultured for <6 months after authentication.

### Pharmacological treatments in cell lines

Cells were treated daily with decitabine (100 nM) for seven consecutive days. After 7 days, fresh media was added, and cells were collected on day 7 (day-7 decitabine). Control cells were cultured for a total of 11 days in normal media and collected as 'control early' on day 11. For the decitabine recovery samples, cells were treated daily with decitabine (100 nM) for seven consecutive days, after which fresh media was added; cells were cultured for 21 additional days and collected on day 28 ('recovery'; reintroduction of DNA methylation). Matched control cells were cultured for 28 days in normal media and collected at day 28 as 'control late'. DNMT1 protein levels were confirmed by western blot (see Supplementary Note).

### Immunohistochemistry and quantification

Tumor tissue was collected and immediately fixed in 10% neutral-buffered formalin at 4 °C overnight before dehydration and paraffin embedding. Antibodies used for immunohistochemistry were anti-ER (M7047, 1:300, Agilent) and anti-Ki-67 (M7240, 1:400, Agilent). Primary antibodies were detected using biotinylated IgG secondary antibodies (Agilent, 1:400), using streptavidin-HRP (Agilent) for amplification of signal followed by the addition of 3,3′-diaminobenzidine (Sigma) substrate. Images were scanned using Leica Aperio Slide Scanner (Leica Biosystems) and analyzed using QuPath software to differentiate tumor tissue from stroma and necrosis, and to quantify Ki-67 positivity in tumor tissue.

### Low input in situ Hi-C in snap-frozen PDX tumor samples and TAMR cells

Tumor tissue samples were flash-frozen and pulverized in liquid nitrogen before formaldehyde cross-linking in TC buffer. Hi-C was then conducted using the Arima-HiC kit according to the manufacturer's protocols (cat. no. A510008), with minor modifications. In brief, for each Hi-C reaction, between ~100,000 and 500,000 cells were cross-linked with 2% formaldehyde and nuclei were isolated by incubating cross-linked cells in Lysis Buffer at 4°C for 30 min. The Arima kit uses two restriction enzymes recognizing the following sequence motifs: ^GATC and G^ANTC (N can be either of the four genomic bases), which, after ligation of DNA ends, generates four possible ligation junctions in the chimeric reads: GATC–GATC, GANT–GATC, GANT–ANTC and GATC–ANTC. Hi-C libraries were prepared using the Swift Biosciences Accel-NGS 2S Plus DNA Library Kit with a modified protocol provided by Arima, with eight PCR cycles for library amplification as required. Hi-C libraries were sequenced on Illumina HiSeq X10 in 150 bp paired-end mode.

### Promoter Capture Hi-C

To perform PCHi-C, we computationally designed RNA probes that capture promoter regions of previously annotated human protein-coding genes. Promoter capture was performed as previously described[54] using the Arima HiC+ kit for Promoter CHi-C (human) (cat. nos. A510008, A303010, A302010 and A301010). First, to identify promoter capture targets, 23,711 unique Ensembl annotated genes were extracted from the GRCh38 gene annotation file in the Ensembl database (v.95). These comprised protein-coding (18,741), antisense (84), lincRNA (170), miRNA (1,878), snoRNA (938), snRNA (1,898) or multiple (2) transcripts. We then located the TSS of each gene, mapped the TSS coordinates to the in silico digested genome (^GATC and G^ANTC) and extracted the restriction fragment containing the TSS as well as one restriction fragment upstream and one restriction fragment downstream for each TSS. The final target list of TSS mapped to three consecutive restriction fragments. The average length of the three consecutive restriction fragments for each TSS is 786 bp and the median is 927 bp, with a range of 54–4174 bp.

Moreover, for the individual restriction fragments smaller than 700 bp, all nucleotides within these fragments are less than or equal to

to 350 bp from the nearest cut site, and therefore the entire restriction fragment was defined as a target region for subsequent probe design. This scenario represents the vast majority of cases because the mean length of an individual restriction fragment is 263 bp, with a median of 431 bp. However, if an individual restriction fragment was greater than 700 bp, then the 350 bp on each inward-facing edge of the restriction fragment was defined as a target region for probe design, and the center-most portion of the restriction fragment was excluded from the probe design. After this final processing, a final BED file of target bait regions was input into the Agilent SureDesign tool, and probes were designed using a 1× tiling approach, with moderate repeat masking and balanced boosting. Promoter capture was carried out using Hi-C libraries from three vehicle-treated tumor samples and three decitabine-treated tumor samples with the SureSelect target enrichment system and RNA bait library according to the manufacturer's instructions (Agilent Technologies kit), using 12 post-capture PCR cycles as required. PCHi-C libraries were sequenced on the Illumina HiSeq X10 platform in 150 bp paired-end mode.

## Microarray genome-wide DNA methylation

DNA from four decitabine-treated and four vehicle-treated tumors from two PDX models (Gar15-13 and HCI-005) was isolated from snap-frozen tumor samples using the Qiagen QIAamp DNA Mini Kit. DNA (500 ng) was treated with sodium bisulfite using the EZ-96 DNA methylation kit (Zymo Research CA, USA). DNA methylation was quantified using the Illumina Infinium MethylationEPIC (EPIC) BeadChip (Illumina, CA, USA), run on the HiScan System (Illumina, CA, USA) following the manufacturer's standard protocol.

## ChIP-seq

Tumor samples were snap-frozen in Optimal Cutting Temperature compound (Tissue-Tek) and used for ER ChIP-seq experiments. Using a cryostat (Leica, CM3050-S), a minimum of 50 sections (30 μm each) were cut from each tumor at −20 °C and subjected to double cross-linking with DSG and formaldehyde as previously described[14]. ER ChIP-seq was performed with an anti-ER antibody (Santa Cruz, SC-543X). A total of 5 μg of antibody was used to ChIP each tumor sample and 10 ng of immunoprecipitated DNA was submitted to the David R. Gunn Genomics Facility at the South Australian Health and Medical Research Institute (SAHMRI) for sequencing. Conversion of the DNA into sequencing libraries was performed using the Ultralow Input Library Kit (Qiagen, cat. no. 180495) and sequenced on the Illumina NextSeq 500 (Illumina) in 75 bp single-end mode to achieve a minimum of 20 million reads per sample.

## CUT&RUN

CUT&RUN was performed using the CUTANA CUT&RUN Protocol (www.epicypher.com), which is an optimized version of a previously described protocol[16]. For each sample, 1–2 mg of tumor chunk was finely minced on ice with a clean scalpel, followed by light cross-linking (0.1% formaldehyde for 2 min). Cross-linking was stopped by adding 2.5 M glycine to a final concentration of 125 mM, and tissue pieces were processed into a single-cell suspension by douncing, followed by nuclei isolation as per CUT&RUN protocol. Antibodies used were CUT&RUN–Epicypher validated antibodies CTCF (CTCF CUTANA CUT&RUN Antibody (cat. no. 13-2014); 0.5 μg per reaction) and H3K27ac (Histone H3K27ac Antibody, SNAP-ChIP Certified (cat. no. 13-0045); 0.5 μg per reaction). CUT&RUN-enriched DNA was purified and ~5 ng was used to prepare sequencing libraries with the CUTANA CUT&RUN Library preparation kit. Libraries were sequenced with the Illumina NextSeq 500 system (2 x 75 bp).

## RNA-seq

RNA was extracted from snap-frozen tumor PDX tissue and TAMR cell line samples using the RNeasy Mini Kit (Qaigen), and the quality of purified RNA was confirmed with RNA ScreenTape TapeStation (Agilent).

All samples processed for RNA-seq had a RIN equivalent quality score of ≥8.0. Total RNA was supplied to the genomics core facility (Kinghorn Centre for Clinical Genomics) for library preparation and sequencing. RNA was prepared for sequencing using the TruSeq Stranded mRNA Library Prep kit (Illumina), and libraries were sequenced on an Illumina NovaSeq 6000 S4 in paired-end mode.

## EPIC DNA methylation analyses

Raw intensity data files were imported and quality controlled using the *minfi* package (v.1.34.0)[55]. Data were then normalized with background correction. To reduce the risk of false discoveries, we removed probes affected for cross-hybridization to multiple locations on the genome or that overlapped single-nucleotide polymorphisms, as previously described[56]. Beta (β) values were calculated from unmethylated (*U*) and methylated (*M*) signal [$M / (U + M + 100)$] and ranged from zero to one (0–100% methylation). β-values of loci whose detection *P* values were >0.01 were assigned 'NA' in the output file. To map EPIC arrays to the hg38/GRCh38 assembly, all probes were annotated with the EPIC. hg38.manifest.tsv.gz files as previously described[57].

For initial visualization of the EPIC data, multidimensional scaling plots were generated using the 'mdsPlot' function in the *minfi* Bioconductor package (v.1.34.0)[55]. Differential analyses were then performed between treatment arms with decitabine versus vehicle samples. For each comparison, β-values were transformed using logit transformation: $M = \log_2[\beta / (1 - \beta)]$. The R package DMRcate (v.2.2.3)[58] was used to identify DMRs, with DMP *P* value cut-offs of FDR < 0.01. DMRs were defined as regions with a maximum of 1,000 nucleotides between consecutive probes, a minimum of two CpG sites and a methylation change of >30%; we applied Benjamini–Hochberg correction for multiple testing. ChromHMM data downloaded from the Gene Expression Omnibus (GEO) (GSE118716) for TAMR MCF7 cells was used to annotate DMPs to chromatin states. REMP R package (v.1.14.0)[17] was used to assess genome-wide locus-specific DNA methylation of repeat elements (LTR, LINE1 and Alu) from EPIC data with IlluminaHuman-MethylationEPICanno.ilm10b5.hg38 annotation (GitHub).

## Hi-C analyses

Hi-C sequenced reads (150 bp paired-end) were quality checked with FastQ Screen v.0.14.1[59] for mouse host reads contamination. Reads were then processed with Xenome (v.1.0.1)[60] as previously described[61]. The remaining reads were aligned to the human genome (hg38/GRCh38) using HiC-Pro[62] (v.2.11.4). Initially, to generate Hi-C contact matrices, the aligned Hi-C reads were filtered and corrected using the ICE correction algorithm built into HiC-Pro, which corrects for the copy number variation-related variability in the tumors. Inter-chromosomal interactions were excluded from further analyses to control for the effect of inter-chromosomal translocations in the tumors. Contact matrices for 3D genome feature annotation and visualization were created and Knight–Ruiz normalized using Juicer tools[63] using contact matrices in.hic format generated by hicpro-2juciebox script in HiC-Pro as input (hic file v.8). We confirmed data quality by assessing the proportion of cis–trans interactions and the percentage of valid fragments for each library. Overall, we obtained an average of 100 million unique, valid contacts per replicate (~310 million per treatment arm), for an average resolution of 10 kb. Statistics for each library can be found in Supplementary Table 5. These data were used to derive loops, TAD boundaries and chromosomal compartment structures.

## Insulation score and identification of TAD boundaries

TAD boundary calling was performed by calculating insulation scores in ICE normalized contact matrices at 20 kb resolution using TADtool[64]. To identify appropriate parameters, we called TADs across chromosome 1 using contact matrices at 20 kb and threshold values of 10, 50 and 100. The final TADs were called for all chromosomes at window 102,353 and

cut-off value 50. Boundaries that were found overlapping by at least one genomic bin between replicates were merged. Boundaries separated by at least one genomic bin were considered different between datasets (that is, consistently lost or gained across all replicates). Pyramid-like heatmap plots were generated with GENOVA[65].

### Identification of compartments A and B

For each chromosome in each sample, compartments were called using the standard PCA method[18] in the HOMER package (v.4.10). The resolution was set to 50 kb and the window size to 100 kb. Compartments were defined as regions of continuous positive or negative PC1 values using the findHiCCompartments.pl. To detect which compartment is the open A-type and which is the closed B-type, the genome-wide gene density was calculated to assign the A-type and B-type compartmentalization. To identify genomic regions that switch between two compartment types, we used the correlation difference script (getHiCcorrDiff.pl) with findHiCCompartments.pl. Compartments were considered common if they had the same compartment definition within the same genomic bin. Compartment changes between treatment arms were computed after considering compartments that were overlapping between biological replicates unless otherwise indicated.

To directly quantify the tendency of each region to interact with the other regions in either A or B compartments, we calculated the A:B interaction ratio, defined for each 100 kb genomic window as the ratio of interaction frequency with A versus B compartments using O/E matrix with GENOVA[65] (v.1.0.0). $\log_2$ contact enrichments were plotted as a heat saddle plot. Summarized A–A, B–B and B–A compartment strengths were calculated as the mean $\log_2$ contact enrichment between the top (A) or bottom (B) 20% of PC1 percentiles. The compartment strength ratio was calculated as $\log_2$(A–A:B–B).

### PCHi-C analyses

PCHi-C sequenced reads were mapped and filtered using HiCUP (v.0.7.4)[66] with the hg38/GRCh38 genome digested with the –arima flag and minimum di-tag length set to 20. Statistics for each library can be found in Supplementary Table 9. On target rate was calculated by counting the number of valid, unique reads overlapping bait fragments (minimum overlap > 0.6). Unique, valid mapped reads from HiCUP were converted into .chinput files using bam2chicago.sh utility, and obtained .chinput files were further filtered and processed with CHiCAGO (v.1.14.0)[67]. CHiCAGO design files were created with the following parameters to account for multiple restriction enzymes used in the Arima-HiC kit and the Arima-specific design of the bait fragments: MaxLBrowndist, 75,000; binsize, 1,500; minFragLen, 25; maxFragLen, 1,200. Significant interactions were called with CHiCAGO using a score cut-off of five. All bait-to-bait interactions were discarded. Chicdiff package[19] (v.0.6) was used to compare PCHi-C data from vehicle-treated and decitabine-treated tumors, and the difference in the mean asinh-transformed CHiCAGO scores between conditions above one was used to prioritize the potential differential promoter-anchored interactions. Only interactions with a CHiCAGO score of more than five in at least two replicates were included for downstream analysis. For downstream analysis of merged replicate data and for visualization of interactions in WashU Epigenome Browser[68], replicates were merged with CHiCAGO. We defined reprogrammed enhancer–promoter interactions by constructing a consensus, gained and lost subset of promoter-anchors (baits) and OE anchors based on CHiCAGO promoter interactions, Chicdiff analysis and setdiff R function across the replicates. The following criteria were used to obtain these regions: CHiCAGO score of >5 in two out of three replicates in either condition, Chicdiff generated asinh-transformed CHiCAGO scores between conditions above one and no overlap between regions, allowing for 10 kb maximal gap in three out of three replicates. Further quality control analyses are included in the Supplementary Note.

ChromHMM data downloaded from GEO (GSE118716) for TAMR MCF7 cells was used to annotate promoter-anchored interactions to chromatin states.

### RNA-seq data analyses

For canonical gene expression, RNA-seq raw reads were quality controlled and sequence adaptors were trimmed using Trim Galore (v.0.11.2), reads were processed with Xenome v.10.1 (ref. 60) to remove mouse sequences and the remaining reads were mapped with STAR (v.2.7.7a)[69] to the hg38/GRCh38 human genome build with GENCODE v.33 used as a reference transcriptome (parameter settings: –quantMode TranscriptomeSAM –outFilterMatchNmin 101 –outFilterMultimapNmax 20). Statistics for each library can be found in Supplementary Table 6 and differential gene expression analyses are included in Supplementary Table 7. TMM normalization was applied to normalize for RNA composition[70] and differential expression was performed with edgeR v.3.18.1 (ref. 71) using the generalized linear model. RNA-seq tracks were generated using bedtools v.2.22 genomeCoverageBed to create normalized .bedGraph files and bedGraphToBigWig (USCS utils) to create .bigwig files. Further quality control analyses are included in the Supplementary Note.

### ChIP-seq data analyses

ChIP-seq reads were aligned against the human genome (hg38/GRCh38) using bowtie2 with default parameters[72]. Non-uniquely mapped, low quality (MAPQ < 15) and PCR duplicate reads were removed. Peak calling of individual ChIP-seq experiments was performed with MACS2 with default parameters[73]. Statistics for each library can be found in Supplementary Table 4. Consensus peaks were identified by intersecting MACS2 peaks obtained from each sample using bedtools intersect (v.2.25.0) with minimum overlap > 0.6. Differential binding analyses were performed using DiffBind (v.3.0.9)[3] with FDR < 5%. Enrichment analyses were performed using GAT[74]. Merged bigwig tracks for visualization were created from merged bam files from all replicates using the bamCoverage function with scaling factor normalization and heatmaps and average profiles were plotted with deepTools2 (ref. 75). Further quality control analyses are included in the Supplementary Note. ChromHMM data downloaded from GEO (GSE118716) for TAMR MCF7 cells was used to annotate ERBS to chromatin states. Merged bigwig tracks for visualization were created from merged bam files from all replicates using the bamCoverage function with scaling factor normalization and heatmaps and average profiles were plotted with deepTools2 (ref. 75).

### Motif analyses

The HOMER motif discovery suite (v.4.10) was used for motif analysis, using random, matched regions as background. Motifs were ranked by log $P$ values from hypergeometric enrichment calculations (or binomial) to determine motif enrichment.

### CUT&RUN data analyses

Paired-end fastq files were down-sampled to 10 M reads per sample and aligned to the hg38 reference genome using the Bowtie v.2 algorithm. Only uniquely aligned reads were retained for subsequent analyses. Peak calling of individual CUT&RUN experiments was performed with MACS2 with default parameters[73]. Statistics for each library can be found in Supplementary Table 4. Consensus peaks were identified by intersecting MACS2 peaks obtained from each sample using bedtools intersect (v.2.25.0) with minimum overlap > 0.6. Differential binding analyses were performed using DiffBind (v.3.0.9)[3] with FDR < 5%. Merged bigwig tracks for visualization were created from merged bam files from all replicates using the bamCoverage function with scaling factor normalization, and heatmaps and average profiles were plotted with deepTools2 (ref. 75). Further quality control analyses are included in the Supplementary Note.

## Gene ontology analyses

Gene ontology enrichment analysis and pathway enrichment were done using GSEA (v.4.1.0) and MSigDB 7.2 (ref. 20). All significant biological processes and pathways had an adjusted $P$ value of <0.001.

## Statistical analyses

The Mann–Whitney–Wilcoxon test was used for two-group non-parametric comparisons. Unless otherwise stated, statistical tests were two-sided. A permutation test ($n = 1,000$ permutations) was used to calculate empirical $P$ values; the test does not make any assumptions about the underlying distribution of the data. The Benjamini–Hochberg method was used to control for multiple testing using an FDR procedure. Tumor growth curve data were analyzed at ethical endpoint using a two-tailed unpaired Student's $t$-test. Immunohistochemistry data were analyzed by a two-tailed, unpaired Student's $t$-test.

## Public datasets

ChIP-seq datasets were previously downloaded from GSE32222 (ref. 3). ChromHMM data was previously downloaded from GSE118716 (ref. 7).

## Reporting summary

Further information on research design is available in the Nature Portfolio Reporting Summary linked to this article.

## Data availability

All sequencing data created in this study have been uploaded to GEO (https://www.ncbi.nlm.nih.gov/geo/) and are available under primary accession codes GSE171074 and GSE216989. The public database of the hg38 genome and annotation files (v.33) are available from the GENCODE portal (https://www.gencodegenes.org). Biological material used in this study can be obtained from the authors upon request. Source data are provided with this paper.

## Code availability

Python script language (v.2.7.8 and v.3.9.1) and R (v.3.6.3 and v.4.0.3) were used to develop the bioinformatics methods and algorithms in this work. All code for Hi-C and PCHi-C analyses is available within the GitHub repository (https://github.com/JoannaAch/PDX_Decitabine_3DEpigenome).

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

## Acknowledgements

We thank members of the Clark Laboratory for helpful discussions and careful reading of the manuscript. We thank Arima Genomics for advice with performing Hi-C and PCHi-C in snap-frozen tumor tissues. S.J.C. is a National Health and Medical Research Council (NHMRC) Senior Principal Research Fellow (no. 1063559). J.A.K. is a National Breast Cancer Foundation (NBCF) Mavis Robertson Fellow (IIRS-21-047). E.L. is an NBCF Endowed Chair (EC17-02). T.E.H. is an NBCF Fellow (IIRS-19-009). C.S. is an NBCF Fellow (IIRS-18-137). Q.D. is a NHMRC Investigator Grant recipient (no. 1177792). R.P. is a NHMRC Investigator Grant recipient (no. 2010156). J.M.W.G is supported by a Breast Cancer Now Fellowship and the Tenovus Cancer Charity and Cardiff University. C.E.C. is an NBCF Mavis Robertson Fellow (IIRS-21-066) and Cancer Institute NSW Fellow (CDF1071). This research was supported by the NHMRC Project Grant (no. 1128916 to S.J.C), Movember & National Breast Cancer Foundation Collaboration Initiative grant (MNBCF-17-012 to T.E.H., E.L. and S.J.C.), National Breast Cancer Foundation Investigator-Initiated Research Scheme (IIRS-21-047 to J.A.K.) and the Cancer Council NSW grants (RG16-09 to S.J.C and RG20-04 to J.A.K.). Research funding (to S.J.C.) was provided by Van Andel Institute through the Van Andel Institute—Stand Up To Cancer Epigenetics Dream Team. Stand Up To Cancer is a division of

the Entertainment Industry Foundation, administered by the American Association for Cancer Research. The contents of the published material are solely the responsibility of the administering institution and individual authors and do not reflect the views of the NHMRC. The funders had no role in study design, data collection and analysis, decision to publish or preparation of the manuscript.

## Author contributions

S.J.C., J.A.K. and C.S. conceived the project. J.A.K., K.M.C., N.P., T.E.H., E.L., C.S. and S.J.C. created the experimental design. J.A.K., K.M.C., N.P., E.C., A.Y., A.W., G.L.L., S.N., W.Q., A.S., E.W., J.G., A.K., S.A. and J.S. were involved with methodology and data acquisition. J.A.K., E.C., H.H.M., B.M., R.P., W.C., S.C. and Q.D. analyzed the data. J.A.K., K.M.C., N.P., E.C., E.C.C., Q.D., W.C., T.E.H., E.L., C.S. and S.J.C. interpreted the data. J.A.K., T.E.H., E.L. and S.J.C. acquired funding. J.A.K. and S.J.C. wrote the manuscript with input from all authors.

## Competing interests

A.S. is an employee of Arima Genomics. The other authors declare no competing interests.

## Additional information

**Extended data** is available for this paper at https://doi.org/10.1038/s41594-023-01181-7.

**Correspondence and requests for materials** should be addressed to Joanna Achinger-Kawecka or Susan J. Clark.

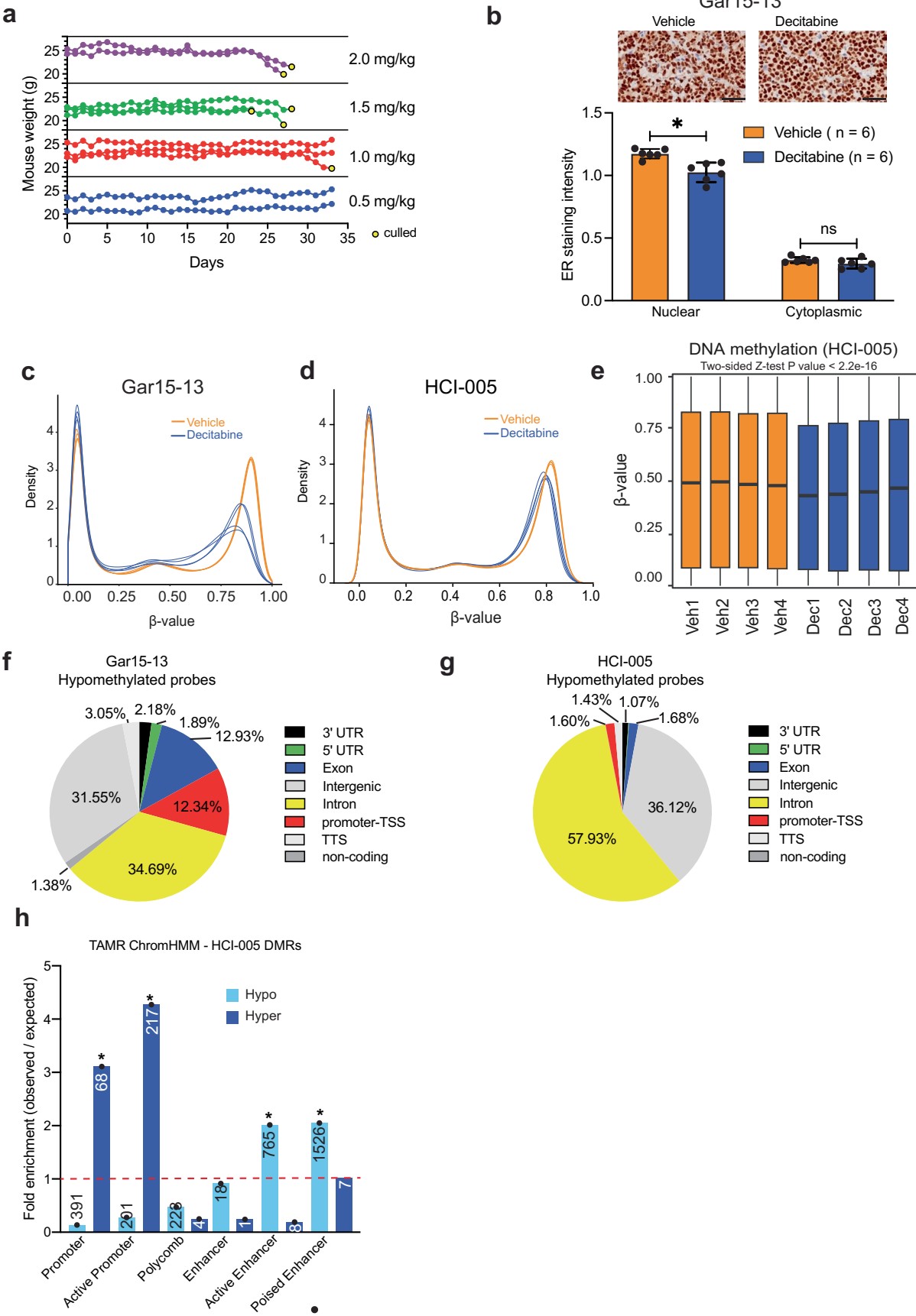

**Extended Data Fig. 1 | See next page for caption.**

**Extended Data Fig. 1 | Decitabine treatment induces DNA hypomethylation.**
**(a)** Mice were treated with indicated doses of Decitabine for 35 consecutive days and mice weight was assayed to determine the most appropriate Decitabine concentration. **(b)** Representative images of ER immunohistochemistry staining in Gar15-13 Vehicle- and Decitabine-treated PDXs. Scale bars, 50 μm. Below: Quantification of the ER immunohistochemistry staining in Gar15-13 Vehicle- and Decitabine-treated PDXs (n = 6 mice each). Data represented as mean ± SEM. Data analysed using a two-tailed Wilcoxon matched-pairs signed rank test. * P value < 0.01. Endpoint test details are mean Diff = 0.149 95% CI (0.06021-0.2388), P value = 0.0027. **(c)** Distribution of DNA methylation in Vehicle and Decitabine-treated Gar15-13 PDXs (n = 4 biological replicates each) for all EPIC probes. Black line indicates median ± SD. Box plots show median, inter-quartile range and maximum/minimum DNA methylation. Data analysed using the two-sided Z test. **(d)** Distribution of DNA methylation in Vehicle and Decitabine-treated HCI-005 PDXs (n = 4 biological replicates each) for all EPIC probes. Black line indicates median ± SD. Box plots show median, inter-quartile range and maximum/minimum DNA methylation. Data analysed using the two-sided Z test. **(e)** Distribution of DNA methylation in HCI-005 Vehicle and Decitabine-treated (n = 4 biological replicates each) tumours for all EPIC probes. Box plots show median, inter-quartile range and maximum/minimum DNA methylation. Data analysed using the two-sided Z test. **(f)** RefSeq annotation of Vehicle vs. Decitabine hypomethylated probes (Gar15-13). **(g)** RefSeq annotation of Vehicle vs. Decitabine hypomethylated probes (HCI-005). **(h)** Observed over expected fold change enrichment of DMRs in HCI-005 Decitabine treatment compared to Vehicle across TAMR ChromHMM regulatory regions. * P value < 0.001 (permutation test). The numbers of DMRs located within each specific regions are presented in the respective column. DMR, differentially methylated region.

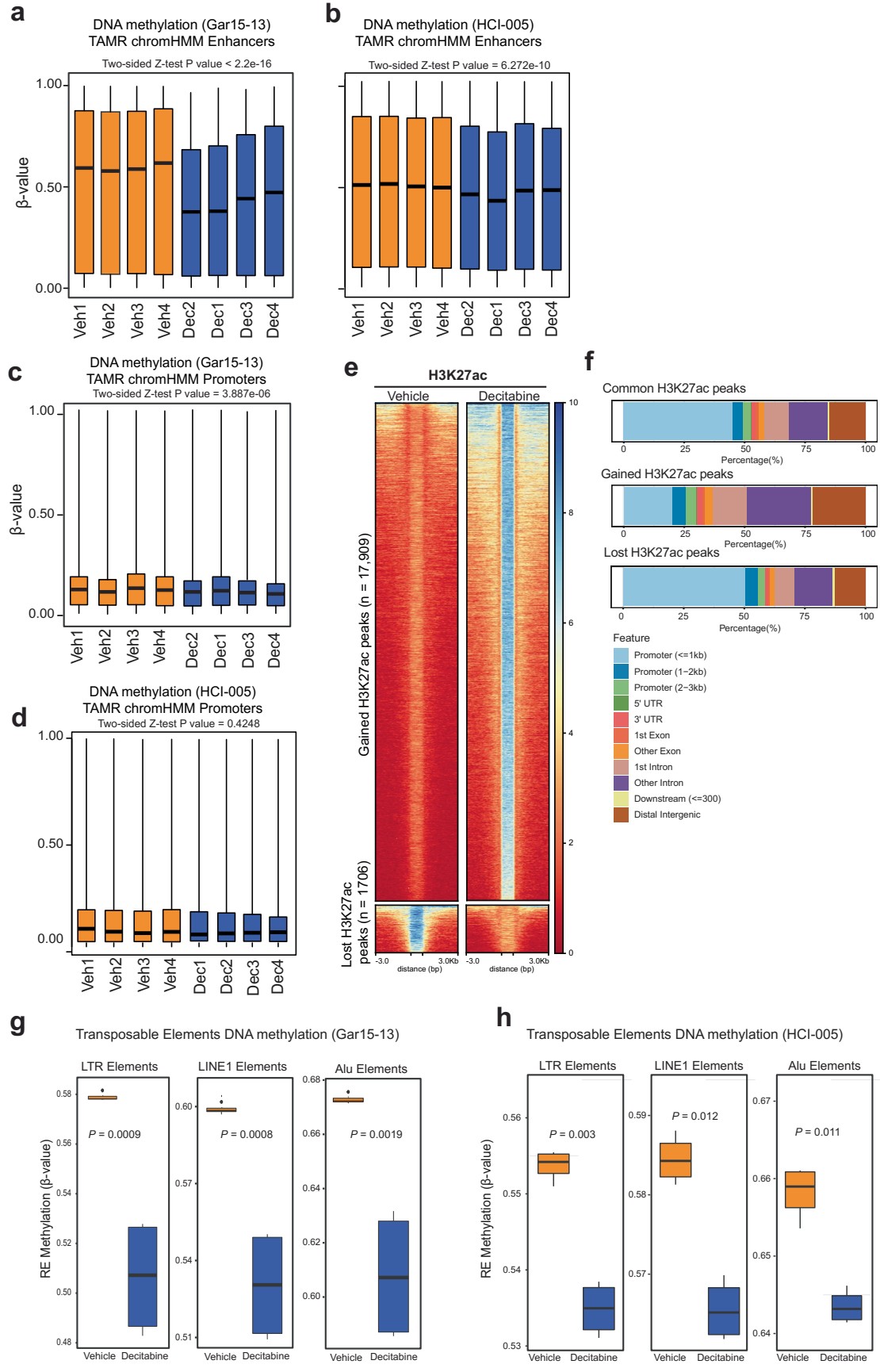

**Extended Data Fig. 2 | See next page for caption.**

**Extended Data Fig. 2 | Decitabine induced DNA hypomethylation at regulatory elements. (a)** Distribution of DNA methylation for Vehicle and Decitabine-treated (n = 4 biological replicates each) Gar15-13 PDXs for EPIC probes located at TAMR ChromHMM enhancer regions. Black line indicates median ± SD. Box plots show median, inter-quartile range and maximum/minimum DNA methylation. Data analysed using the two-sided Z test. **(b)** Distribution of DNA methylation for Vehicle and Decitabine-treated (n = 4 biological replicates each) HCI-005 PDXs for EPIC probes located at TAMR ChromHMM enhancer regions. Black line indicates median ± SD. Box plots show median, inter-quartile range and maximum/minimum DNA methylation. Data analysed using the two-sided Z test. **(c)** Distribution of DNA methylation for Vehicle and Decitabine-treated (n = 4 biological replicates each) Gar15-13 PDXs for EPIC probes located at TAMR ChromHMM promoter regions. Black line indicates median ± SD. Box plots show median, inter-quartile range and maximum/minimum DNA methylation. Data analysed using the two-sided Z test. **(d)** Distribution of DNA methylation for Vehicle and Decitabine-treated (n = 4 biological replicates each) HCI-005 for EPIC probes located at TAMR ChromHMM promoter regions. Black line indicates median ± SD. Box plots show median, inter-quartile range and maximum/minimum DNA methylation. Data analysed using the two-sided Z test. **(e)** H3K27ac CUT&RUN heatmap at H3K27ac binding sites gained and lost in Decitabine compared to Vehicle-treated Gar15-13 PDXs. **(f)** RefSeq annotation of Vehicle vs. Decitabine common, gained and lost H3K27ac binding sites in Gar15-13. **(g)** Distribution of DNA methylation for Vehicle and Decitabine-treated (n = 4 biological replicates each) Gar15-13 PDXs for EPIC probes mapping to LTR, LINE1 and Alu elements (REMP annotation). Black line indicates median ± SD. Box plots show median, inter-quartile range and maximum/minimum DNA methylation. Data analysed using the two-sided Z test. **(h)** Distribution of DNA methylation for Vehicle and Decitabine-treated HCI-005 PDXs for EPIC probes mapping to LTR, LINE1 and Alu elements (REMP annotation) (n = 4 biological replicates each). Black line indicates median ± SD. Box plots show median, inter-quartile range and maximum/minimum DNA methylation. Data analysed using the two-sided Z test.

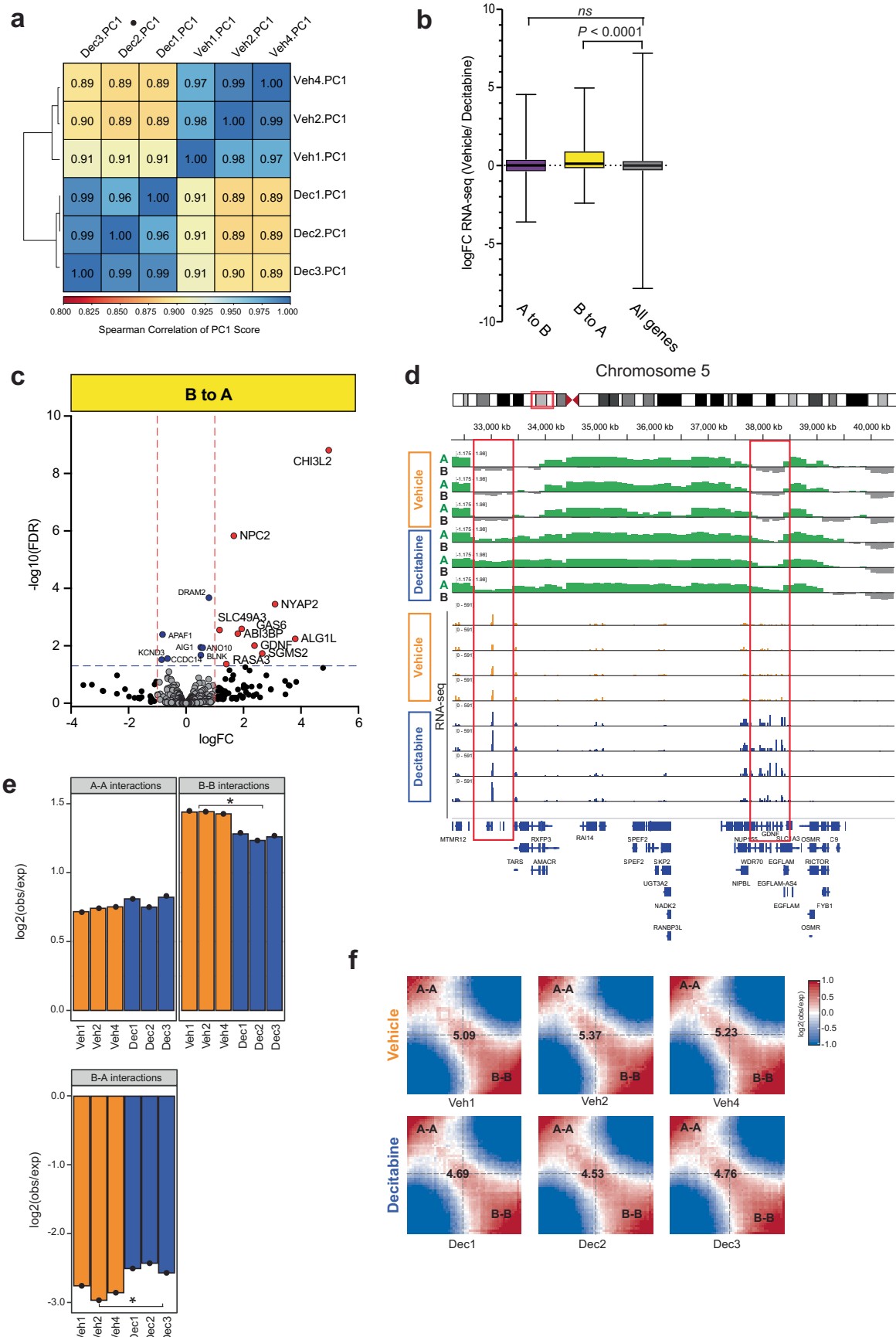

**Extended Data Fig. 3 | See next page for caption.**

**Extended Data Fig. 3 | Alteration to A/B compartment structure upon Decitabine treatment. (a)** Spearman pairwise correlations between the eigenvectors (PC1) in Decitabine-treated and Vehicle (n = 3 biological replicates each) Gar15-13 PDXs. Samples are ordered according to complete linkage hierarchal clustering. **(b)** logFC expression between Vehicle and Decitabine-treated (n = 3 biological replicates each) Gar15-13 PDXs of genes located either at A to B or B to A switching compartments. *P* value: two-sided Wilcox rank sum test. Box plots show median, inter-quartile range and maximum/minimum log fold change. **(c)** Decitabine *vs*. Vehicle differential expression of genes located at compartment that switched from B to A assignment in Decitabine-treated tumours. Information on all genes included in Supplementary Table 8. **(d)** Browser snapshot of Hi-C eigenvectors and RNA-seq in Vehicle and Decitabine-treated tumours (n = 3 biological replicates for Hi-C and n = 4 biological replicates for RNA-seq each), showing demarcation of open (A-type; positive values) and closed (B-type; negative values) compartment changes across a region on chromosome 5, which is associated with increased expression of the *GDNF* gene located within this region. **(e)** log2 observed over expected A – A, B – B and B – A compartment interactions in Vehicle (n = 3 biological replicates) for and Decitabine (n = 3 biological replicates) Gar15-13 PDXs. * *P* value two-tailed Students *t*-test < 0.05. **(f)** Average contact enrichment between pairs of 50Kb loci arranged by their PC1 eigenvector in Vehicle and Decitabine-treated tumours. The numbers at the center of the heatmaps indicate compartment strength calculated as the log2 ratio of (A–A + B–B)/ (A–B + B–A) using the mean values.

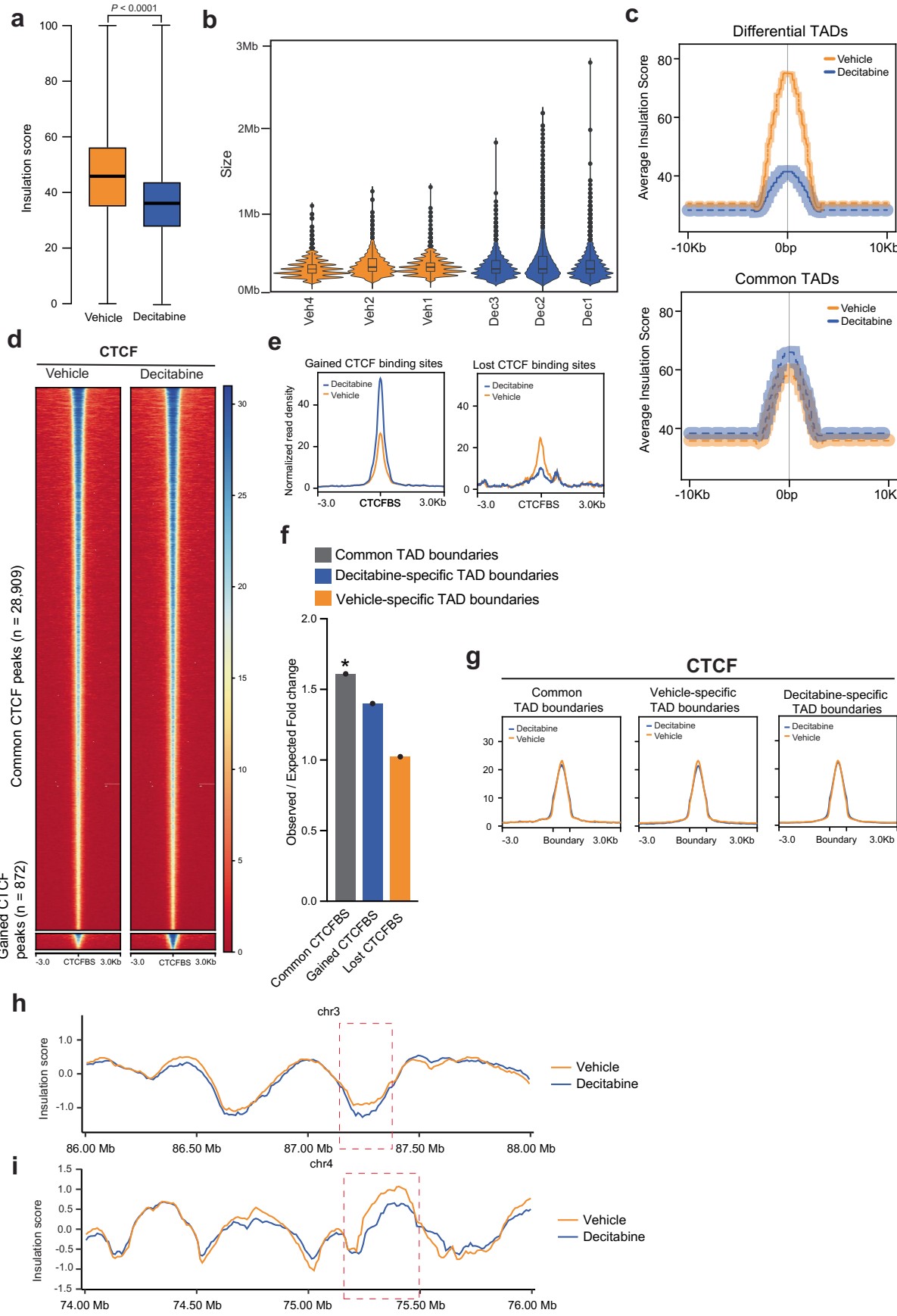

**Extended Data Fig. 4 | See next page for caption.**

**Extended Data Fig. 4 | Alteration to TAD structure upon Decitabine treatment. (a)** Average insulation score calculated by TADtool at 50Kb resolution in three biological replicates of Decitabine-treated and Vehicle Gar15-13 PDXs. *P* value two-sided Wilcox rank sum test. Box plots show median, inter-quartile range and maximum/minimum insulation score. **(b)** Distribution in TAD sizes for Vehicle and Decitabine treated Gar15-13 PDXs (n = 3 biological replicates each). Box plots show median, inter-quartile range and maximum/minimum log fold change. **(c)** Average insulation score at differential and common TADs. Lines show mean values, while light shading represents SEM. **(d)** CTCF CUT&RUN heatmaps at CTCF binding sites gained and lost in Decitabine compared to Vehicle-treated Gar15-13 PDXs. **(e)** Average signal intensity of CTCF CUT&RUN binding (Gar15-13 Vehicle- and Decitabine-treated PDXs) at gained and lost CTCF binding sites with Decitabine treatment. **(f)** Observed over expected fold change enrichment of common, gained, and lost CTCF binding sites

(CTCFBS) in Decitabine treated Gar15-13 tumours compared to Vehicle across unaltered, Decitabine-specific and Vehicle-specific TAD boundaries. * P value < 0.001 (permutation test). **(g)** Average signal intensity of CTCF CUT&RUN binding (Gar15-13 Vehicle- and Decitabine-treated tumours) at unaltered, Vehicle-specific and Decitabine-specific TAD boundaries. **(h)** Snapshot of region on chromosome 3, showing insulation score calculated in Vehicle and Decitabine-treated tumour Hi-C matrixes, demonstrating loss of TAD boundary insulation is Decitabine-treated samples (indicated with a red box). Merged Hi-C data from n = 3 biological replicates shown at 10Kb resolution. **(i)** Snapshot of region on chromosome 4, showing insulation score calculated in Vehicle and Decitabine-treated tumour Hi-C matrixes, demonstrating loss of TAD boundary insulation is Decitabine-treated samples (indicated with a red box). Merged Hi-C data from n = 3 biological replicates shown at 10Kb resolution.

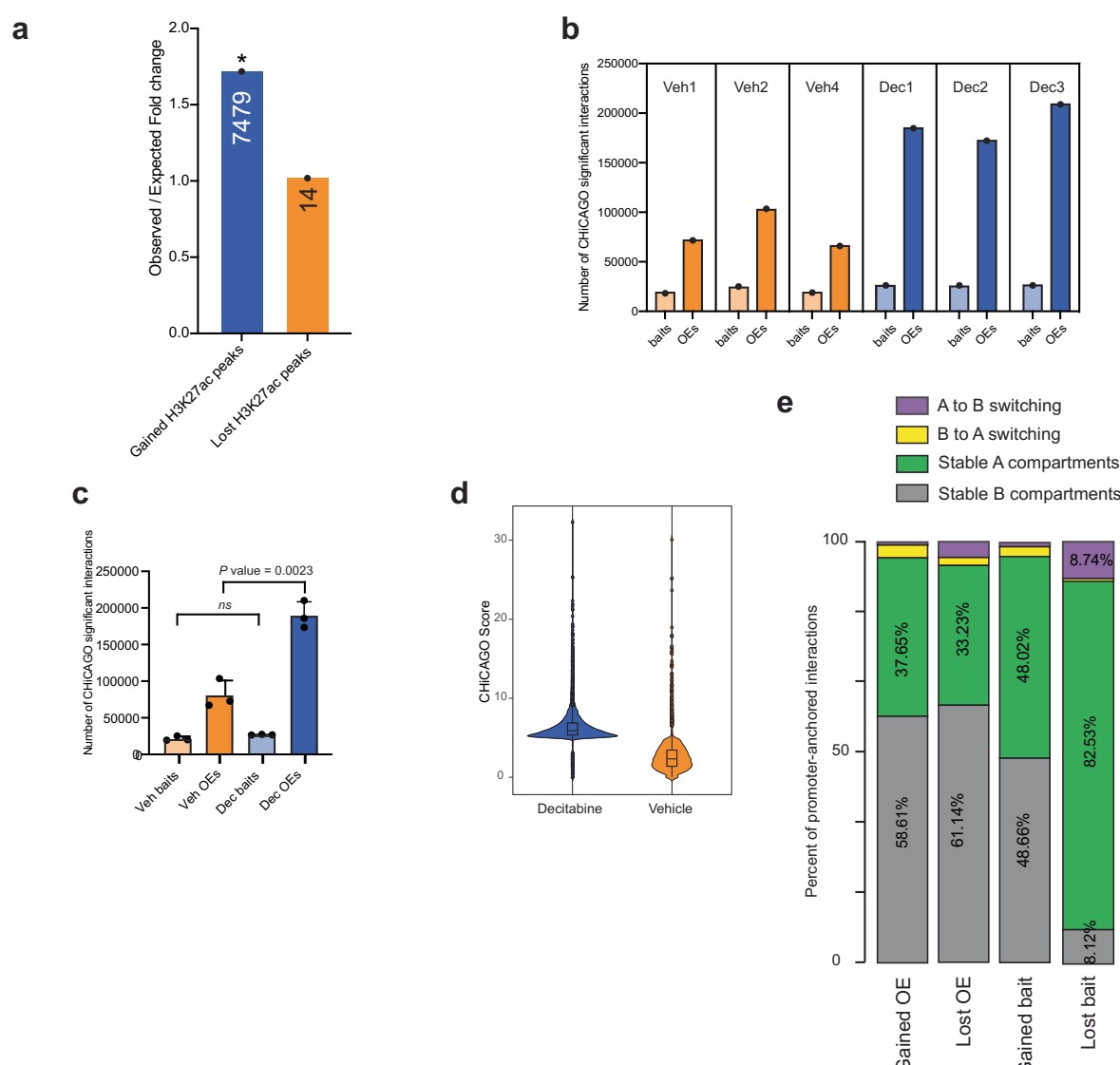

**Extended Data Fig. 5 | Alteration to promoter-anchored interactions upon Decitabine treatment. (a)** Observed over expected fold change enrichment of gained OE interacting regions and gained and lost H3K27ac binding sites. * P value < 0.001 (permutation test). Size of the overlap is presented in the respective column. **(b)** Number of promoter baits and enhancer OEs involved in significant CHiCAGO interactions for each of the PCHi-C maps from Vehicle and Decitabine-treated Gar15-13 tumours (n = 3 biological replicates each). **(c)** Average number of promoter baits and enhancer OEs involved in significant CHiCAGO interactions across the three Vehicle and three Decitabine-treated PCHi-C replicates. Error bars indicate SD. P value two-sided Wilcoxon rank-sum test. **(d)** ChiCAGO scores of promoter-anchored interactions identified from PCHi-C data in Decitabine and Vehicle tumours. Data from n = 3 biological replicate tumours shown. Box plots show median, inter-quartile range and maximum/minimum log fold change. **(e)** Percentage of promoter-anchored interactions located in chromosomal compartments that switched assignments (A to B or B to A) and stable A or B compartments.

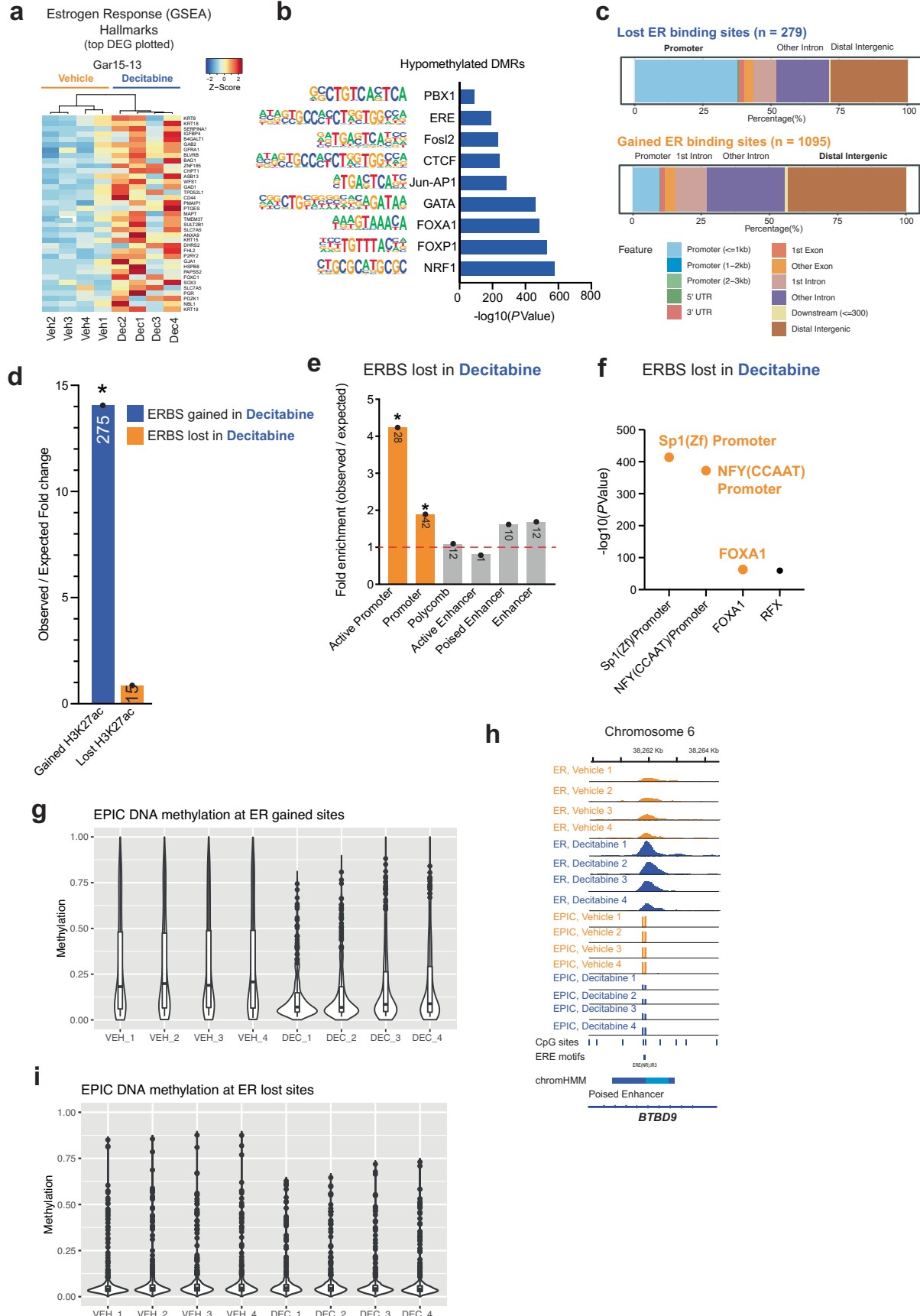

**Extended Data Fig. 6 | See next page for caption.**

**Extended Data Fig. 6 | Alterations to Estrogen Receptor binding upon Decitabine treatment. (a)** RNA-seq heatmap of Decitabine-induced changes in expression of genes belonging to the Estrogen Response (GSEA) Hallmarks. Top differentially expressed genes plotted (FDR < 0.05). **(b)** Transcriptions factor motifs enriched at hypomethylated DMRs between Vehicle- and Decitabine-treated Gar15-13 PDXs compared to matched random regions across the genome. Only motifs with binomial *P* value < 0.05 are shown. **(c)** RefSeq annotation of Vehicle vs. Decitabine lost and gained ERBS in Gar15-13. **(d)** Observed over expected fold change enrichment of gained ER binding sites (ERBS) and gained and lost H3K27ac binding sites. *P value < 0.001 (permutation test). Size of the overlap is presented in the respective column. **(e)** ChromHMM (TAMR) annotation (*P value < 0.001, permutation test) of ER binding sites lost with Decitabine treatment compared to matched random regions across the genome. Size of the overlap is presented in the respective column. **(f)** Motifs enriched

at ERBS lost with Decitabine treatment compared to matched random regions generated from ERE binding motifs across the genome. Only motifs with binomial *P* value < 0.05 are shown. **(g)** DNA methylation levels at gained ERBS in Decitabine-treated (n = 4 biological replicates) and Vehicle (n = 4 biological replicates) PDX Gar15-13 PDXs. Box plots show median, inter-quartile range and maximum/minimum log fold change. **(h)** Browser snapshot of ER ChIP-seq and EPIC DNA methylation (Vehicle and Decitabine-treated PDXs, n = 4 biological replicates each), showing concomitant gain of ER binding and loss of DNA methylation at enhancer of ER target gene *BTBD9*. **(i)** DNA methylation levels at lost ERBS in Decitabine-treated (n = 4 biological replicates) and Vehicle (n = 4 biological replicates) Gar15-13 PDXs. Box plots show median, inter-quartile range and maximum/minimum log fold change. DMR, differentially methylated region. ERBS, ER binding site.

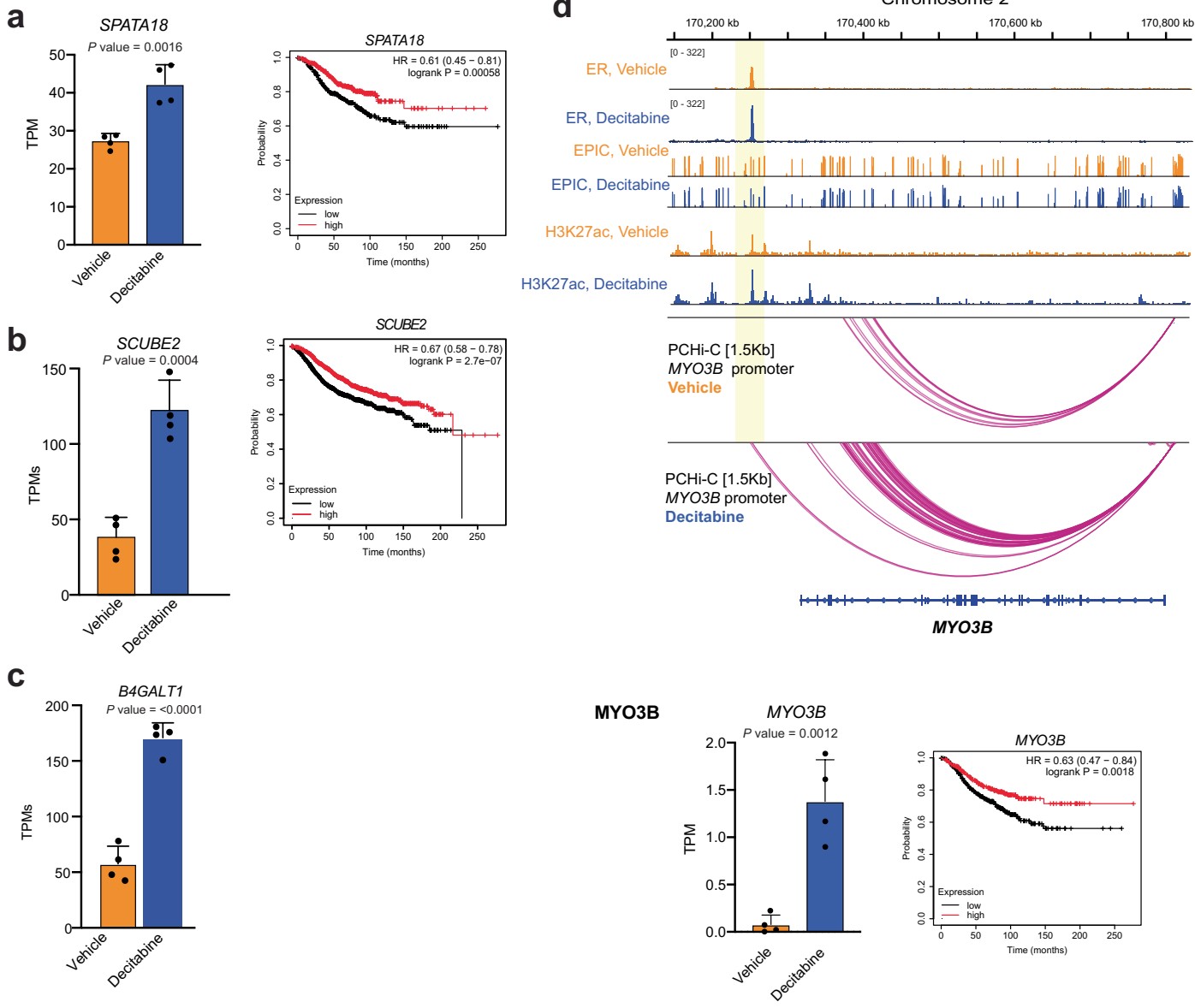

**Extended Data Fig. 7 | Expression of genes at ER-bound chromatin interactions. a)** The relative mRNA expression levels of *SPATA18* gene from RNA-seq (two-tailed *t*-test *P* < 0.05 derived from n = 4 biological replicates). Error bars indicate SD from four samples. Kaplan–Meier survival plot showing the ability of *SPATA18* gene to stratify ER+ breast cancer patients in the METABRIC cohort into good and poor outcome groups. Data analysed using the log-rank test. *P* values indicated within the graph. **(b)** The relative mRNA expression levels of *SCUBE2* gene from RNA-seq (two-tailed *t*-test *P* < 0.05 derived from n = 4 biological replicates). All data are represented as mean ± SD. Error bars indicate SD from four samples. Kaplan–Meier survival plot showing the ability of *SCUBE2* gene to stratify ER+ breast cancer patients in the METABRIC cohort into good and poor outcome groups. Data analysed using the log-rank test. *P* values indicated within the graph. **(c)** The relative mRNA expression levels of *B4GALT1* gene from RNA-seq (two-tailed *t*-test *P* < 0.05 derived from n = 4 biological replicates). All

data are represented as mean ± SD. Error bars indicate SD from four samples. **(d)** Browser snapshots showing the promoter-anchored interactions at the *MYO3B* ER target gene, together with ER ChIP-seq, EPIC DNA methylation, H3K27ac CUT&RUN signal, ChromHMM track and PCHi-C interaction track. Merged replicate data shown (n = 4 biological replicates each, n = 3 biological replicates for CUT&RUN and PCHi-C). In Decitabine-treated tumours, the *MYO3B* promoter displays increased number of interactions with an enhancer, which gains ER and H3K27ac binding with Decitabine treatment. The relative expression of the *MYO3B* gene was significantly upregulated in Decitabine-treated tumours (two-tailed *t*-test *P* < 0.05 derived from n = 4 biological replicates) and associated with good outcome in ER+ breast cancer patients in the METABRIC cohort. Data analysed using the log-rank test. *P* values indicated within the graph. All data are represented as mean ± SD. Error bars indicate SD from four samples.

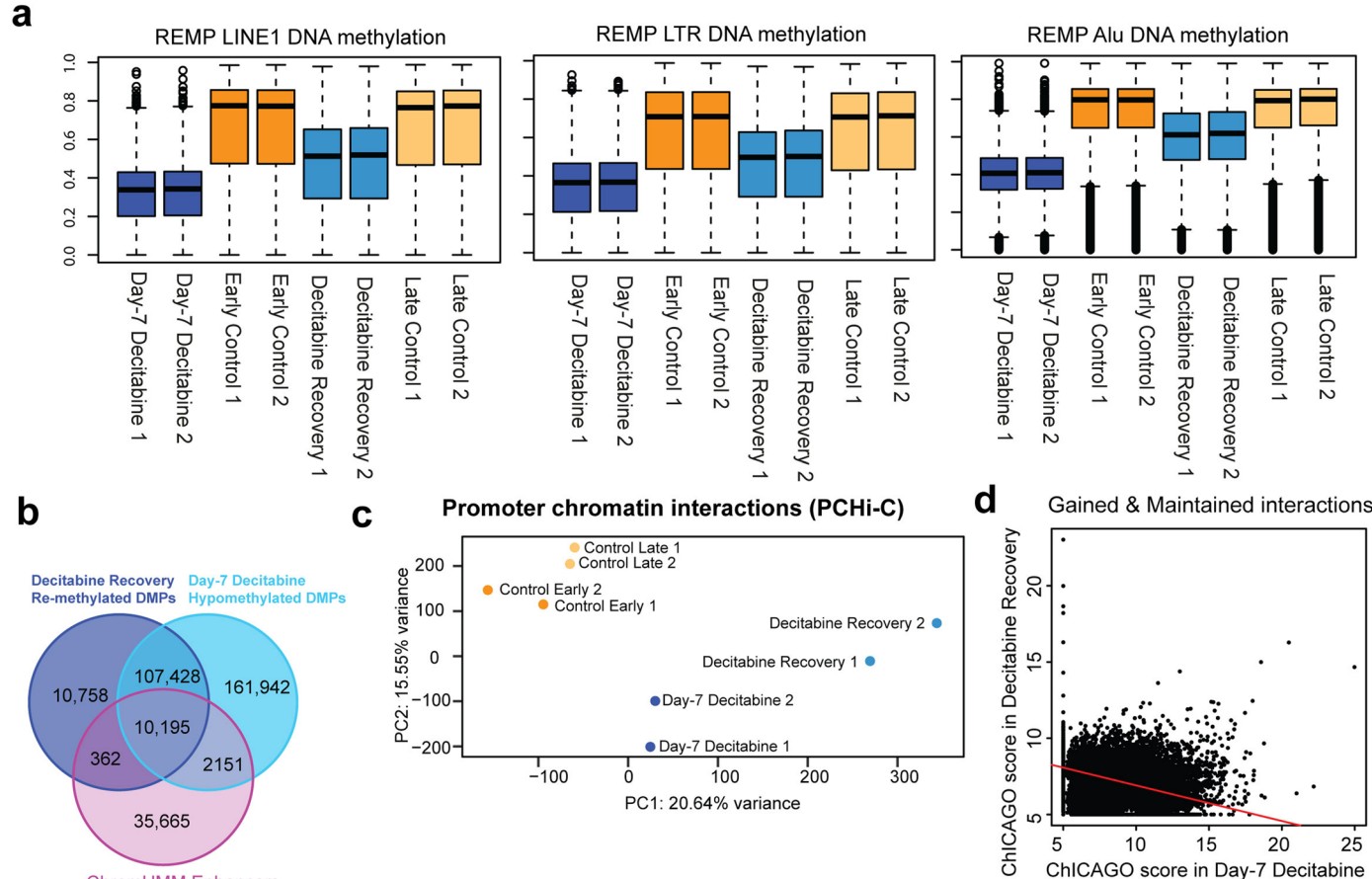

**Extended Data Fig. 8 | 3D epigenome dynamics in time-course of Decitabine treatment. (a)** Distribution of DNA methylation for Control (Control Early/Late), Day-7 Decitabine and Decitabine Recovery (n = 2 technical replicates each) TAMR cells for EPIC probes mapping to LTR, LINE1 and Alu elements (REMP annotation). Black line indicates median ± SD. Box plots show median, inter-quartile range and maximum/minimum DNA methylation. **(b)** Overlap of DMPs that are re-methylated in Decitabine Recovery compared to Day-7 Decitabine, Day-7 Decitabine hypomethylated DMPs and ChromHMM enhancer regions. **(c)** Principal component analysis showing the relationship among the PCHi-C promoter-anchored interactions of TAMR cells treated with Decitabine (Day-7 Decitabine) and following 28 days recovery (Decitabine Recovery) as well as matched control cells (Control Early and Late) (n = 2 technical replicates each). Data plotted for normalised ChICAGO interaction scores. **(d)** CHiCAGO significance scores for each "Gained & Maintained" chromatin interaction in Day-7 Decitabine and Decitabine Recovery TAMR cells. Merged data across n = 2 technical replicates shown.

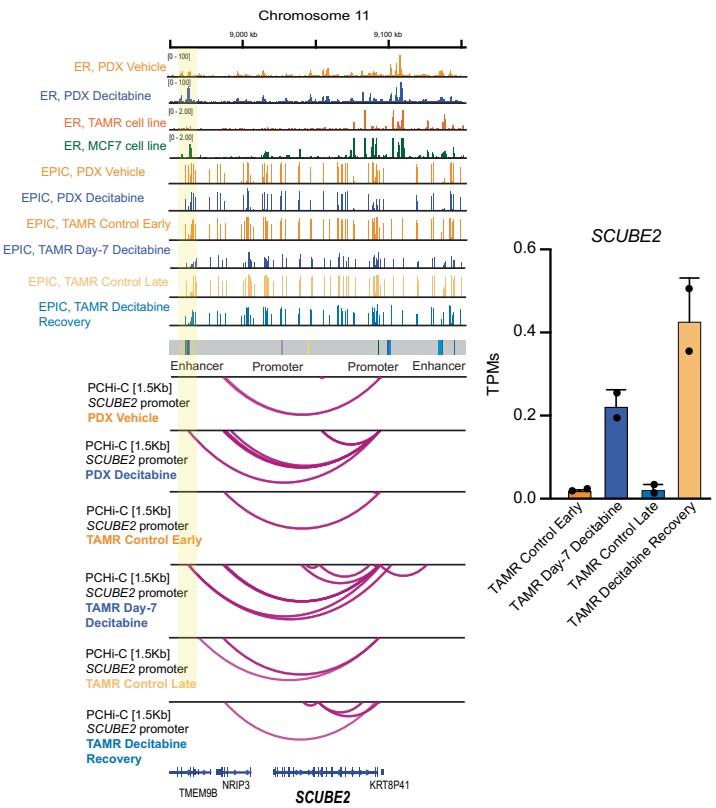

**Extended Data Fig. 9 | See next page for caption.**

**Extended Data Fig. 9 | Dynamic expression of genes at ER-bound chromatin interactions in time-course of Decitabine treatment. (a)** Browser snapshots showing promoter-anchored interactions at the *EVL* ER target gene. Gar15-13 PDX Vehicle- and Decitabine-treated data tracks are overlayed with ER ChIP-seq for TAMR/MCF7 cell lines[3]; EPIC methylation for TAMRs; ChromHMM track and finally PCHi-C for TAMR cell line data. Merged replicate data shown (n = 4 biological replicates each for Gar15-13 and n = 2 technical replicates for TAMRs) In Decitabine-treated PDXs and TAMRs (Day-7 Decitabine), the *EVL* promoter displays additional interactions with enhancer region, which gains ER binding with Decitabine treatment in PDXs, concomitant with loss of DNA methylation in both PDXs and TAMRs. These ectopic interactions are lost after 28 days of recovery with recovery of DNA methylation at that locus. Expression of the *EVL* gene was significantly upregulated in Decitabine-treated *vs*. Vehicle PDXs (bottom right, RNA-seq TPM) and increased in Day-7 Decitabine TAMRs compared to Control Early and was restored in Decitabine Recovery (top right, RNA-seq TPM). **(b)** Browser snapshots showing promoter-anchored interactions at the *MYO3B* ER target gene. Gar15-13 Vehicle- and Decitabine-treated PDX data tracks are overlayed with ER ChIP-seq for TAMR/MCF7[3]; EPIC methylation for TAMRs; ChromHMM track and finally PCHi-C for TAMRs. In Decitabine-treated PDXs and TAMRs (Day-7 Decitabine), the *MYO3B* promoter displays increased number of interactions with an enhancer, which gains ER binding with Decitabine treatment in PDXs. The relative expression of the *MYO3B* gene increased in Day-7 Decitabine-treated TAMRs (RNA-seq TPM). Error bars indicate SD from two samples. **(c)** Browser snapshots showing promoter-anchored interactions at the *SCUBE2* ER target gene. Gar15-13 PDX Vehicle- and Decitabine treatment data tracks are overlayed with ER ChIP-seq for TAMR/MCF7[3]; EPIC methylation for TAMRs, ChromHMM track and finally PCHi-C. In Day-7 Decitabine-treated PDXs and TAMRs, the *SCUBE2* promoter displays additional interactions with a distal enhancer, which gains ER binding with Decitabine treatment. Expression of the *SCUBE2* gene was upregulated in Day-7 Decitabine and its expression continued to increase in Decitabine Recovery. Error bars indicate SD from two samples. Merged replicate data shown (n = 4 biological replicates for Gar15-13 and n = 2 technical replicates for TAMRs).

# Reporting Summary

## Statistics

For all statistical analyses, confirm that the following items are present in the figure legend, table legend, main text, or Methods section.

| n/a | Confirmed | |
|---|---|---|
| ☐ | ☒ | The exact sample size (*n*) for each experimental group/condition, given as a discrete number and unit of measurement |
| ☐ | ☒ | A statement on whether measurements were taken from distinct samples or whether the same sample was measured repeatedly |
| ☐ | ☒ | The statistical test(s) used AND whether they are one- or two-sided *Only common tests should be described solely by name; describe more complex techniques in the Methods section.* |
| ☒ | ☐ | A description of all covariates tested |
| ☐ | ☒ | A description of any assumptions or corrections, such as tests of normality and adjustment for multiple comparisons |
| ☐ | ☒ | A full description of the statistical parameters including central tendency (e.g. means) or other basic estimates (e.g. regression coefficient) AND variation (e.g. standard deviation) or associated estimates of uncertainty (e.g. confidence intervals) |
| ☐ | ☒ | For null hypothesis testing, the test statistic (e.g. *F*, *t*, *r*) with confidence intervals, effect sizes, degrees of freedom and *P* value noted *Give P values as exact values whenever suitable.* |
| ☒ | ☐ | For Bayesian analysis, information on the choice of priors and Markov chain Monte Carlo settings |
| ☒ | ☐ | For hierarchical and complex designs, identification of the appropriate level for tests and full reporting of outcomes |
| ☐ | ☒ | Estimates of effect sizes (e.g. Cohen's *d*, Pearson's *r*), indicating how they were calculated |

*Our web collection on statistics for biologists contains articles on many of the points above.*

## Software and code

Policy information about availability of computer code

Data collection | All data collection used open-access or commercially available software as outlined below:

The Illumina NextSeq 500, HiSeq X10 and NovaSeq S4 were used for RNA-seq, ChIP-seq, CUT&RUN, Hi-C and PCHi-C studies.
The Nanozoomer slide scanner was used to capture images of stained tissue sections.

Graphpad Prism 8 (Graphpad Software, Inc.)
minfi (v.1.34.0) (Aryee et al., 2014)
limma (v.3.46) (Ritchie et al., 2015)
conumee (v.1.9.0)
DMRcate (v.2.2.3) (Peters et al., 2015)
REMP (v.1.14.0) (Zheng et al., 2017)
FastQ Screen (v.0.14.1) (Wingett and Andrews, 2018)
Bismark (v.0.24.0) (Krueger et al., 2011)
methclone (v.0.1.0) (Li et al., 2014)
metheor (v.1) (Lee et al., 2022)
sCNAphase (Chen et al., 2017)
Xenome (v.1.0.1) (Conway et al., 2012)
HiC-Pro (v.2.11.4) (Servant et al., 2015)
Juicer (v.1.6) (Durand et al., 2016)
TADtool (v.0.76) (Kruse et al., 2016)
GENOVA (v.0.95) (van der Weide et al., 2021)
Homer (v.4.8) (Heinz et al., 2010)
HiCUP (v.0.7.4) (Wingett et al., 2015)
HiNT (v.2.2.7) (Wang et al., 2020)

CHiCAGO (v.1.14.0) (Cairns et al., 2016)
Chicdiff (v.0.6) (Cairns et al., 2019)
EnhancedVolcano (v.1.8.0) (Blighe et al., 2018)
STAR (v.2.7.7a) (Dobin et al., 2013)
edgeR (v.3.18.1) (Robinson et al., 2010)
bedtools (v.2.25) (Quinian and Hall, 2010)
TEtranscripts (v.2.2.1) (Jin et al., 2015)
Bowtie2 (v2.3.4.1) (Langmead and Salzberg, 2012)
TrimGalore (v0.6.10)
MACS2 (v2.2.6) (Zhang et al., 2008)
DESeq2 (v.1.3.0) (Love et al., 2014)
GAT (v.1.3.4) (Heger et al., 2013)
DiffBind (v.3.0.9) (Ross-Innes et al., 2012)
ChIPseeker (v.1.26.0) (Yu et al., 2015)
deepTools2 (v.3.5.0) (Ramirez et al., 2016)
GSEA (v.4.1.0)
MSigDB (v.7.2) (Subramanian et al., 2005)
cBioPortal (Cerami et al., 2012)
survminer (v.0.4.9)

Data analysis | All analyses were performed using open source software. All software code used to analyze the data for this study is publicly available as described in the methods section. Python script language (v.2.7.8 and v.3.9.1) and R (v.3.6.3 and v.4.0.3) were used for bioinformatics methods and algorithms in this work. All code for Hi-C, PCHi-C, ChIP-seq and RNA-seq analyses is publicly available within the GitHub repository https://github.com/JoannaAch/PDX_Decitabine_3DEpigenome.

For manuscripts utilizing custom algorithms or software that are central to the research but not yet described in published literature, software must be made available to editors and reviewers. We strongly encourage code deposition in a community repository (e.g. GitHub). See the Nature Portfolio guidelines for submitting code & software for further information.

## Data

Policy information about availability of data

All manuscripts must include a data availability statement. This statement should provide the following information, where applicable:
- Accession codes, unique identifiers, or web links for publicly available datasets
- A description of any restrictions on data availability
- For clinical datasets or third party data, please ensure that the statement adheres to our policy

All sequencing data created in this study have been uploaded to the Gene Expression Omnibus (GEO; https://www.ncbi.nlm.nih.gov/geo/) and are available under primary accession code GSE171074 and GSE216989. Public datasets include: ChIP-seq data sets downloaded from GSE32222 by Ross-Innes et al., 2012, ChromHMM data downloaded from GSE118716 by Achinger-Kawecka et al., 2020. All data was mapped to hg38 human reference genome.

# Field-specific reporting

Please select the one below that is the best fit for your research. If you are not sure, read the appropriate sections before making your selection.

☒ Life sciences    ☐ Behavioural & social sciences    ☐ Ecological, evolutionary & environmental sciences

For a reference copy of the document with all sections, see nature.com/documents/nr-reporting-summary-flat.pdf

# Life sciences study design

All studies must disclose on these points even when the disclosure is negative.

Sample size | Patient-derived tumour xenograft (PDX) models generated from two different endocrine-resistant, metastatic ER+ breast cancer patients (Gar15-13 and HCI-005) were used to account for biological and clinical variability between patients. Decitabine treatment was performed on 8 (Gar15-13) and 7 (HCI-005) individual PDX mice to obtain sufficient sample size based on sample size calculation for standard statistical tests (80% statistical power to detect 1.3 SD difference and 95% power to detect 1.7 SD difference), with the exact number of replicates in the figure legends.

No statistical method was used to determine sample sizes in cell line experiments. Sample sizes were selected prior to knowledge of the outcome. No power analyses were carried out.

EPIC DNA methylation, ER ChIP-seq and RNA-seq experiments were performed in quadruplicates and Hi-C, Promoter Capture Hi-C and CUT&RUN were performed in triplicates to assess statistical significance.

EPIC DNA methylation, RNA-seq experiments and Promoter Capture Hi-C in TAMR cells were performed in duplicates.

Data exclusions | Sample sizes differed between in vivo xenograft tumour growth experiments, as some tumours did not grow at the expected rate. Such outliers were excluded from further data analyses.

Replication | Decitabine treatment was performed in two independent patient-derived xenografts (PDXs) and the tumour inhibiting effect of Decitabine

| Replication | was replicated in both models, across multiple mice. Two unique tumour xenograft models (Gar15-13, HCI-005) were used in this study to ensure consistent responses across a variety of tumours. Most assays were performed in at least a biological triplicate. All experiments were able to be reliably reproduced. |
|---|---|
| | Hi-C experiments were performed in triplicates and reproducibility between replicates was verified using HiCRep (Yang T (2018)). EPIC, RNA-seq and ER ChIP-seq experiments were performed in four replicates in 2 PDX models. All findings were reproducible and instances of variability are discussed in the text. |
| | Final conclusions were validated in an independent cell line model of endocrine-resistance (TAMR) with 7 days of Decitabine treatment. |
| Randomization | PDX mice were randomised to treatment arms when tumours reached 200mm3 using an online randomisation tool (https://www.graphpad.com/quickcalcs/randomize1.cfm) (n = 6 - 8 mice per group for therapeutic studies, exact numbers specified in figure legends). Cells were randomly split from the same pool of cells before subject to treatments. Randomization was not applicable to other experiments. |
| Blinding | In vivo experiments utilized blinded animal technicians for assessing disease severity. The investigators were not blinded to the group allocation during data collection and outcome assessment. In order to analyse data and assign samples to the correct group, experimenters needed to be unblinded. |

# Reporting for specific materials, systems and methods

We require information from authors about some types of materials, experimental systems and methods used in many studies. Here, indicate whether each material, system or method listed is relevant to your study. If you are not sure if a list item applies to your research, read the appropriate section before selecting a response.

## Materials & experimental systems

| n/a | Involved in the study |
|---|---|
| ☐ | ☒ Antibodies |
| ☐ | ☒ Eukaryotic cell lines |
| ☒ | ☐ Palaeontology and archaeology |
| ☐ | ☒ Animals and other organisms |
| ☒ | ☐ Human research participants |
| ☒ | ☐ Clinical data |
| ☒ | ☐ Dual use research of concern |

## Methods

| n/a | Involved in the study |
|---|---|
| ☐ | ☒ ChIP-seq |
| ☒ | ☐ Flow cytometry |
| ☒ | ☐ MRI-based neuroimaging |

## Antibodies

| Antibodies used | ChIP-seq:<br>5ug Anti-Estrogen Receptor alpha (HC-20), Santa Cruz Biotechnology (Cat#sc-543; RRID: AB_631471)<br>CUT&RUN:<br>0.5ug CTCF (CTCF CUTANA™ CUT&RUN Antibody (cat. #13-2014))<br>0.5ug H3K27ac (Histone H3K27ac Antibody, SNAP-ChIP Certified (cat. #13-0045))<br>0.5ug IgG (CUTANA Rabbit IgG CUT&RUN Negative Control (cat. #13-0042))<br><br>IHC:<br>Monoclonal Mouse Anti-Human Estrogen Receptor α, Clone 1D5, Agilent (Cat# M7047, RRID:AB_2101946)<br>Monoclonal Mouse Anti-Human Ki-67 Antigen, Clone MIB-1, Agilent (Cat# M7240, RRID:AB_2142367)<br>Western blot:<br>C-terminal DNMT1 antibody, Abcam (Cat#ab92314) (1:1000)<br>GAPHD antibody, Invitrogen Antibodies (Cat#AM4300) (1:1000) |
|---|---|
| Validation | Validation of the antibodies was performed either through indirect validation through published literature (Hickey et al., Nat Medicine, 2021) or from the antibody manufacturers/distributors themselves. |

## Eukaryotic cell lines

Policy information about cell lines

| Cell line source(s) | Parental MCF7 breast cancer cells and endocrine-resistant TAMR and FASR cells were obtained from our collaborator Dr Julia Gee (Cardiff University, UK). |
|---|---|
| Authentication | All cell lines were authenticated by short-tandem repeat profiling (CellBank Australia, Westmead, NSW, Australia) and cultured for <6 months after authentication. |
| Mycoplasma contamination | All cell lines used in-house tested negative for mycoplasma using the MycoAlert Mycoplasma Detection Kit (Lonza, #LT07-318). |
| Commonly misidentified lines (See ICLAC register) | No cell lines from the ICLAC register were used. |

# Animals and other organisms

Policy information about [studies involving animals](); [ARRIVE guidelines]() recommended for reporting animal research

| | |
|---|---|
| Laboratory animals | 6–8-week-old female NOD-scid IL2Rγnull (NSG) mice, obtained from Australian BioResources (Sydney, Australia) were used in the study. Mice were socially housed at the Garvan Institute of Medical Research specific pathogen free (SPF) animal facility, in temperature and light cycle-controlled rooms and given ad lib access to food, water and nesting materials. |
| Wild animals | The study did not involve wild animals. |
| Field-collected samples | The study did not involve samples collected from the field. |
| Ethics oversight | All in vivo experiments, procedures and endpoints were approved by the Garvan Institute of Medical Research Animal Ethics Committee (HREC #14/35, #15/25, ARA #21/11) and performed at the Garvan Institute of Medical Research using standard techniques in accordance with relevant national and international guidelines. |

Note that full information on the approval of the study protocol must also be provided in the manuscript.

# ChIP-seq

## Data deposition

☒ Confirm that both raw and final processed data have been deposited in a public database such as [GEO]().

☒ Confirm that you have deposited or provided access to graph files (e.g. BED files) for the called peaks.

| | |
|---|---|
| Data access links<br>*May remain private before publication.* | GSE171074 and GSE216989 |
| Files in database submission | Both raw (*.fastq.gz) and processed (*.bed and *.bigwig) files are made available for download.<br><br>GSM5218278  Gar15-13 Vehicle 1 ER<br>GSM5218279  Gar15-13 Vehicle 2 ER<br>GSM5218280  Gar15-13 Vehicle 3 ER<br>GSM5218281  Gar15-13 Vehicle 4 ER<br>GSM5218282  Gar15-13 Decitabine 1 ER<br>GSM5218283  Gar15-13 Decitabine 2 ER<br>GSM5218284  Gar15-13 Decitabine 3 ER<br>GSM5218285  Gar15-13 Decitabine 4 ER<br>GSM5218286  Gar15-13 Input ER<br><br>CUT&RUN:<br>GSM7648680  Gar15-13 Vehicle 1 CTCF<br>GSM7648681  Gar15-13 Vehicle 2 CTCF<br>GSM7648682  Gar15-13 Vehicle 3 CTCF<br>GSM7648683  Gar15-13 Decitabine 1 CTCF<br>GSM7648684  Gar15-13 Decitabine 2 CTCF<br>GSM7648685  Gar15-13 Decitabine 3 CTCF<br>GSM7648686  Gar15-13 Vehicle 1 H3K27ac<br>GSM7648687  Gar15-13 Vehicle 2 H3K27ac<br>GSM7648688  Gar15-13 Vehicle 3 H3K27ac<br>GSM7648689  Gar15-13 Decitabine 1 H3K27ac<br>GSM7648690  Gar15-13 Decitabine 2 H3K27ac<br>GSM7648691  Gar15-13 Decitabine 3 H3K27ac<br>GSM7648692  Gar15-13 IgG |
| Genome browser session<br>(e.g. [UCSC]()) | Hi-C and PCHi-C browser files are provided in the GEO submission. These files can be imported directly into JuiceBox and WashU Browser. ChIP-seq and CUT&RUN data generated in this paper is provided in the GEO submission. |

## Methodology

| | |
|---|---|
| Replicates | Decitabine treatment was performed in two independent patient-derived xenografts (PDXs) and 7 to 8 individual PDX mice were used in therapeutic studies to obtain sufficient sample size.<br><br>EPIC DNA methylation, ER ChIP-seq and RNA-seq experiments were performed in quadruplicates and CUT&RUN, Hi-C and Promoter Capture Hi-C were performed in triplicates to assess statistical significance. Public ER ChIP-seq datasets used in this study were performed on multiple primary patient breast tumour samples as described in the respective papers. |
| Sequencing depth | Sequencing depth and summary statistics for all generated sequencing datasets (ChIP-seq, RNA-seq, Hi-C and PCHi-C) are provided in Supplementary Tables 2-5. For ChIP-seq datasets, each sample was sequenced in order to target a read depth of ~20+ million 75bp single-end reads. For CUT&RUN datasets, each sample was sequenced in order to target a read depth of 10 million 150bp paired-end reads. |

| | |
|---|---|
| Antibodies | Antibodies used were: Anti-Estrogen Receptor alpha (HC-20), Abcam (Cat# ab23738; RRID: AB_2104842), CTCF (CTCF CUTANA™ CUT&RUN Antibody (cat. #13-2014)), H3K27ac (Histone H3K27ac Antibody, SNAP-ChIP Certified (cat. #13-0045)) |
| Peak calling parameters | Peaks were called with MACS2 (v2.2.6) (Zhang et al., 2008) under the default parameters (band width = 300, model fold = [5, 50], q value cutoff = 5.00e-02) |
| Data quality | All experiments were performed in multiple replicates. Specifically, ChIP-seq experiments were performed in four replicates for Vehicle and Decitabine-treated tumours in 2 PDX models and CUT&RUN in three replicates. All peaks are below the Macs2 FDR cut off. |
| Software | ChIP-seq and CUT&RUN reads were aligned against human genome (hg38/GRCh38) using bowtie2 with default parameters (--dovtail for CUT&RUN). Non-uniquely mapped, low quality (MAPQ<15) and PCR duplicate reads were removed. Peak calling of individual ChIP–seq and CT&RUN experiments was performed with MACS2 with default parameters. Statistics for each library can be found in Supplementary Table 4. Consensus peaks were identified by intersecting MACS2 peaks obtained from each sample using bedtools intersect (v.2.25.0) with min. overlap > 0.6. Differential binding analyses were performed using DiffBind (v.3.0.9) and DESeq2 (v.1.3.0) with FDR < 5%. Enrichment analyses were performed using GAT, ChIPseeker (v.1.26.0) and normalised to library size. Merged bigwig tracks for visualisation were created from merged bam files from all replicates using the bamCoverage function with scaling factor normalisation and heatmaps and average profiles were plotted with deepTools2. All code used to process and analyze ChIP-seq data is publicly available within the GitHub repository https://github.com/JoannaAch/PDX_Decitabine_3DEpigenome. |

