## [Peer Review File · Nature Structural & Molecular Biology]

Peer Review Information

Manuscript Title: The potential of epigenetic therapy to target the 3D epigenome in endocrine-resistant breast cancer

Corresponding author name(s): Susan Clark, Joanna Achinger-Kawecka

Reviewer Comments & Decisions:

Decision Letter, initial version:
--

Message: Nature Structural & Molecular Biology NSMB-A47134

10th Mar 2023

Dear Prof. Clark,

Thank you for submitting your manuscript, "Epigenetic therapy targets the 3D epigenome in endocrine-resistant breast cancer". I sincerely apologise for the delay while we awaited the comments (copied below) from the 3 reviewers who evaluated your manuscript. Nevertheless, we have now received their reports and have editorially discussed them. Unfortunately, after carefully considering their comments, we cannot offer to publish your manuscript in Nature Structural & Molecular Biology.

You will see that while the referees acknowledge the extent of work and the wealth of data presented, they raise serious concerns which cast doubt on the strength of the conclusions that can be drawn from the study. More specifically, Reviewer# 3 raises significant concerns about the experimental design and the potential (unaccounted for) confounding effect of differential cell growth in the analysis, while Reviewer #2 notes that the extent/analysis of the genomic data lacks depth and is restricted to a small number of data points, while at the same time missing pertinent additional experiments. In light of these serious concerns, as well as the other technical issues raised by the referees, I am afraid that we cannot offer to continue to consider this manuscript for publication in Nature Structural & Molecular Biology.

I am sorry we could not be more positive on this occasion. We hope that the referees' comments will be helpful and useful to you in revising the manuscript for submission elsewhere.

Sincerely,

Dimitris Typas

Associate Editor
Nature Structural & Molecular Biology
ORCID: 0000-0002-8737-1319

Reviewer #1 (Remarks to the Author):

Achinger-Kawecka et al demonstrate that the DNA methyltransferase inhibitor Decitabine inhibits tumour growth in two PDX models of ER+ breast cancer. Using Hi-C and Promoter-capture Hi-C, they go on to show that Decitabine induces large-scale (compartments, TADs) and finer-scale (promoter-enhancer) alterations in the 3D chromatin organisation. They find that DNA hypomethylation in response to Decitabine results in more interactions involving a new set of enhancers. These enhancers are occupied by ER, and the authors suggest that reprogramming of ER binding is involved in establishing these Decitabine-induced interactions. Using TAMR cells in culture, they go on to show that Decitabine-induced interactions are largely lost, and Decitabine-activated genes are largely repressed again upon removal of Decitabine.

This is a very interesting and well-written manuscript with several important findings. The authors comprehensively address the impact of Decitabine on chromatin in ER+ breast cancer using state-of-the-art genomics methods and highly relevant PDX mouse models, and the data and analyses appear to be thorough and robust. This detailed characterisation of the effect of Decitabine on ER+ breast cancer is novel and highly relevant from a mechanistic as well as translational point of view. The authors begin to delineate many of the mechanisms underlying the impact of this drug on breast tumour cells. This includes a suggested link between Decitabine treatment and the ER pathway. This link is highly interesting and clinically relevant but requires slightly more work to become robust. This is really my only major comment to this important piece of work.

Comments

1. The authors propose that Decitabine reprogrammes ER binding, which establishes new chromatin interactions promoting a target gene programme that inhibits tumour growth. However, it is unclear if this new target gene programme is indeed activated by ER and if it is required for the effect of Decitabine. Transcription factors often show a high degree of opportunistic binding with limited functional impact, so reprogramming of ER to these new enhancers does not necessarily mean that ER is important for establishing these new interactions and for regulating this Decitabine-induced gene programme. This issue becomes particularly relevant given that Decitabine would likely be given together with endocrine therapy in a clinical setting, which would inhibit ER function. So the question is: Does endocrine therapy negate some of the effects of Decitabine or do these treatments have additive effects? I would suggest investigating this in the TAMR model by doing combination treatment of Decitabine and fulvestrant, which TAMR cells usually still respond to. This would allow the authors to investigate if the Decitabine-induced interactions and target gene programme depend on ER activity and determine if the combination treatment provides a greater or smaller effect on cell proliferation compared to either treatment alone.

2. Line 184-190: Here, the authors compare interactions between A and B compartments in each treatment-group. They conclude that contacts between A compartments increase in Decitabine-treated tumours, although this is not significant. This statement should be

revised to take this into account. In addition, it would be informative to also determine if compartments activated by Decitabine (B to A switching) interact more with other A compartments in Decitabine-treated compared to veh-treated tumours. Put another way, do these B-to-A-switching compartments interact with other A compartments in Decitabine-treated tumours to the same extent as compartments that are active in both treatments?

3. Fig. 4: Here the authors nicely show that ER binding is reprogrammed in response to Decitabine treatment. To solidify this conclusion, it would be worth validating that ER (and FOXA1) expression is not changing in response to this treatment. Ideally, this should be done by IHC, but since Decitabine is an epigenetic drug regulating transcription, merely checking the expression of these transcription factors in the RNA-seq would suffice.

4. Fig. 6: In panel h, the authors show that most of the gained enhancer OEs in Day 7 Decitabine are not gained in Decitabine recovery. This comparison is done in a binary manner using venn diagrams, so the quantitative nature of this recovery is lost. It would help to visualise these dynamic interactions in a quantitative manner to help assess the level of recovery, e.g., is the signal at gained enhancer OEs in Day 7 Decitabine completely lost in Decitabine recovery for the "recovered interactions" and is the signal maybe still going down in some of the "non-recovered interactions" even though the interaction is still significant? Such quantitative visualisation of these dynamic interactions, where Day 7 Decitabine and Decitabine recovery are also directly compared would be helpful. This would make the analysis go beyond seeing interactions as yes/no based on arbitrary thresholds and give a more nuanced view of the tendency of genomic regions to interact. Finally, "Recovered interactions" is not a very intuitive name as these are interactions established in Day 7 Decitabine that are lost again in Decitabine recovery. This naming implies that interactions are lost and then re-established in the recovery samples.

Reviewer #2 (Remarks to the Author):

In this report Kawecka et. al. generated two metastatic ER+ breast tumor xenograft to study the effect of an FDA improved drug in the progression of these tumors. The authors show that Decitabine decrease the tumor growth by reducing the genome wide DNA methylation levels, a well-described consequence of endocrine-resistant tumors. They show that the Decitabine-mediated hypomethylation results in an overall change in the 3D genome folding. Namely, they performed Hi-C and observed a switch between the A/B compartments mostly from the inactive B to the active A and they report that these changes are correlated with decreased levels of methylation in the Decitabine-treated group. They also show that this switch results in mild gene expression changes. The switch in the compartments, in the TADs and in gene expression encouraged them to perform promoter Capture-C to investigate the connections between promoter and enhancers. They claim that new connections are achieved between promoters and enhancers as a result of the Decitabine-mediated hypomethylation and they show that these new connections resulted in increased expression of genes that contribute to the decline of the tumor growth. They also report that these connections are facilitated by increased binding of ER which is known to be affected by the DNA methylation levels. Finally, the authors are replicating a subset of their experiments in an established endocrine-resistant cell line where they also perform a recovery experiment that shows

that removal of Decitabine and culturing of these cells restores the DNA methylation levels.

A major concern with the manuscript relates to the following. The authors did a large amount of work, they generated a diverse set of high-throughput data and they clearly wish to draw very general (and seemingly important) conclusions. However, as detailed below, analyses are often too superficial, data shown or highlighted are too selective and necessary validation experiments are too often lacking to justify this. The authors also don't guide the reader in specifying what is novel versus fully expected and/or shown previously. After all: we all know that DNA demethylation can reduce tumor growth, can change the regulatory landscape, alters ER binding, can change transcription of selected genes and we all understand that this in turn can result in (or be accompanied by) changes in genome topology. See for example their own recent study by Du et al., Cell Reports 2021, but also, as examples: Wang et al., Sci. Adv. 2018; Zhou et al, Nat Comm. 2019 ; Yin et al. Science 2017; Achinger-Kawecka et al., Nat Comm. 2020). So, their title Epigenetic therapy (DNA demethylation) is fully expected to also have impact on/target the 3D epigenome, not only in endocrine-1 resistant breast cancer but in any cell type. What else do we really learn beyond this, other than having a better description of one or two selected PDX models? Every cell population, no matter how heterogeneous (like tumors) has an average 3D genome, but how important is this really for understanding tumor biology and finding treatments (unless the tumor is driven by a mutated architectural protein)?

Figure 1 gives the impression that two PDX models were used for HiC, PchIC, methylome, RNA-seq and CHIP-seq analyses, but I don't think this is the case. Please be clear in this figure and in the text what was done in what PDX model. Also, please mention/show clearly at what stage (how many days of vehicle and Decitabine treatment) each experiment was performed.

HiC. Please clarify in the main text what number and what percentage of TADs is scored as 'differential'. And specify what this means across replicates: are 'differential TADs' consistently called across the three vehicle replicates, and consistently lost across the three Decitabine-treated replicates? This information is needed to judge (modify?) the validity of conclusions drawn around line 210.

What is the relationship between the differential TADs, the DEGs, DMRs, the rewired E-P population?

The authors suggest that Decitabine results in compartment switching and they highlight the B to A compartments switching and the accompanying upregulation of 87 genes but the authors do not comment on these genes.: are these genes related to (can they explain) the effect of Decitabine on the tumor? Also, are upregulated genes enriched in B->A compartments?

PC-HiC: Similarly, please better explain the PC-HiC data and results:

- Please also specify the average number of unique informative reads per promoter for each replicate experiment. Exclude reads that analyze interchromosomal ligations, far-cis ligations (e.g. >5Mb), promoter-promoter (bait-bait) ligations and the very local ligations ((e.g. <2 kb), as these are non-informative. Please specify these numbers per replicate.
- The authors score contacts that are conserved and differential between conditions (vehicle versus Decitabine-treated). Presumably they have combined the replicates per condition to come to a consensus set of vehicle and Decitabine interactions? Please clarify in the text.

Line 238: "In total, we found 238 13,088 stable (no change) and 4,111 dynamic (gained or lost) contacts for promoters and 55,186 stable and 26,912 dynamic contacts for enhancer OEs (Fig. 3c)." To interpret such results we need to better understand the reproducibility of PC-HiC data. What percentage of scored promoter interactions is

reproduced between the replicate experiments, within and between conditions (vehicle and Decitabine-treated)? Please provide this analysis. In our experience, PC-HiC reproducibility is often disappointingly low. If true here too, the rewiring could well be (partially) explained by the poor reliability of scored interactions. Authors please provide this information, discuss reproducibility of results and explain readers how this impacts data interpretation.

- Related: CHiCAGO is known to call a high average number of contacts per promoter and many (the majority) of these contacts will not be-functional. Please specify numbers further: how many contacts are scored in each condition for inactive genes, for non-responding active genes, for upregulated genes and for downregulated genes?

RNA-seq: Again, show reproducibility between replicates within and between conditions. How many genes are found upregulated and downregulated in Decitabine-treated versus vehicle tumors? What number (and percentage) may simply be explained by differential DNA methylation at the promoter? What number (and percentage) of the up- and downregulated genes is found to have increased/decreased numbers of contacting partners? Do the DEGs explain the action of decitabine and the decrease of the tumor growth?

The author perform Gene set enrichment analysis (GSEA) of all differentially expressed genes. Did they do this for the total collection of up- and down-regulated genes? If so, please repeat for each category separately.

Enhancers. A critical point missing is the characterization of enhancers. The 3D genome (rewiring) is presented as being important for cancer biology, but rewiring is likely the consequence of other epigenetic modifications induced by Decitabine. Understanding the (differences in) enhancer landscapes of vehicle and Decitabine treated samples is crucial for the interpretation of results. H3K27Ac ChIP-seq needs to be performed in multiple vehicle and treated PDX samples, to find the conserved, newly formed and lost enhancers upon Decitabine treatment. Again, authors should show reproducibility between replicates (see above).

As said, it does not suffice to use ChromHMM data to assume locations of active promoter, enhancers, poised enhancer etc. Also, authors mention that this analysis is using data from downloaded from GEO (GSE73783) for tamoxifen-resistant (TAMR) MCF7 cells, but the accession number they provide appears to refer to a study that does not contain this cell line and does not contain H3K27ac and H3K27me3?

Once identified by ChIP-seq, one wants to know the relationship between differential DNA methylation, differential enhancers and differential promoter contacts. What number/percentage of conserved/gained/lost enhancers coincides with conserved/gained/lost methylated regions? What number/percentage of conserved/gained/lost contacts coincides with conserved/gained/lost enhancers? Is 3D rewiring by Decitabine the consequence of a new enhancer landscape created by altered DNA methylation patterns?

The authors decide to perform DNA motif enrichment analysis on the gained 'enhancer' contacts. Irrespective of their involvement in promoter contacts, Decitabine treatment results in a much larger set of differentially methylated regions (DMRs), and probably in the creation and loss of enhancers (that need to be identified by ChIP-seq; see above). Do the authors find identical motif enrichments in the lost methylated regions, and in the gained enhancers? If not, what is their explanation?

The most significantly enriched binding motif in the gained enhancer OEs is that of CTCF. Although the authors (line 494-502) like to conclude that this is in agreement with literature and their other data, they don't discuss that their most relevant observation related to CTCF's function, namely that contact insulation decreases and TADs dissolve in Decitabine treated samples, is highly unexpected if CTCF binds better to DNA in Decitabine

treated cells. The authors must have realized this, it is somewhat disturbing that they decided to not mention this. How do the authors explain this discrepancy? They need to perform CTCF ChIP-seq experiments to understand the relationship between reduced insulation and TAD destabilization, versus altered (probably increased) CTCF recruitment. Line 355: "For example, at the SPATA18 locus, multiple 3D enhancer-promoter interactions are gained with Decitabine treatment, concomitant with gain in ER binding at putative enhancer, loss of DNA methylation and activation of the ER target gene (Fig. 5d and Extended Data Fig. 5a)." Authors, replace "activation of the ER target gene" by "a 1.5 fold upregulation of the ER target gene". Upregulation is really not impressive, the gene is highly active already in vehicle treated tumors, so this wording is needed to be more precise to put things in perspective.

Line 366: "These results reveal a link between Decitabine-induced DNA hypomethylation, rewiring of ER-bound enhancer-promoter interactions and alteration in the ER transcriptional program." See concern expressed above: 'these results' refer to an inspection of 3 genes. The reader cannot judge whether these are cherry picked, and such general conclusion cannot be drawn without careful systematic analyses. Authors should also comment on the levels of upregulation, and distinguish newly activated genes from genes further increasing their expression.

The recovery experiments are interesting, but unfortunately done in the MCF7 cell line instead of PDX models, which somewhat disconnects this part from the rest of the study. Other points:

- 1) logFC to score the DEG are different in figure 4b and in 7a. is there a reason for this?
- 2) Figure 2a: the extended figure 2b is more informative about the compartment switching and thus it should be exchanged with the main figure 2a.
- 3) There is not a western blot for DNMT1 in the figure 1 to show the effect of Decitabine. Authors perform indeed a western blot in the MCF7 cells.
- 4) The cartoon in figure 8 contains too many untested assumptions (see comments above).

Reviewer #3 (Remarks to the Author):

In this paper the authors explore the effects of DNA hypomethylation on the 3D organization of the genome, ER chromatin binding and gene expression programs in endocrine resistant breast cancer patient derived xenografts and cell lines. They show that the DNA methyltransferase inhibitor decitabine inhibits the growth in mice of two ER+ human PDX models including both the Gar15-13 ER+/HER2- PDX and the HCI-005 PDX which is ER+/HER2+ and harbors a constitutively active ER mutation. They show that following 40 days of decitabine treatment that the levels of DNA methylation are significantly reduced, especially in regions of the genome that are active ER bound enhancers leading to increased ER binding at these enhancers, increased enhancer-promoter looping and gene expression. In addition, the authors perform a time-course study of DNA hypomethylation for 7 days followed by 28 days of re-methylation in the tamoxifen resistant MCF7 TAMR cell line model. The results of this experiment largely confirm the findings in the PDX models.

While the epigenomic and chromatin architecture studies are interesting, there is an important problem with the overall conclusion of the paper that the differences between control and decitabine treated PDX represents epigenetic reprogramming. As the authors show, there is a very significant difference in the growth of the decitabine treated tumors. Thus, the cell populations present in the two conditions are likely very different and thus

the epigenomic differences may be the result rather than the cause of the growth differences. In addition, while the cell line studies demonstrate similar enhancer hypomethylation at active ER enhancers following 7 days of treatment and largely remethylation following removal of decitabine for 28 days, no data on the effects of decitabine on growth in the cell line model are provided.

Other concerns:

- 1) The level of ER expression in the tumors following decitabine treatment needs to be shown as increased ER levels may play a role in the expanded ER chromatin binding. This also needs to be shown for the cell line studies.
- 2) The model figure suggests that specific tumor suppressors are induced following DNA hypomethylation and are responsible for the growth suppression caused by decitabine. None of the studies in the manuscript directly address this hypothesis.

Author Rebuttal to Initial comments

Manuscript Nature Structural & Molecular Biology NSMB-A47134

Responses to referees for revised manuscript NSMB-A47134, “Epigenetic therapy targets the 3D epigenome in endocrine-resistant ER+ breast cancer” by Achinger-Kawecka et al.

We thank all the Reviewers for their time and diligence in consideration of our manuscript, and for their contribution to its improvement. In response to their suggestions, we have now performed additional experiments and analyses to support the conclusions of the study, address important additional questions, and improve clarity.

Reviewer #1 (Remarks to the Author):

Achinger-Kawecka et al demonstrate that the DNA methyltransferase inhibitor Decitabine inhibits tumour growth in two PDX models of ER+ breast cancer. Using Hi-C and Promoter-capture Hi-C, they go on to show that Decitabine induces large-scale (compartments, TADs) and finer-scale (promoter-enhancer) alterations in the 3D chromatin organisation. They find that DNA hypomethylation in response to Decitabine results in more interactions involving a new set of enhancers. These enhancers are occupied by ER, and the authors suggest that reprogramming of ER binding is involved in establishing these Decitabine-induced interactions. Using TAMR cells in culture, they go on to show that Decitabine-induced interactions are largely lost, and Decitabine-activated genes are largely repressed again upon removal of Decitabine.

This is a very interesting and well-written manuscript with several important findings. The authors comprehensively address the impact of Decitabine on chromatin in ER+ breast cancer using state-of-the-art genomics methods and highly relevant PDX mouse models, and the data and analyses appear to be thorough and robust. This detailed characterisation of the effect of Decitabine on ER+

breast cancer is novel and highly relevant from a mechanistic as well as translational point of view. The authors begin to delineate many of the mechanisms underlying the impact of this drug on breast tumour cells. This includes a suggested link between Decitabine treatment and the ER pathway. This link is highly interesting and clinically relevant but requires slightly more work to become robust. This is really my only major comment to this important piece of work.

Response: We thank Reviewer #1 for their kind words and their insights, which we believe have made significant improvements to the manuscript. In response to their thoughtful suggestions, we have conducted additional experiments and analyses detailed below, and clarified the text of the manuscript.

1. The authors propose that Decitabine reprogrammes ER binding, which establishes new chromatin interactions promoting a target gene programme that inhibits tumour growth. However, it is unclear if this new target gene programme is indeed activated by ER and if it is required for the effect of Decitabine. Transcription factors often show a high degree of opportunistic binding with limited functional impact, so reprogramming of ER to these new enhancers does not necessarily mean that ER is important for establishing these new interactions and for regulating this Decitabine-induced gene programme. This issue becomes particularly relevant given that Decitabine would likely be given together with endocrine therapy in a clinical setting, which would inhibit ER function. So the question is: Does endocrine therapy negate some of the effects of Decitabine or do these treatments have additive effects? I would suggest investigating this in the TAMR model by doing combination treatment of Decitabine and fulvestrant, which TAMR cells usually still respond to. This would allow the authors to investigate if the Decitabine-induced interactions and target gene programme depend on ER activity and determine if the combination treatment provides a greater or smaller effect on cell proliferation compared to either treatment alone.

Response: We have tested the effect of combination treatment (Decitabine and endocrine therapies Tamoxifen (**Rebuttal Fig. 1a**) and Fulvestrant (**Rebuttal Fig. 1b**)) on tumour growth in the PDX model (Gar15-13). We observed that combination treatment also has a significant growth inhibitory effect which shows that endocrine therapy does not negate the effects of Decitabine but potentially provides an additive effect on inhibition of cell growth (**Rebuttal Fig. 1a-b**). We have not included this experiment in the paper as it is part of our ongoing studies on combination therapies.

Endocrine treatment inhibits ER binding to the DNA in endocrine sensitive cells, but it does not inhibit ER binding in resistant cells. This has been previously demonstrated, with many studies (e.g. [1-3] showing large reprogramming of ER binding in metastatic endocrine resistant breast cancer tissues and cell lines; also reviewed in [4]). As our PDX models are endocrine resistant, our experiments combining endocrine therapy with Decitabine are not able to directly address if the

Decitabine-induced interactions and target gene programme depend on ER activity. We agree this is an interesting question but beyond the scope of the current paper.

Rebuttal Figure 1

2. Line 184-190: Here, the authors compare interactions between A and B compartments in each treatment-group. They conclude that contacts between A compartments increase in Decitabine-treated tumours, although this is not significant. This statement should be revised to take this into account. In addition, it would be informative to also determine if compartments activated by Decitabine (B to A switching) interact more with other A compartments in Decitabine-treated compared to veh-treated tumours. Put another way, do these B-to-A-switching compartments interact with other A compartments in Decitabine-treated tumours to the same extent as compartments that are active in both treatments?

Response: We have now revised the statement as suggested by Reviewer #1. **Line 207-211** now reads: “(...) we quantified A–A and B–B interaction frequencies in Decitabine and Vehicle-treated tumours and found significantly decreased interaction strength between closed compartments (B–B interactions; $P = 0.025$, two-tailed Students *t*-test), no change in contacts between active compartments (A–A interactions; $P = 0.26$, two-tailed Students *t*-test) and increased contacts between A–B compartments ($P = 0.011$, two-tailed Students *t*-test) (Fig. 2d, e and Extended Data Fig. 2e, f)”. Additionally, to test if new A compartments interact with other A compartments more than expected by chance for all A compartments, we performed observed / expected analyses for gained A-interactions (that are driving the switch from B-to-A) and stable A compartments. We found that indeed gained interactions are significantly enriched for other A compartments, confirming that B-to-A regions interact more with other A compartments as compared to stable A compartments. These results have now been added to the Results section (**lines 212-214**): “Gained A-compartment interactions were significantly enriched for stable A compartments ($P < 0.001$, $O/E = 1.7$), suggesting increased interactivity between new A compartments and stable A compartments”.

3. Fig. 4: Here the authors nicely show that ER binding is reprogrammed in response to Decitabine treatment. To solidify this conclusion, it would be worth validating that ER (and FOXA1) expression is not changing in response to this treatment. Ideally, this should be done by IHC, but since Decitabine is an epigenetic drug regulating transcription, merely checking the expression of these transcription factors in the RNA-seq would suffice.

Response: We did not observe a significant change in ESR1 and FOXA1 gene expression upon Decitabine treatment (see Rebuttal Fig. 2). The expression data for ESR1 was already included in the Supplementary Table 7 (DEG analyses). Additionally, we did not observe a significant change in ER IHC staining between Vehicle- and Decitabine-treated tumours but we did find a small but significant reduction in nuclear ER staining with Decitabine treatment (**new Extended Data Fig. 1b**) and we have now added this to the Results section (**lines: 120-123**): “Immunohistological quantification of the Estrogen Receptor at end-point showed no significant change in the proportion on ER positive cells however there was a small but significant reduction in nuclear ER staining with Decitabine treatment (**Extended Data Fig. 1b**).”

4. Fig. 6: In panel h, the authors show that most of the gained enhancer OEs in Day 7 Decitabine are not gained in Decitabine recovery. This comparison is done in a binary manner using venn diagrams, so the quantitative nature of this recovery is lost. It would help to visualise these dynamic interactions in a quantitative manner to help assess the level of recovery, e.g., is the signal at gained enhancer OEs in Day 7 Decitabine completely lost in Decitabine recovery for the “recovered interactions” and is the signal maybe still going down in some of the “non-recovered interactions” even though the interaction is still significant? Such quantitative visualisation of these dynamic interactions, where Day 7 Decitabine and Decitabine recovery are also directly compared would be helpful. This would make the analysis go beyond seeing interactions as yes/no based on arbitrary thresholds and give a more nuanced view of the tendency of genomic regions to interact. Finally, “Recovered interactions” is not a very intuitive name as these are interactions established in Day 7 Decitabine that are lost again in Decitabine recovery. This naming implies that interactions are lost and then re-established in the recovery samples.

Response: First, we agree with Reviewer #1 that on reflection the naming we used for “Recovered interactions is not intuitive. Therefore, to improve clarity of our results we have now renamed the two classes of identified altered interactions, into: “gained & lost” and “gained & maintained” interactions. We have now added a **new Figure 6h** schematic to visualise and better interpret these two classes of gained chromatin interactions.

Second, to quantitatively represent the level of “recovery” for gained OE interactions, we calculated ChICAGO scores for “Gained & Maintained” interactions in Day 7 Decitabine and Day 28 Decitabine “Recovery” samples and plotted them on a scatter plot. This new plot has now been added as **new Extended Data Fig. 6d**. “Gained & Maintained” OE interactions have a similar average ChICAGO score in Day 7 Decitabine (mean = 6.73, median = 7.34) and in Day 28 Decitabine “Recovery” samples (mean = 6.12, median = 5.64), with some decrease in the Day 28 Decitabine “Recovery” samples observed based on regression line. These results have now been added to the Results section (**lines: 462-467**): *“Importantly, the majority of Day 7 Decitabine gained OE enhancers (64,044) interactions were lost in Decitabine Recovery samples (47,007 OE enhancers “Gained & Lost”) (Fig. 6i), while “Gained & Maintained” interactions showed decreasing ChICAGO significance scores, suggesting some reduction in interaction strength after 28 days of DNA methylation recovery (Extended Data Fig. 6d).”*

Reviewer #2 (Remarks to the Author):

In this report Kawecka et. al. generated two metastatic ER+ breast tumor xenograft to study the effect of an FDA improved drug in the progression of these tumors. The authors show that Decitabine decrease the tumor growth by reducing the genome wide DNA methylation levels, a well-described consequence of endocrine-resistant tumors. They show that the Decitabine-mediated hypomethylation results in an overall change in the 3D genome folding. Namely, they performed Hi-C and observed a switch between the A/B compartments mostly from the inactive B to the active A and they report that these changes are correlated with decreased levels of methylation in the Decitabine-treated group. They also show that this switch results in mild gene expression changes. The switch in the compartments, in the TADs and in gene expression encouraged them to perform promoter Capture-C to investigate the connections between promoter and enhancers. They claim that new connections are achieved between promoters and enhancers as a result of the Decitabine-mediated hypomethylation and they show that these new connections resulted in increased expression of genes that contribute to the decline of the tumor growth. They also report that these connections are facilitated by increased binding of ER which is known to be affected by the DNA methylation levels. Finally, the authors are replicating a subset of their experiments in an established endocrine-resistant cell line where they also perform a recovery experiment that shows that removal of Decitabine and culturing of these cells restores the DNA methylation levels.

A major concern with the manuscript relates to the following. The authors did a large amount of work, they generated a diverse set of high-throughput data and they clearly wish to draw very general (and seemingly important) conclusions. However, as detailed below, analyses are often too superficial, data shown or highlighted are too selective and necessary validation experiments are too often lacking to justify this.

Response: We thank Reviewer #2 for their detailed review of our manuscript and for bringing forward experiments and analyses.

The authors also don't guide the reader in specifying what is novel versus fully expected and/or shown previously. After all: we all know that DNA demethylation can reduce tumor growth, can change the regulatory landscape, alters ER binding, can change transcription of selected genes and we all understand that this in turn can result in (or be accompanied by) changes in genome topology. See for example their own recent study by Du et al., Cell Reports 2021, but also, as examples: Wang et al., Sci. Adv. 2018; Zhou et al, Nat Comm. 2019 ; Yin et al. Science 2017; Achinger-Kawecka et al., Nat Comm. 2020). So, their title Epigenetic therapy (DNA demethylation) is fully expected to also have impact on/target the 3D epigenome, not only in endocrine-1 resistant breast cancer but in any cell type. What else do we really learn beyond this, other than having a better description of one or two selected PDX models? Every cell population, no matter how heterogeneous (like tumors) has an average 3D genome, but how important is this really for understanding tumor biology and finding treatments (unless the tumor is driven by a mutated architectural protein)?

Response: We have modified our discussion to enhance the novelty of findings. Specifically, we included additional details on how our discoveries add new information to previous studies (see Discussion **lines: 551-568**). The main novel findings from our work are:

- We show for the first time that epigenetic therapy (Decitabine), suppresses tumour growth in **metastatic ER+ endocrine-resistant breast cancer**,
- Epigenetic therapy induces genome-wide DNA hypomethylation and subsequent de-compaction of chromatin and altered TAD boundary insulation,
- Specifically, we show *in vivo* and *in vitro* direct evidence that epigenetic therapy-induced DNA hypomethylation at ER-enhancer elements causes **(1) gain in ER binding, (2) enhancer activation** and **(3) rewiring of long-range ER-mediated enhancer-promoter interactions**. Altered ER-mediated enhancer-promoter interactions **(4) re-activate specific ER target genes**, and
- **Mechanistic evidence** from temporal *in vitro* study demonstrates that long-term withdrawal of epigenetic therapy partially **(1) restores methylation at ER-enhancer** elements, resulting in **(2) loss of ectopic 3D enhancer-promoter interactions** and associated **(3) gene repression**.

Figure 1 gives the impression that two PDX models were used for HiC, PcHiC, methylome, RNA-seq and CHIP-seq analyses, but I don't think this is the case. Please be clear in this figure and in the text what was done in what PDX model. Also, please mention/show clearly at what stage (how many days of vehicle and Decitabine treatment) each experiment was performed.

Response: Fig. 1 includes a summary of assays performed with n indicating number of samples analysed and how many days of vehicle and Decitabine treatment. We have now further modified this section of Figure 1a to include additional information on the specific PDX model that was used for each assay listed, rather than a total number of samples performed (**new Figure 1a**). Third sentence in the Hi-C section (**lines 182-184**) already stated “*We analysed in situ Hi-C data (...) corresponding to three biological replicates of Vehicle and Decitabine-treated tumours in Gar15-13 PDX.*” Information on the number of days post Decitabine treatment that the experiments were performed was already included in the Results section (**lines 115-116**): “*Treatment continued with twice-weekly measurement of tumour volume for 35 days or until tumour volume exceeded 1000 mm³. At endpoint mice were sacrificed and tumour material collected for analysis.*”

HiC. Please clarify in the main text what number and what percentage of TADs is scored as ‘differential’. And specify what this means across replicates: are ‘differential TADs’ consistently called across the three vehicle replicates, and consistently lost across the three Decitabine-treated replicates? This information is needed to judge (modify?) the validity of conclusions drawn around line 210.

Response: Information on identification of differential TADs was provided already in the Methods section but we have now expanded this to further clarify how replicates were used (**lines 821-824**): “*Boundaries that were found overlapping by at least 1 genomic bin between replicates were merged. Boundaries separated by at least one genomic bin were considered different between datasets (i.e. consistently lost or gained across all replicates).*”. Figure 2h already includes the number of common and differential TAD boundaries and **lines 225-227** already include information on percentage of differential TAD boundaries: “*Analysis of differential TAD boundaries revealed a large percentage (43.2%) of Vehicle-specific boundaries, which were lost in Decitabine tumours (Fig. 2h)*”

What is the relationship between the differential TADs, the DEGs, DMRs, the rewired E-P population?

Response: We have now performed the additional analyses as requested by Reviewer #2. Namely, we analysed the association between differential TADs (“Overlapping TADs”, “Decitabine-specific TADs” and “Vehicle-specific TADs” identified as above) and (1) differential interactions (**Rebuttal Fig. 3a**), differentially methylated regions (DMRs) (**Rebuttal Fig. 3b**) and differentially expressed genes (DEG) (**Rebuttal Fig. 3c**) by performing observed/expected analyses using permutation test ($n = 1000$). However, none of the observed associations were statistically significant.

Although TADs were first proposed to serve as regulatory units for controlling gene expression by promoting and constraining long-range enhancer-promoter interactions (e.g. Schoenfelder and Fraser 2019 [5]), recent work re-examined the relationship between gene regulation and TADs by observing that disruption of TAD features can alter expression for only a small number of genes (Despang et al. 2019 [6]; Ghavi-Helm et al. 2019 [7], reviewed in [8]). Therefore, these analyses align with current literature, and do not add any novel information to the manuscript to warrant inclusion.

Rebuttal Figure 3

The authors suggest that Decitabine results in compartment switching and they highlight the B to A compartments switching and the accompanying upregulation of 87 genes but the authors do not comment on these genes.: are these genes related to (can they explain) the effect of Decitabine on the tumor? Also, are upregulated genes enriched in B->A compartments?

Response: Gene names for upregulated genes are already included in the Extended Data Fig. 2c. Upregulated genes are indeed enriched at B→A compartment switches (2.2-fold enrichment, P value < 0.001). This data has been now added to the paper (**lines 205-207**): “(...) and upregulated

genes were significantly enriched at B to A switching compartments (2.2-fold O/E, P value < 0.001). Additionally, we have now added **new Supplementary Table 8**, which includes information on expression of all genes present at B to A switching compartments. While some of the upregulated genes that are located at B→A compartments have been previously shown to play a role in breast cancer treatment resistance (e.g. *GDNF* gene [9] shown on the IGV screenshot in Extended Data Fig. 2d), we did not find any significant gene set enrichment terms for these gene sets.

PC-HiC: Similarly, please better explain the PC-HiC data and results:

- Please also specify the average number of unique informative reads per promoter for each replicate experiment.

Response: All QC read summary statistics per replicate and for pooled data (used for WashU visualisations only) are already provided in Supplementary Table 9. This includes total reads per replicate, valid reads, unique reads, cis long- and short-range reads and “on target” reads (i.e. reads that overlap a bait promoter – as already described in Methods, **lines: 849-852**).

Exclude reads that analyze interchromosomal ligations, far-cis ligations (e.g. $>5\text{Mb}$), promoter-promoter (bait-bait) ligations and the very local ligations ((e.g. $<2\text{ kb}$), as these are non-informative. Please specify these numbers per replicate.

Response: These non-informative interactions as described by Reviewer #2 have already been excluded in data pre-processing steps (see Methods, **lines: 849-852**) and percentages per replicate are already provided in Supplementary Table 9. Final ChICAGO called interactions include steps to correct for such artefacts in the data (see [10]).

The authors score contacts that are conserved and differential between conditions (vehicle versus Decitabine-treated). Presumably they have combined the replicates per condition to come to a consensus set of vehicle and Decitabine interactions? Please clarify in the text.

Response: No, the replicates were not combined for the differential interactions analyses. These analyses were performed using statistical software specifically designed to analyse Capture Hi-C data, which is based on DESeq2 – Chicdiff. Replicates are used to measure variation within each condition and are essential for any differential analyses. This information is already included in Results (**lines 274-276**: “In order to directly identify differential promoter-anchored interactions, we integrated the results generated using Chicdiff pipeline with methods to intersect the promoter bait and enhancer OE regions for each interaction (see Methods).” with additional details already provided in the Supplementary Note.

Line 238: “In total, we found 238 13,088 stable (no change) and 4,111 dynamic (gained or lost) contacts for promoters and 55,186 stable and 26,912 dynamic contacts for enhancer OEs (Fig. 3c).” To interpret such results we need to better understand the reproducibility of PC-HiC data.

What percentage of scored promoter interactions is reproduced between the replicate experiments, within and between conditions (vehicle and Decitabine-treated)? Please provide this analysis. In our experience, PC-HiC reproducibility is often disappointingly low. If true here too, the rewiring could well be (partially) explained by the poor reliability of scored interactions. Authors please provide this information, discuss reproducibility of results and explain readers how this impacts data interpretation.

Response: We did not observe any issues with the reproducibility of our PCHi-C data. We have already provided the PCA plot showing correlation between replicates of the PCHi-C data in TAMR cells (Extended Data Fig. 6c) and genome-wide Hi-C data (Supplementary Note; Supplementary Fig. 3a). We have now added PCA plot for PCHi-C data in Gar15-13 PDX to the Supplementary Note (**new Supplementary Fig. 3b**). Additionally, we have now added information on the overlap between interactions identified in each replicate, showing that >84% of ChiCAGO identified interactions are common across all replicates in both Decitabine and Vehicle treated tumours (**new Supplementary Fig. 5c and d** and **Supplementary Note lines 177-181**). Therefore, our presented PCHi-C data is highly reproducible and there is no impact of reproducibility on the interpretation of our results.

- Related: ChiCAGO is known to call a high average number of contacts per promoter and many (the majority) of these contacts will not be-functional. Please specify numbers further: how many contacts are scored in each condition for inactive genes, for non-responding active genes, for upregulated genes and for downregulated genes?

Response: ChiCAGO has been developed specifically for Capture Hi-C analysis and is the gold-standard for Capture Hi-C interaction calling (360 citations). While not all contacts detected by ChiCAGO will be directly related to change in gene expression, all of these contacts can be potentially “functional” by bringing different regulatory elements into close proximity to gene promoters. We have already performed analyses to show how many of the differential interactions are related to change in gene expression and found that up-regulated genes were enriched at differential interactions (Results, **lines 340 – 341** and Figure 4b and 5c). We have now modified Supplementary Table 10 to include information on gene expression change (UP, DOWN and NC in Gar15-13 Decitabine vs. Vehicle) for all genes involved in differential interactions (**new Supplementary Table 10**).

RNA-seq: Again, show reproducibility between replicates within and between conditions. How many genes are found upregulated and downregulated in Decitabine-treated versus vehicle tumors? What number (and percentage) may simply be explained by differential DNA methylation at the promoter? What number (and percentage) of the up- and downregulated genes is found to have increased/decreased numbers of contacting partners? Do the DEGs explain the action of decitabine and the decrease of the tumor growth?

Response: PCA plots showing reproducibility of the RNA-seq data between replicates and conditions are already provided in the Supplementary Note (Supplementary Fig. 3e) and QC summary statistics are already provided in Supplementary Table 6. We did not observe any significant change in DNA methylation at gene promoters (Extended data Fig. 1k and 1l). We have already provided a table with expression of genes present at differential interactions (Supplementary Table 10), which already includes information on expression of genes present at differential promoter interactions. As per main Figure 4a, differentially expressed genes were enriched for hallmarks related to cell cycle and proliferation, which is consistent with the tumour suppressing effect of Decitabine. Additionally, we found significant enrichment for pathways related to viral mimicry response, as already discussed in the Supplementary Note (Supplementary Fig. 2g-h). At 5% FDR and 1.0 logFC cut-off, we found 116 genes were up-regulated and 94 were down-regulated (Supplementary Table 10).

The author perform Gene set enrichment analysis (GSEA) of all differentially expressed genes. Did they do this for the total collection of up- and down-regulated genes? If so, please repeat for each category separately.

Response: GSEA analyses are performed on pre-ranked list of all expressed genes. Performing GSEA on a subset of genes will be an incorrect use of GSEA. GSEA is based on global trends of gene ranking metrics i.e. if many genes from the same pathway tend to be modestly but not significantly upregulated then their cumulative effect might still cause a biologically-meaningful effect. Different tools can be used on DEG subsets of genes e.g. Goseq, but are less informative as they depend on arbitrary cut-off of “significance” and ignore genes that fall just below the cut-off (see original GSEA paper: [11] and further discussion of “gold standard” pathway enrichment analyses in [12]). Thus, GSEA is the most optimal tool to be used for our study design.

Enhancers. A critical point missing is the characterization of enhancers. The 3D genome (rewiring) is presented as being important for cancer biology, but rewiring is likely the consequence of other epigenetic modifications induced by Decitabine. Understanding the (differences in) enhancer landscapes of vehicle and Decitabine treated samples is crucial for the interpretation of results. H3K27Ac ChIP-seq needs to be performed in multiple vehicle and treated PDX samples, to find the conserved, newly formed and lost enhancers upon Decitabine treatment. Again, authors should show reproducibility between replicates (see above). As said, it does not suffice to use ChromHMM data to assume locations of active promoter, enhancers, poised enhancer etc. Also, authors mention that this analysis is using data from downloaded from GEO (GSE73783) for tamoxifen-resistant (TAMR) MCF7 cells, but the accession number they provide appears to refer to a study that does not contain this cell line and does not contain H3K27ac and H3K27me3? Once identified by ChIP-seq, one wants to know the relationship between differential DNA methylation, differential enhancers and differential promoter contacts. What number/percentage of conserved/gained/lost enhancers coincides with conserved/gained/lost methylated regions? What number/percentage of conserved/gained/lost contacts coincides with conserved/gained/lost

enhancers? Is 3D rewiring by Decitabine the consequence of a new enhancer landscape created by altered DNA methylation patterns?

Response: Firstly, we thank Reviewer#2 for picking up this mistake, the correct accession number is GSE118716 and it includes chromHMM tracks and ChIP-seq datasets (H3K27ac, H3K4me1, H3K4me3 and H3K27me3) for TAMR cell line.

Secondly, we also agree with the Reviewer #2 that additional inclusion of H3K27ac to better define remodelled enhancers in our study would be beneficial. We have now performed these experiments in Gar15-13 tumours (3 replicates Vehicle and 3 replicates Decitabine). Correlations between replicates and between conditions are included in **new Supplementary Note Supplementary Fig. 3f** and QC statistics have been added to **new Supplementary Table 4**. We have performed analyses as suggested by Reviewer #2 and these have been added to the main manuscript (Results **lines 151-160, 281-283 and 372**; Methods **lines 737-749 and 908-921**; Discussion **lines 575-578**), **new Figure 1g-h** and **new Extended Data Fig. 1m-n**. Specifically, we have performed diffbind [1] analyses to identify differential enhancers between Vehicle- and Decitabine-treated Gar15-13 tumours and integrated this data with differential DNA methylation (**new Figure 1h**), differential ER binding (**new Extended Data Fig. 4d**) and differential chromatin interactions (**new Extended Data Fig. 3a**). Additionally, the H3K27ac track has now been added to IGV screenshots shown in **Fig. 5d-f** and **Extended Data Fig. 5d** and the data has been deposited to the public repository GEO (Accession number: GSE237769).

Our collective data, together with the addition of H3K27ac from the Vehicle- and Decitabine-treated Gar15-13 tumours, further supports our conclusion that Decitabine-induced DNA hypomethylation is directly associated with enhancer activation and gain in 3D interactions.

The authors decide to perform DNA motif enrichment analysis on the gained ‘enhancer’ contacts. Irrespective of their involvement in promoter contacts, Decitabine treatment results in a much larger set of differentially methylated regions (DMRs), and probably in the creation and loss of enhancers (that need to be identified by ChIP-seq; see above). Do the authors find identical motif enrichments in the lost methylated regions, and in the gained enhancers? If not, what is their explanation?

Response: We have now performed motif analyses at all DNA hypomethylated DMRs (**new Extended Data Fig. 4b**). Indeed, we found an overlap between motifs enriched at gained interacting enhancers and motifs enriched at hypomethylated regions (CTCF, ERE, PBX and NRF1) – with an addition of known methylation-sensitive transcription factors (AP1, Jun, NRF1 [13]) and pioneer factors for ER binding FOXA1, FOXP1 and Fos12 [2, 4, 14]. Two motifs enriched at gained interacting enhancers (ELF5 and ZNF165) were not enriched at DNA hypomethylated DMRs, suggesting that their DNA binding is not directly DNA methylation dependent. The new motif analyses were now added to **new Extended Data Fig. 4b** and Results

section **lines 351-357**: *Additionally, we compared TF motifs enriched at gained interactions (Fig. 4c) to those enriched at DNA hypomethylated DMRs (Extended Data Fig. 4b) and found a number of overlapping motifs (CTCF, ERE, PBX and NRF1) – with an addition of known methylation-sensitive transcription factors (AP1, Jun, NRF1 [13]) and pioneer factors for ER binding FOXA1, FOXP1 and Fos12 [2, 4, 14]. Together, these suggest a potential role of DNA hypomethylation in facilitating these new interactions.”.*

The most significantly enriched binding motif in the gained enhancer oEs is that of CTCF. Although the authors (line 494-502) like to conclude that this is in agreement with literature and their other data, they don't discuss that their most relevant observation related to CTCF's function, namely that contact insulation decreases and TADs dissolve in Decitabine treated samples, is highly unexpected if CTCF binds better to DNA in Decitabine treated cells. The authors must have realized this, it is somewhat disturbing that they decided to not mention this. How do the authors explain this discrepancy? They need to perform CTCF ChIP-seq experiments to understand the relationship between reduced insulation and TAD destabilization, versus altered (probably increased) CTCF recruitment.

Response: We have now performed new CTCF CUT&RUN experiments in 3 replicates Vehicle and 3 replicates Decitabine-treated Gar15-13 tumours. Correlation between replicates and conditions is shown in **new Supplementary Note Supplementary Fig. 3g**, QC metrics are included in **new Supplementary Table 4**. We used diffbind [1] to identify differentially bound CTCF sites in Vehicle- and Decitabine-treated tumours. At FDR < 5%, we identified 872 gained and 35 lost CTCF binding peaks (**new Extended Data Fig. 2j-k**).

This is not unexpected, as previous papers have shown that only small proportion of CTCF (~1-2%) is methylation-sensitive, i.e. gained following loss of DNA methylation [15-17]. To ensure the changes in CTCF binding we observed are potentially related to DNA methylation, we overlapped gained Decitabine CTCF peaks from this study with DKO vs. HCT-116 and 5-Aza K562 gained CTCF peaks from Maurano et al., study [15] and found majority (54.2%) were common “methylation sensitive” CTCF peaks between these datasets (**Rebuttal Figure 4**).

We then explored the relationship between gained, lost and common CTCF peaks and overlapping and differential TAD boundaries. Common CTCF peaks were significantly enriched at overlapping TAD boundaries (**new Extended Data Fig. 2l**). In agreement with the enrichment analyses, the average CTCF binding signal was similar at all TAD boundaries (**new Extended Data Fig. 2m**). These results have now been added to the Results section (**lines: 230-242 and 248-249**) and Discussion section

Rebuttal Figure 4

■ HCT-116 DKO (Maurano et al)
■ K562 5-aza (Maurano et al)
■ Gar15-13 PDX Decitabine

a
that

(lines: 563-566) and the data has been deposited to the public repository GEO (Accession number: GSE237769).

Line 355: “For example, at the SPATA18 locus, multiple 3D enhancer-promoter interactions are gained with Decitabine treatment, concomitant with gain in ER binding at putative enhancer, loss of DNA methylation and activation of the ER target gene (Fig. 5d and Extended Data Fig. 5a).” Authors, replace “activation of the ER target gene” by “a 1.5 fold upregulation of the ER target gene”. Upregulation is really not impressive, the gene is highly active already in vehicle treated tumors, so this wording is needed to be more precise to put things in perspective.

Response: We have now modified this in text, Results line 407 “ (...) *loss of DNA methylation and 1.5-fold upregulation of the ER target gene*”.

Line 366: “These results reveal a link between Decitabine-induced DNA hypomethylation, rewiring of ER-bound enhancer-promoter interactions and alteration in the ER transcriptional program.” See concern expressed above: ‘these results’ refer to an inspection of 3 genes. The reader cannot judge whether these are cherry picked, and such general conclusion cannot be drawn without careful systematic analyses. Authors should also comment on the levels of upregulation, and distinguish newly activated genes from genes further increasing their expression.

Response: Our conclusions are based on comprehensive genome-wide systematic analyses of multiple layers of the epigenome and transcriptome. These include statistically significant relationships between: (1) genome-wide DNA hypomethylation and enhancer activation (**new Figure 1h**) at increased ER binding (**Figure 4h** and **new Extended Data Fig. 4d**), (2) genome-wide enrichment of new enhancers at gained 3D enhancer-promoter interactions (**new Extended Data Fig. 3a**), (3) genome-wide enrichment of gained ER at new 3D enhancer-promoter interactions (**Figure 5a**), (3) genome-wide enrichment of new interactions for DEG (**Figure 4b** and **5c**). The “3 genes” presented in **Figure 5d-f** are for visualisation purposes only to better illustrate the observed genome-wide associations. **Figure 5c** already shows the levels of upregulation for all genes present at gained enhancer-promoter interactions and the DEG analyses for these genes are already included in **Supplementary Table 10**.

The recovery experiments are interesting, but unfortunately done in the MCF7 cell line instead of PDX models, which somewhat disconnects this part from the rest of the study.

Response: We could only do the recovery experiment in the TAMR cell line model as ethically we could not perform this in the PDX models as the vehicle control arm requires the mice to be sacrificed at 60 days or when tumour volume reaches 1000 mm³.

Other points:

1) logFC to score the DEG are different in figure 4b and in 7a. is there a reason for this?

Response: We have now corrected logFC cut-off for DEG on all DEG volcano (Figure 4b, Figure 7a and 7b) plots and added minor tick marks to the x-axis to make these figures more consistent.

2) Figure 2a: the extended figure 2b is more informative about the compartment switching and thus it should be exchanged with the main figure 2a.

Response: We have now modified the figures as suggested by Reviewer#2 to include both Figure 2a and Extended Figure 2b in the main figure (**new Figure 2b**).

3) There is not a western blot for DNMT1 in the figure 1 to show the effect of Decitabine. Authors perform indeed a western blot in the MCM7 cells.

Response: DNMT1 WB for cell line study was required to select an appropriate time-point to perform the DNA methylation analyses (see similar approach in <https://insight.jci.org/articles/view/137569>). Western blot is not typically required for our PDX models as they were treated with twice-weekly injections for 35 days or until ethical end-point and importantly we showed that our treatment conditions resulted in robust DNA hypomethylation.

4) The cartoon in figure 8 contains too many untested assumptions (see comments above).

Response: As both Reviewer #2 and Reviewer #3 did not find Figure 8 to be informative, we have removed it from the revised manuscript.

Reviewer #3 (Remarks to the Author):

In this paper the authors explore the effects of DNA hypomethylation on the 3D organization of the genome, ER chromatin binding and gene expression programs in endocrine resistant breast cancer patient derived xenografts and cell lines. They show that the DNA methyltransferase inhibitor decitabine inhibits the growth in mice of two ER+ human PDX models including both the Gar15-13 ER+/HER2- PDX and the HCI-005 PDX which is ER+/HER2+ and harbors a constitutively active ER mutation. They show that following 40 days of decitabine treatment that the levels of DNA methylation are significantly reduced, especially in regions of the genome that are active ER bound enhancers leading to increased ER binding at these enhancers, increased enhancer-promoter looping and gene expression. In addition, the authors perform a time-course study of DNA hypomethylation for 7 days followed by 28 days of re-methylation in the tamoxifen resistant MCF7 TAMR cell line model. The results of this experiment largely confirm the findings in the PDX models.

While the epigenomic and chromatin architecture studies are interesting, there is an important problem with the overall conclusion of the paper that the differences between control and

decitabine treated PDX represents epigenetic reprogramming. As the authors show, there is a very significant difference in the growth of the decitabine treated tumors. Thus, the cell populations present in the two conditions are likely very different and thus the epigenomic differences may be the result rather than the cause of the growth differences.

Update (24/03/2023):

The question is how do the authors rule-out the possibility that clonal selection rather than epigenetic reprogramming is responsible for the differences in epigenetic state that they observed in the PDX model at the end of therapy. Others have utilized molecular bar-coding or other genetic analyses to follow a cell population over time and under selective pressure to help distinguish these two alternatives.

Response: We agree with Reviewer #3 that it is important to rule out the possibility of clonal selection rather than epigenetic reprogramming as a main driver of observed changes. To answer this important QC question using genetic analyses, as suggested by the reviewer, we have taken advantage of the wealth of genetic and epigenetic data we have generated across different genomic technologies (new WGBS, EPIC, Hi-C) in Gar15-13 PDX samples. Specifically, we used somatic copy number changes (sCNVs) (as described in [18]) and variant allele frequency (VAF) distribution of single nucleotide variants (SNVs) to study clonal dynamics following Decitabine-treatment (as described in [19, 20]). Additionally, we used our new DNA methylation WGBS data to infer epigenetic clonality after Decitabine treatment. We have included these new analyses as a quality control step at the beginning of our study Results **lines 123-126, new Supplementary Note lines 4-76; Supplementary Fig. 1**, and Discussion **lines 501-505**. Together, our genetic and epigenetic analyses demonstrate that the PDX tumour's cellular population retained a high degree of intra-tumour clonal heterogeneity following Decitabine treatment in both Gar15-13 and HCI-005 PDX models. These data support our conclusion that epigenetic reprogramming is the main mechanism driven by the short-term, low-dose Decitabine treatment used in our study. Recently described clonal barcoding methods [21] are challenging in Patient Derived Xenograft models, however, this is beyond the scope of this paper.

In addition, while the cell line studies demonstrate similar enhancer hypomethylation at active ER enhancers following 7 days of treatment and largely remethylation following removal of decitabine for 28 days, no data on the effects of decitabine on growth in the cell line model are provided.

Response: Data on the growth response upon Decitabine treatment (increasing concentrations from 0.1 – 100uM) is already provided in Supplementary Note Supplementary Fig. 6.

Other concerns:

1) The level of ER expression in the tumors following decitabine treatment needs to be shown as increased ER levels may play a role in the expanded ER chromatin binding. This also needs to be shown in cell line studies.

Response: We did not observe a significant change in ER gene expression upon Decitabine treatment (see Supplementary Table 7). We have now added **new Extended Data Fig. 1b** to show ER quantification by IHC in Vehicle- and Decitabine-treated Gar15-tumours. In the TAMR cells, mRNA ESR1 expression is shown in **Rebuttal Figure 5**.

2) The model figure suggests that specific tumor suppressors are induced following DNA hypomethylation and are responsible for the growth suppression caused by decitabine. None of the studies in the manuscript directly address this hypothesis.

Response: As both Reviewer #2 and Reviewer #3 did not find Figure 8 to be informative, we have removed it from the revised manuscript.

1. Ross-Innes, C.S., et al., *Differential oestrogen receptor binding is associated with clinical outcome in breast cancer*. Nature, 2012. **481**(7381): p. 389-93.
2. Hurtado, A., et al., *FOXA1 is a key determinant of estrogen receptor function and endocrine response*. Nat Genet, 2011. **43**(1): p. 27-33.
3. Bi, M., et al., *Enhancer reprogramming driven by high-order assemblies of transcription factors promotes phenotypic plasticity and breast cancer endocrine resistance*. Nat Cell Biol, 2020. **22**(6): p. 701-715.
4. Farcas, A.M., et al., *Genome-Wide Estrogen Receptor Activity in Breast Cancer*. Endocrinology, 2021. **162**(2).
5. Schoenfelder, S. and P. Fraser, *Long-range enhancer-promoter contacts in gene expression control*. Nat Rev Genet, 2019. **20**(8): p. 437-455.
6. Despang, A., et al., *Functional dissection of the Sox9-Kcnj2 locus identifies nonessential and instructive roles of TAD architecture*. Nat Genet, 2019. **51**(8): p. 1263-1271.
7. Ghavi-Helm, Y., et al., *Highly rearranged chromosomes reveal uncoupling between genome topology and gene expression*. Nat Genet, 2019. **51**(8): p. 1272-1282.
8. de Wit, E., *TADs as the Caller Calls Them*. J Mol Biol, 2020. **432**(3): p. 638-642.

9. Morandi, A., et al., *GDNF-RET signaling in ER-positive breast cancers is a key determinant of response and resistance to aromatase inhibitors*. *Cancer Res*, 2013. **73**(12): p. 3783-95.
10. Cairns, J., et al., *CHiCAGO: robust detection of DNA looping interactions in Capture Hi-C data*. *Genome Biol*, 2016. **17**(1): p. 127.
11. Subramanian, A., et al., *Gene set enrichment analysis: a knowledge-based approach for interpreting genome-wide expression profiles*. *Proc Natl Acad Sci U S A*, 2005. **102**(43): p. 15545-50.
12. Khatri, P., M. Sirota, and A.J. Butte, *Ten years of pathway analysis: current approaches and outstanding challenges*. *PLoS Comput Biol*, 2012. **8**(2): p. e1002375.
13. Kreibich, E., et al., *Single-molecule footprinting identifies context-dependent regulation of enhancers by DNA methylation*. *Mol Cell*, 2023. **83**(5): p. 787-802 e9.
14. Shigekawa, T., et al., *FOXP1, an estrogen-inducible transcription factor, modulates cell proliferation in breast cancer cells and 5-year recurrence-free survival of patients with tamoxifen-treated breast cancer*. *Horm Cancer*, 2011. **2**(5): p. 286-97.
15. Maurano, M.T., et al., *Role of DNA Methylation in Modulating Transcription Factor Occupancy*. *Cell Rep*, 2015. **12**(7): p. 1184-95.
16. Lee, D.S., et al., *Simultaneous profiling of 3D genome structure and DNA methylation in single human cells*. *Nat Methods*, 2019. **16**(10): p. 999-1006.
17. Spracklin, G., et al., *Diverse silent chromatin states modulate genome compartmentalization and loop extrusion barriers*. *Nat Struct Mol Biol*, 2023. **30**(1): p. 38-51.
18. Kader, T., M. Zethoven, and K.L. Gorringer, *Evaluating statistical approaches to define clonal origin of tumours using bulk DNA sequencing: context is everything*. *Genome Biol*, 2022. **23**(1): p. 43.
19. Coorens, T.H.H., et al., *Extensive phylogenies of human development inferred from somatic mutations*. *Nature*, 2021. **597**(7876): p. 387-392.
20. Tarabichi, M., et al., *A practical guide to cancer subclonal reconstruction from DNA sequencing*. *Nat Methods*, 2021. **18**(2): p. 144-155.
21. Goyal, Y., et al., *Diverse clonal fates emerge upon drug treatment of homogeneous cancer cells*. *Nature*, 2023.

Decision Letter, first revision:

Message: Our ref: NSMB-A47134A-Z

20th Sep 2023

Dear Professor Clark,

Thank you for submitting your revised manuscript "Epigenetic therapy targets the 3D epigenome in endocrine-resistant breast cancer" (NSMB-A47134A-Z). I apologise for the delay in reaching back to you, which, as conveyed in our correspondence, came to be due to the withdrawal of one referee, holding crucial expertise, and the need to replace that expert. Nevertheless, the manuscript has now been seen by two of the original referees, and reviewer #4 who was recruited due to their well-founded expertise, similar to these of the withdrawn reviewer. The comments of the referees are below. The reviewers find that the paper has improved in revision, and therefore we'll be happy to accept it in principle in Nature Structural & Molecular Biology, pending minor revisions to satisfy the referees' final requests and to comply with our editorial and formatting guidelines.

We are now performing detailed checks on your paper and will send you a checklist detailing our editorial and formatting requirements in about two weeks. Please do not upload the final materials and make any revisions until you receive this additional information from us.

To facilitate our work at this stage, it is important that we have a copy of the main text as a word file. If you could please send along a word version of this file as soon as possible, we would greatly appreciate it; please make sure to copy the NSMB account (cc'ed above).

Sincerely,

Dimitris Typas
Associate Editor
Nature Structural & Molecular Biology
ORCID: 0000-0002-8737-1319

Reviewer #1 (Remarks to the Author):

The authors have clearly made an effort to address most of my comments. However, I have a few additional comments to the rebuttal.

1. The number of mice included in the IHC analysis in ext. fig 1b should be indicated in the legend.
2. Fig. 6h,i: The "gained & maintained" group of sites is slightly confusing. In my mind, this group should only include the 17,037 sites in the bottom venn diagram in Fig. 6i. The 8,742 sites are not gained upon Decitabine treatment but are only found upon recovery. Thus, to me, it is slightly misleading to include these sites in the "gained & maintained" category that is illustrated in Fig. 6h. This also affects the scatterplot in ext. Fig. 6d. As suggested in my original comments to the authors, I would still suggest visualizing these changes in interactions in a quantitative manner (similar to ext. fig. 6d) rather than using simple venn diagrams to show overlaps.

Reviewer #3 (Remarks to the Author):

The authors have addressed well my main critiques concerning the possibility that clonal selection rather than epigenetic reprogramming played the major role in the observed changes and whether ER levels were significantly changed by the demethylation treatment. I feel that the revised manuscript is now suitable for publication in NSMB.

Reviewer #4 (Remarks to the Author):

The authors have revised their manuscript adequately and addressed most of the points that were raised by Reviewer 2. However, from my perspective, it would be helpful if the authors would further address one comment. In response to the question "What is the relationship between the differential TADs, the DEGs, DMRs, the rewired E-P population?", the authors have performed additional analyses which are presented in Rebuttal Figure 3. It would be very helpful if the authors could incorporate this figure in their manuscript as well and comment on these findings, as I think that many readers will have this question.

Author Rebuttal, first revision:

Manuscript Nature Structural & Molecular Biology NSMB-A4713-Z

Responses to referees for revised manuscript NSMB-A47134-Z, "Epigenetic therapy targets the 3D epigenome in endocrine-resistant ER+ breast cancer" by Achinger-Kawecka et al.

We thank the reviewers for their comments, which we believe have helped us to improve our manuscript. We include below a point-by-point response to the final minor points made by the reviewers:

Reviewer #1 (Remarks to the Author):

The authors have clearly made an effort to address most of my comments. However, I have a few additional comments to the rebuttal.

1. The number of mice included in the IHC analysis in ext. fig 1b should be indicated in the legend.
2. Fig. 6h,i: The "gained & maintained" group of sites is slightly confusing. In my mind, this group should only include the 17,037 sites in the bottom venn diagram in Fig. 6i. The 8,742 sites are not gained upon Decitabine treatment but are only found upon recovery. Thus, to me, it is slightly misleading to include these sites in the "gained & maintained" category that is illustrated in Fig. 6h. This also affects the scatterplot in ext. Fig. 6d. As suggested in my original comments to the authors, I would still suggest visualizing these changes in interactions in a quantitative manner (similar to ext. fig. 6d) rather than using simple venn diagrams to show overlaps.

Response: We thank Reviewer #1 for their comments and further support of our manuscript. We have now made changes to **Extended Data Fig. 1b** and **Figure 6** as requested by Reviewer #1. Specifically,

1. We have now added the number of mice included in the ER IHC analyses to the legend in **Extended Data Fig. 1b**.
2. We thank the reviewer for their careful review of Figure 6. We have now corrected the labelling of the arrow on the diagram which now only points to the 17,037 sites in the bottom Venn diagram in **Figure 6i** as “Gained & Maintained” interactions. These ‘Gained & Maintained’ (17,037) interactions are visualised with the scatter plot in Extended Data Fig. 8d.

Reviewer #3 (Remarks to the Author):

The authors have addressed well my main critiques concerning the possibility that clonal selection rather than epigenetic reprogramming played the major role in the observed changes and whether ER levels were significantly changed by the demethylation treatment. I feel that the revised manuscript is now suitable for publication in NSMB.

Response: We thank Reviewer #3 for their support of our manuscript.

Reviewer #4 (Remarks to the Author):

The authors have revised their manuscript adequately and addressed most of the points that were raised by Reviewer 2. However, from my perspective, it would be helpful if the authors would further address one comment. In response to the question "What is the relationship between the differential TADs, the DEGs, DMRs, the rewired E-P population?", the authors have performed additional analyses which are presented in Rebuttal Figure 3. It would be very helpful if the authors could incorporate this figure in their manuscript as well and comment on these findings, as I think that many readers will have this question.

Response: We thank the reviewer for their comments and support of our manuscript. As the reviewer suggested, we have now included the additional analyses on the relationship between differential TADs, DEG, DMRs and differential interactions in the Supplementary Note and **new Supplementary Figure 4e-g** and referenced in the main manuscript (lines: 228-230) for those readers that will be interested in this question.

Final Decision Letter:

Message 15th Nov 2023

:

Dear Professor Clark,

We are now happy to accept your revised paper "The potential of epigenetic therapy to target the 3D epigenome in endocrine-resistant breast cancer" for publication as an Article in Nature Structural & Molecular Biology.

As soon as your article is published, you can generate your shareable link by entering the DOI of your article here: http://authors.springernature.com/share. Corresponding authors will also receive an automated email with the shareable link

Your paper will be published online soon after we receive proof corrections and will appear in print in the next available issue. You can find out your date of online publication by contacting the production team shortly after sending your proof corrections. Content is published online weekly on Mondays and Thursdays, and the embargo is set at 16:00 London time (GMT)/11:00 am US Eastern time (EST) on the day of publication. Now is the time to inform your Public Relations or Press Office about your paper, as they might be interested in promoting its publication. This will allow them time to prepare an accurate and satisfactory press release. Include your manuscript tracking number (NSMB-A47134B)

and our journal name, which they will need when they contact our press office.

About one week before your paper is published online, we shall be distributing a press release to news organizations worldwide, which may very well include details of your work. We are happy for your institution or funding agency to prepare its own press release, but it must mention the embargo date and Nature Structural & Molecular Biology. If you or your Press Office have any enquiries in the meantime, please contact press@nature.com.

Please note that *Nature Structural & Molecular Biology* is a Transformative Journal (TJ). Authors may publish their research with us through the traditional subscription access route or make their paper immediately open access through payment of an article-processing charge (APC). Authors will not be required to make a final decision about access to their article until it has been accepted. <https://www.springernature.com/gp/open-research/transformative-journals> Find out more about Transformative Journals

Sincerely,

Dimitris Typas
Associate Editor
Nature Structural & Molecular Biology
ORCID: 0000-0002-8737-1319